# Revisiting Class-Incremental Learning with Pre-Trained Models: Generalizability and Adaptivity are All You Need

## Abstract

Class-incremental learning (CIL) aims to adapt to emerging new classes without forgetting old ones. Traditional CIL models are trained from scratch to continually acquire knowledge as data evolves. Recently, pre-training has achieved substantial progress, making vast pre-trained models (PTMs) accessible for CIL. Contrary to traditional methods, PTMs possess generalizable embeddings, which can be easily transferred for CIL. In this work, we revisit CIL with PTMs and argue that the core factors in CIL are adaptivity for model updating and generalizability for knowledge transferring. **1)** We first reveal that frozen PTM can already provide generalizable embeddings for CIL. Surprisingly, a simple baseline (**SimpleCIL**) which continually sets the classifiers of PTM to prototype features can beat state-of-the-art even without training on the downstream task. **2)** Due to the distribution gap between pre-trained and downstream datasets, PTM can be further cultivated with adaptivity via model adaptation. We propose **AdaPt and mERge (**Aper**)**, which aggregates the embeddings of PTM and adapted models for classifier construction. Aper is a general framework that can be orthogonally combined with any parameter-efficient tuning method, which holds the advantages of PTM's generalizability and adapted model's adaptivity. **3)** Additionally, considering previous ImageNet-based benchmarks are unsuitable in the era of PTM due to data overlapping, we propose four new benchmarks for assessment, namely ImageNet-A, ObjectNet, OmniBenchmark, and VTAB. Extensive experiments validate the effectiveness of Aper with a unified and concise framework.

## 1 Introduction

With the advancement of deep learning, deep models have achieved impressive feats in many fields (He et al., 2016; Simonyan & Zisserman, 2014; Tan et al., 2020). However, most research focuses on recognizing a limited number of classes in static environments. In the real world, applications often deal with streaming data with incoming new classes (Gomes et al., 2017). To address this issue, Class-Incremental Learning (CIL) has been proposed, which allows the model to learn from the evolving data and *continuously* build a unified classification model. Nevertheless, when new classes are added sequentially, the notorious *catastrophic forgetting* occurs (French, 1999), which erases the previously learned knowledge. Many prior works (Li & Hoiem, 2017; Masana et al., 2022; De Lange et al., 2021) are designed to continually build a holistic embedding without forgetting.

While typical methods assume that the model is "*trained from scratch*," recent advancements in pre-training (Han et al., 2021) have made Pre-Trained Models (PTMs) more accessible for designing models in downstream tasks. These PTMs are often trained on massive corpus (Radford et al., 2021) or countless images (Deng et al., 2009; Ridnik et al., 2021) with handcrafted tricks (Steiner et al., 2021), resulting in strong *generalizability*. Consequently, several methods (Wang et al., 2022e;d;b; Villa et al., 2022) propose to leverage PTM for better incremental learning.

Powerful PTMs alleviate the burden of the learning process, substantially surpassing the performance upper bound of non-PTM-based methods (Zhou et al., 2023a). However, upon revisiting the objective of CIL, we find essential differences between these protocols.

Without PTMs, CIL models are trained from *random initialization* to *continually acquire* the knowledge of new classes and build a unified embedding space, which requires the **adaptivity** for sequential updating. In contrast, PTMs are trained with massive datasets, which makes it easier to achieve an ideal knowledge and embedding space with strong **generalizability**. Take the human learning process for an example; non-PTM methods aim to teach an *infant* to grow up and continually acquire knowledge through college, while PTM-based methods teach an experienced *adult* to do the same thing, which is much easier.

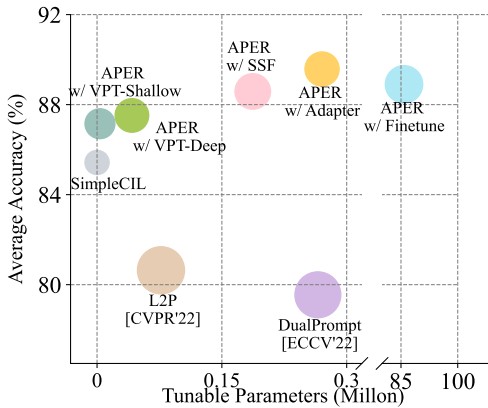

Figure 1: Comparison of different PTM-based CIL methods on VTAB dataset. X-axis stands for the number of tunable parameters, and Y-axis stands for the average accuracy. The radius stands for the training time. Although consuming more tuning parameters and training time, current state-of-the-arts (*i.e.*, L2P (Wang et al., 2022e) and DualPrompt (Wang et al., 2022d)) still show inferior performance than the baseline method SimpleCIL. By contrast, our APER consistently improves the baseline with tiny costs.

To evaluate the generalizability of PTMs, we formulate a CIL task using the VTAB (Zhai et al., 2019) dataset and test the performance of state-of-the-art PTM-based methods (Wang et al., 2022d;e) with a pre-trained ViT-B/16-IN1K in Figure 1. As a comparison, we present a simple baseline **SimpleCIL** to evaluate the quality of the pre-trained features. With the pre-trained embedding function frozen, SimpleCIL sets the classifier weights to the average embeddings (Snell et al., 2017) of each new class for classification. If PTMs already possess generalizable features, directly matching the average pattern to each query instance could also achieve competitive results. Surprisingly, we find that SimpleCIL outperforms the current SOTA by 5% even without any tuning on these downstream tasks, verifying its strong *generalizability* in knowledge transfer.

Although PTMs are generalizable for CIL, a *domain gap* may still exist between pre-trained and incremental datasets (Zhou et al., 2022b; You et al., 2020). For instance, the ImageNet pre-trained model may not generalize well to out-of-distribution (Hendrycks et al., 2021b) or specialized tasks (Alfassy et al., 2022). Under such circumstances, freezing the embedding for knowledge transferring is not a "*panacea*." Accordingly, *adaptivity* becomes essential to enable the model to grasp *task-specific features*. Nevertheless, sequentially tuning the PTM will harm the structural information and weaken the generalizability (Kumar et al., 2022), leading to the irreversible forgetting of previous knowledge. Is there a way to *unify* the generalizability of PTM with the adaptivity of the adapted model?

In this paper, we present **AdaPt and mERge** (APER) for class-incremental learning, which employs PTM to enhance generalizability and adaptivity in a unified framework. To improve adaptivity, we adapt the PTM in the first incremental stage via parameter-efficient tuning. Adapting the model helps to obtain task-specific features and fills the domain gap between PTM and incremental data. We then concatenate the adapted model with the PTM to extract average embeddings as the classifier, thereby maintaining generalizability. APER restricts model tuning in the first stage, striking a balance between adaptivity and generalizability. Moreover, typical ImageNet-based CIL benchmarks are unsuitable for evaluation due to overlapping between pre-trained and downstream tasks. Therefore, we benchmark PTM-based CIL with four new datasets that have *large* domain gaps with the pre-trained data. Extensive experiments under various settings demonstrate the effectiveness of APER.

## 2    RELATED WORK

**Class-Incremental Learning (CIL):** enables a learning system to continually incorporate new concepts without forgetting old ones (Zhou et al., 2023a). Typical CIL methods can be roughly divided into four categories. The first group saves and replays exemplars from old classes to recover former knowledge (Aljundi et al., 2019; Chaudhry et al., 2018; Iscen et al., 2020). The second group utilizes knowledge distillation to align the outputs of old and new models, thereby maintaining knowledge of old concepts (Li & Hoiem, 2017; Rebuffi et al., 2017b; Douillard et al., 2020; Zhang et al., 2020; Hu et al., 2021). The third group rectifies the inductive bias in the incremental model through normalization and logit/feature adjustment (Shi et al., 2022; Belouadah & Popescu, 2019; Pham et al., 2022). Lastly, other works expand the network when needed to enhance representation

ability (Yoon et al., 2018; Yan et al., 2021; Douillard et al., 2022; Wang et al., 2022a;d;e;b).

**CIL with PTM:** is becoming a popular topic with the increasing prevalence of PTMs (Dosovitskiy et al., 2020; Radford et al., 2021). The aim is to sequentially adjust the PTM to stream data with new classes. L2P (Wang et al., 2022e) applies visual prompt tuning (Jia et al., 2022) to CIL based on the pre-trained Vision Transformer (Dosovitskiy et al., 2020) and learns a prompt pool to select the instance-specific prompt. DualPrompt (Wang et al., 2022d) extends L2P with general and expert prompts. Different from the key-value search in L2P, CODA-Prompt (Smith et al., 2023) improves the prompt selection process with an attention mechanism. (Wang et al., 2022c) explores the anchor-based energy self-normalization strategy to aggregate multiple pre-trained classifiers. When changing ViT into CLIP (Radford et al., 2021), (Wang et al., 2022b; Villa et al., 2022) extend L2P by learning prompts for both text and image modalities (Zhou et al., 2022c).

**Parameter-Efficient Tuning for PTM:** aims to adapt the PTM to downstream tasks by tuning only a small number of (extra) parameters. Compared to fully finetuning, parameter-efficient tuning obtains competitive or even better performance at a much lower cost. VPT (Jia et al., 2022) prepends tunable prefix tokens (Li & Liang, 2021) to the input or hidden layers. LoRA (Hu et al., 2022) learns low-rank matrices to approximate parameter updates. (Houlsby et al., 2019; Chen et al., 2022a) learn extra adapter (Rebuffi et al., 2017a) modules with downsize and upsize projection. (Pfeiffer et al., 2021) merges the learned adapters with a fusion module. SSF (Lian et al., 2022) addresses the scaling and shifting operation for model tuning. Apart from additional modules in the network, (Bahng et al., 2022) proposes learning tunable parameters in the input space. Finally, (He et al., 2022a) formulates these works in a unified framework.

## 3 FROM OLD CLASSES TO NEW CLASSES

**Class-incremental learning** aims to learn from an evolving data stream with new classes to build a unified classifier (Rebuffi et al., 2017b). There is a sequence of $B$ training tasks $\left\{ \mathcal{D}^1, \mathcal{D}^2, \cdots, \mathcal{D}^B \right\}$, where $\mathcal{D}^b = \left\{ \left( \mathbf{x}_i^b, y_i^b \right) \right\}_{i=1}^{n_b}$ is the $b$-th incremental step with $n_b$ instances. Here, the training instance $\mathbf{x}_i^b \in \mathbb{R}^D$ belongs to class $y_i \in Y_b$, where $Y_b$ is the label space of task $b$. $Y_b \cap Y_{b'} = \varnothing$ for $b \neq b'$. During the $b$-th training stage, we can only access data from $\mathcal{D}^b$ for model updating. This paper focuses on the **exemplar-free** CIL setting (Zhu et al., 2021; Wang et al., 2022e), where *no historical data* can be fetched for rehearsal. The goal of CIL is to incrementally build a unified model for all seen classes, *i.e.*, acquiring knowledge from new classes and meanwhile preserving knowledge from former ones. The model's capability is evaluated over all seen classes $\mathcal{Y}_b = Y_1 \cup \cdots Y_b$ after each incremental task. Formally, the target is to fit a model $f(\mathbf{x}) : X \rightarrow \mathcal{Y}_b$ that minimizes the empirical risk across all testing datasets:

$$\sum_{(\mathbf{x}_j, y_j) \in \mathcal{D}_t^1 \cup \cdots \mathcal{D}_t^b} \ell \left( f \left( \mathbf{x}_j \right), y_j \right) , \tag{1}$$

where $\ell(\cdot, \cdot)$ measures the discrepancy between prediction and ground-truth label. $\mathcal{D}_t^b$ denotes the testing set of task $b$. A good CIL model satisfying Eq. 1 has discriminability among all classes, which strikes a balance between learning new classes and remembering old ones.

Following (Wang et al., 2022e;d), we assume the availability of a pre-trained model (*e.g.*, a ViT (Dosovitskiy et al., 2020) or ResNet (He et al., 2016)) on ImageNet (Deng et al., 2009), which we use as the initialization of $f(\mathbf{x})$. For clarity, we decouple the deep model into two parts: $f(\mathbf{x}) = W^\top \phi(\mathbf{x})$, where $\phi(\cdot) : \mathbb{R}^D \rightarrow \mathbb{R}^d$ is the embedding function and $W \in \mathbb{R}^{d \times |\mathcal{Y}_b|}$ is the classification head. We denote the classifier for class $k$ as $\mathbf{w}_k$: $W = [\mathbf{w}_1, \cdots, \mathbf{w}_{|\mathcal{Y}_b|}]$. We refer to the features after pooling as $\phi(\mathbf{x})$ for convolutional networks. In a plain ViT, the input encoding layer transforms the image into a sequence of output features $\mathbf{x}_e \in \mathbb{R}^{L \times d}$, where $L$ is the sequence length. We assume the first token in $\mathbf{x}_e$ to be the [CLS] token to simplify notation. $\mathbf{x}_e$ is then fed into the subsequent layers (*i.e.*, multi-head self-attention and MLP) to produce the final embeddings. We treat the embedded [CLS] token as $\phi(\mathbf{x})$ for ViT.

### Adaptivity and Generalizability in CIL

**CIL with Adaptivity:** Before introducing PTMs into CIL, models are trained from scratch to gradually acquire knowledge of new classes. The naive idea is to update the incremental model with cross-entropy loss, which equips the model with *adaptivity* to adapt to new tasks:

$$\mathcal{L} = \sum_{(\mathbf{x}_i, y_i) \in \mathcal{D}^b} \ell \left( f \left( \mathbf{x}_i \right), y_i \right) + \mathcal{L}_{reg} , \tag{2}$$

where $\mathcal{L}_{reg}$ stands for the regularization terms to resist forgetting, *e.g.*, knowledge distillation (Hinton et al., 2015; Li & Hoiem, 2017) or parameter regularization (Kirkpatrick et al., 2017).

**CIL with Generalizability:** With the introduction of PTM to CIL (Wang et al., 2022e), continual learners are born with *generalizability*, which can be directly transferred to downstream tasks without learning. Correspondingly, we define a simple baseline, **SimpleCIL**, to transfer PTM for incremental tasks. With the embedding function $\phi(\cdot)$ *frozen* throughout the learning process, we extract average embedding (*i.e.*, prototype (Snell et al., 2017)) of each class:

$$\mathbf{p}_i = \frac{1}{K}\sum_{j=1}^{|\mathcal{D}^b|} \mathbb{I}(y_j = i)\phi(\mathbf{x}_j)\,, \tag{3}$$

where $K = \sum_{j=1}^{|\mathcal{D}^b|} \mathbb{I}(y_j = i)$, and $\mathbb{I}(\cdot)$ is the indicator function. The averaged embedding represents the most common pattern of the corresponding class. We set the prototype as the classifier, *i.e.*, $\mathbf{w}_i = \mathbf{p}_i$, to directly adjust the PTM for CIL. SimpleCIL demonstrates competitive performance in Figure 1, confirming the strong generalizability of PTM.

**Generalizability vs. Adaptivity:** Eq. 2 and Eq. 3 address different aspects of CIL models. The former aims to enhance the adaptivity by enabling the model to be gradually tuned. By contrast, the latter highlights the model's generalizability by freezing it throughout the learning process. To understand their roles in CIL, we conduct an experiment on CIFAR100 with 20 incremental tasks and compare the performance of finetuning versus SimpleCIL. These methods are based on pre-trained ViT-B/16-IN21K, and we separately report the

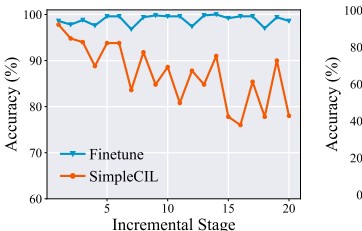 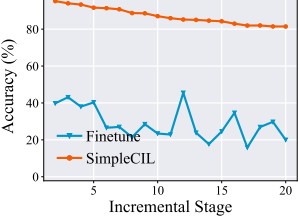

(a) Accuracy of new classes      (b) Accuracy of old classes

Figure 2: Performance of new and old classes in CIL with PTM. Sequentially finetuning the model fills the domain gap and performs better on new classes, while freezing the model has better generalizability and performs better on old classes.

performance of new ($Y_b$) and old ($\mathcal{Y}_{b-1}$) classes in Figure 2. Specifically, SimpleCIL relies on the generalizability of PTM, which works competitively even without training on the target dataset. However, it can be further improved to grasp the task-specific features, and finetuning shows better performance in new classes with the help of adaptivity. However, finetuning suffers catastrophic forgetting of old classes since features are continually changing. To summarize, these characteristics are two core aspects of CIL — adaptivity enables the model to bridge the domain gap between pre-training and incremental learning, while generalizability encourages knowledge transfer from pre-training to incremental learning. Therefore, both of them should be cultivated to facilitate CIL.

# 4 APER: AdaPt and mERge PTMs for CIL

Motivated by the potential for enhancing both generalizability and adaptivity, can we achieve these characteristics in a unified framework? Specifically, we aim to achieve this goal from two aspects. On the one hand, to bridge the domain gap between the PTM and downstream datasets, *model adaptation* is essential to move the PTM towards incremental data. On the other hand, since the adapted model may lose the generalizability of high-level features, we attempt to *merge* the adapted model and PTM into a unified network for future tasks. The merged embedding function is kept frozen throughout the incremental learning process, transferring the generalizable embedding of model sets to incoming new classes. In this way, generalizability and adaptivity are achieved in the unified framework. We first introduce the framework of APER and then discuss the specific techniques for model adaptation.

## 4.1 Training Procedure of APER

Although PTMs have discriminating features, there may exist a significant domain gap between the pre-trained dataset and incremental data. For example, the PTM is optimized to capture the characteristics of classes in ImageNet, while the incremental data stream may correspond to specialized data that requires domain knowledge or has extensive concept drift from ImageNet. To bridge this gap, an adapting process can be developed with the incremental data:

$$f^*(\mathbf{x}) = \mathcal{F}(f(\mathbf{x}), \mathcal{D}, \Theta)\,, \tag{4}$$

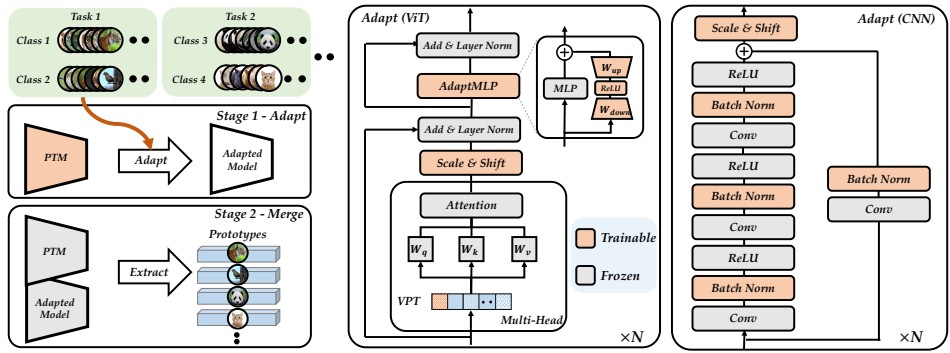

Figure 3: Illustration of APER. **Left**: the training protocol of APER. We adapt the PTM using the first stage training set $\mathcal{D}^1$ and then concatenate the embedding functions of PTM and the adapted model to maintain *generalizability* and *adaptivity*. The aggregated embedding function $[\phi^*(\cdot), \phi(\cdot)]$ is frozen throughout the following stages, and we extract the prototypes via Eq. 6 to set the classifier. **Middle**: adapting pre-trained ViT for CIL. We provide VPT Deep/Shallow, Scale & Shift, and Adapter for model adaptation. **Right**: adapting pre-trained CNN for CIL. We provide BN tuning and Scale & Shift for model adaptation. APER is a general framework that can be orthogonally combined with these adapting techniques. Red modules in the figure are trainable, while gray ones are frozen.

where the adapting algorithm $\mathcal{F}$ takes the current model $f(\mathbf{x})$ and the dataset $\mathcal{D}$ as input. It optimizes the parameter set $\Theta$ and produces the adapted model $f^*(\mathbf{x})$ that gains the domain-specific knowledge in the corresponding dataset. We introduce the variations of $\mathcal{F}$ in Section 4.2. If we could obtain all the incremental training sets at once, adapting the model via $\mathcal{F}(f(\mathbf{x}), \mathcal{D}^1 \cup \mathcal{D}^2 \cdots \cup \mathcal{D}^B, \Theta)$ can transfer the knowledge from the PTM to the incremental dataset and grasp the task-specific features for better performance.

However, since data in CIL arrive sequentially, we cannot hold all the training sets at once. Continuously adapting the model would consequently result in catastrophic forgetting (as shown in Figure 2(b)). Hence, a naive solution is to adapt the model only *in the first incremental stage*:

$$f^*(\mathbf{x}) = \mathcal{F}(f(\mathbf{x}), \mathcal{D}^1, \Theta). \tag{5}$$

Since $\mathcal{D}^1$ is a subset of the incremental data stream, it also possesses *domain-specific* knowledge that could facilitate model adaptation. The tuning process enhances the adaptivity of the CIL model, and the next question is to ensure *generalizability*. Since Eq. 5 forces the original generalizable feature to become more specialized to the downstream task, high-level features irrelevant to $\mathcal{D}^1$ shall be *overwritten and forgotten*. Therefore, a better solution is to concatenate the features extracted by the PTM and the adapted model, *i.e.*, $[\phi^*(\mathbf{x}), \phi(\mathbf{x})]$, where $\phi^*(\mathbf{x})$ and $\phi(\mathbf{x})$ stand for the adapted and pre-trained embedding functions, respectively.

To maintain generalizability, we *freeze* the concatenated embedding functions $[\phi^*(\cdot), \phi(\cdot)]$ after adaptation and extract prototypes for the following classes:

$$\mathbf{p}_i = \frac{1}{K} \sum_{j=1}^{|\mathcal{D}^b|} \mathbb{I}(y_j = i)[\phi^*(\mathbf{x}_j), \phi(\mathbf{x}_j)], \tag{6}$$

where $K = \sum_{j=1}^{|\mathcal{D}^b|} \mathbb{I}(y_j = i)$. Compared to Eq. 3, Eq. 6 contains additional information from the adapted model, which incorporates domain-specific features for better recognition. These prototypes reveal the most common patterns from the adapted and pre-trained models, ensuring both generalizability and adaptivity. We directly adopt the class prototype as the classifier weight, *i.e.*, $\mathbf{w}_i = \mathbf{p}_i$, and utilize a cosine classifier for classification: $f(\mathbf{x}) = \left(\frac{W}{\|W\|_2}\right)^\top \left(\frac{[\phi^*(\mathbf{x}), \phi(\mathbf{x})]}{\|[\phi^*(\mathbf{x}), \phi(\mathbf{x})]\|_2}\right)$. Based on the similarity between instance embedding and class prototype, it assigns a higher probability to the class with a more similar prototype.

**Effect of Adapt and Merge:** We give the visualizations of APER in Figure 3 (left). Although $\mathcal{D}^1$ is a subset of the entire training set, adapting with it still helps transfer the PTM from the upstream dataset to the downstream task. The adapting process can be viewed as a further pre-training procedure, which adapts the PTM to the incremental dataset and bridges the domain gap. By merging the embedding functions of the PTM and the adapted model, the extracted features are more representative than any one of them alone. Additionally, since the model is only trainable in the first incremental task, the efficiency of APER is comparable to SimpleCIL, which does not require sequential tuning. On the other hand, since the model is frozen in the subsequent tasks, it does not suffer catastrophic forgetting of former concepts. We give the pseudo-code of APER in Algorithm 1. In the extreme case where the adaptation process in Eq. 5 does nothing to the PTM, APER will degrade to SimpleCIL, which guarantees the performance lower bound.

## 4.2 ADAPTING THE PTM

To bridge the distribution gap between the pre-trained and incremental datasets, APER's performance depends on the effective adapting algorithm $\mathcal{F}$. In this section, we discuss six specializations of $\mathcal{F}$ in APER that can handle different types of PTMs, such as ViTs and CNNs.

**Fully Finetune**: is a naive idea when transferring the model to downstream tasks. It involves tuning all parameters in the adapting process, *i.e.*, $\Theta = \theta_\phi \cup \theta_W$, and minimizing the discrepancy between the model's output and the ground truth:

$$\min_{\theta_\phi \cup \theta_W} \sum_{(\mathbf{x}_j, y_j) \in \mathcal{D}^1} \ell\left(f\left(\mathbf{x}_j\right), y_j\right) . \tag{7}$$

However, the tuning cost could be relatively *high* for large-scale PTMs, *e.g.*, ViTs. Therefore, some parameter-efficient tuning techniques can alleviate the tuning cost and be better solutions.

**Visual Prompt Tuning (VPT) (Jia et al., 2022)**: is a lightweight tuning technique for adapting ViTs, which only prepends some learnable prompts $\mathbf{P} \in \mathbb{R}^{p \times d}$ to form the extended features $[\mathbf{P}, \mathbf{x}_e]$, where $\mathbf{x}_e$ is the encoded features of the input image. The extended features are then fed into the subsequent layers of ViT to calculate the final embeddings. There are two variations of VPT: **VPT-Deep**, which prepends the prompts at every attention layer, and **VPT-Shallow**, which only prepends the prompts at the first layer. During optimization, it freezes the pre-trained weights in the embedding function and optimizes these prompts and classification head, *i.e.*, $\Theta = \theta_\mathbf{P} \cup \theta_W$.

**Scale & Shift (SSF) (Lian et al., 2022)**: aims to adjust the feature activation by scaling and shifting. It appends an extra SSF layer after each operation layer (*i.e.*, MSA and MLP) and adjusts the output of these operations. Given the input $\mathbf{x}_i \in \mathbb{R}^{L \times d}$, the output $\mathbf{x}_o \in \mathbb{R}^{L \times d}$ is formulated as:

$$\mathbf{x}_o = \gamma \otimes \mathbf{x}_i + \beta , \tag{8}$$

where $\gamma \in \mathbb{R}^d$ and $\beta \in \mathbb{R}^d$ are the scale and shift factors, respectively. $\otimes$ is Hadamard product (element-wise multiplication). The model optimizes the SSF layers and classifier, *i.e.*, $\Theta = \theta_{SSF} \cup \theta_W$, to trace the features of new tasks.

**Adapter (Houlsby et al., 2019; Chen et al., 2022a)**: is a bottleneck module which contains a down-projection $W_{\text{down}} \in \mathbb{R}^{d \times r}$ to reduce the feature dimension, a non-linear activation function, and an up-projection $W_{\text{up}} \in \mathbb{R}^{r \times d}$ to project back to the original dimension. We follow (Chen et al., 2022a) to equip the original MLP structure in ViT with the adapter. Denote the input of the MLP layer as $\mathbf{x}_\ell$, the output of AdaptMLP is formatted as:

$$\text{MLP}(\mathbf{x}_\ell) + \text{ReLU}(\mathbf{x}_\ell W_{\text{down}}) W_{\text{up}} . \tag{9}$$

With pre-trained weights frozen, it optimizes the adapter and classification head, *i.e.*, $\Theta = \theta_{W_{\text{down}}} \cup \theta_{W_{\text{up}}} \cup \theta_W$.

**Batch Normalization Tuning**: If the PTM is a convolutional network, *e.g.*, CNNs, we can adjust the BN (Ioffe & Szegedy, 2015) parameters. Since the running mean and variance in BN are compatible with the upstream data distribution, they could be *unstable* for downstream tasks. Correspondingly, we can reset the running statistics in BN and adapt to the current data via forward passing. No backpropagation is required, making it quick and simple for the pre-trained model.

**Discussions:** We visualize the adapting process of APER in Figure 3. Compared to fully finetuning, parameter-efficient tuning adjusts the PTM towards the downstream task and preserves its generalizability. The adapted model can capture the specialized features in the incremental data, leading to better adaptivity. Since L2P and DualPrompt are based on pre-trained ViT, they cannot be deployed with CNN. In contrast, APER is a general framework that efficiently handles diverse structures. Specifically, APER can be combined with VPT/SSF/Adapter for ViT and SSF/BN Tuning for CNN. Since APER adopts the prototype-based classifier, the linear classifier $W$ will be dropped after adaptation.

## 5 EXPERIMENTS

This section compares APER with SOTA methods on benchmark datasets to show the superiority. Due to the overlap between pre-trained datasets and traditional CIL benchmarks, we also advocate four new benchmarks for evaluating PTM-based methods. Ablations and visualizations verify the effectiveness of APER with new classes. We also explore the performance of different PTMs in CIL. More details and extra results are included in Section C.

Table 1: Average and last performance comparison on seven datasets with **ViT-B/16-IN21K** as the backbone. 'IN-R/A' stands for 'ImageNet-R/A,' 'ObjNet' stands for 'ObjectNet,' and 'OmniBench' stands for 'OmniBenchmark.' We report more results in Section D. The best performance is shown in bold.

| Method | CIFAR B0 Inc5 | | CUB B0 Inc10 | | IN-R B0 Inc5 | | IN-A B0 Inc10 | | ObjNet B0 Inc10 | | OmniBench B0 Inc30 | | VTAB B0 Inc10 | |
|---|---|---|---|---|---|---|---|---|---|---|---|---|---|---|
| | $\bar{\mathcal{A}}$ | $\mathcal{A}_B$ | $\bar{\mathcal{A}}$ | $\mathcal{A}_B$ | $\bar{\mathcal{A}}$ | $\mathcal{A}_B$ | $\bar{\mathcal{A}}$ | $\mathcal{A}_B$ | $\bar{\mathcal{A}}$ | $\mathcal{A}_B$ | $\bar{\mathcal{A}}$ | $\mathcal{A}_B$ | $\bar{\mathcal{A}}$ | $\mathcal{A}_B$ |
| Finetune | 38.90 | 20.17 | 26.08 | 13.96 | 21.61 | 10.79 | 21.60 | 10.96 | 19.14 | 8.73 | 23.61 | 10.57 | 34.95 | 21.25 |
| Finetune Adapter | 60.51 | 49.32 | 66.84 | 52.99 | 47.59 | 40.28 | 43.05 | 37.66 | 50.22 | 35.95 | 62.32 | 50.53 | 48.91 | 45.12 |
| LwF | 46.29 | 41.07 | 48.97 | 32.03 | 39.93 | 26.47 | 35.39 | 23.83 | 33.01 | 20.65 | 47.14 | 33.95 | 40.48 | 27.54 |
| SDC | 68.21 | 63.05 | 70.62 | 66.37 | 52.17 | 49.20 | 26.65 | 23.57 | 39.04 | 29.06 | 60.94 | 50.28 | 45.06 | 22.50 |
| L2P | 85.94 | 79.93 | 67.05 | 56.25 | 66.53 | 59.22 | 47.16 | 38.48 | 63.78 | 52.19 | 73.36 | 64.69 | 77.11 | 77.10 |
| DualPrompt | 87.87 | 81.15 | 77.47 | 66.54 | 63.31 | 55.22 | 52.56 | 42.68 | 59.27 | 49.33 | 73.92 | 65.52 | 83.36 | 81.23 |
| CODA-Prompt | 89.11 | 81.96 | 84.00 | 73.37 | 64.42 | 55.08 | 48.51 | 36.47 | 66.07 | 53.29 | 77.03 | 68.09 | 83.90 | 83.02 |
| SimpleCIL | 87.57 | 81.26 | 92.20 | **86.73** | 62.58 | 54.55 | 60.50 | 49.44 | 65.45 | 53.59 | 79.34 | 73.15 | 85.99 | 84.38 |
| APER w/ Finetune | 87.67 | 81.27 | 91.82 | 86.39 | 70.51 | 62.42 | 61.57 | 50.76 | 61.41 | 48.34 | 73.02 | 65.03 | **87.47** | 80.44 |
| APER w/ VPT-Shallow | 90.43 | 84.57 | 92.02 | 86.51 | 66.63 | 58.32 | 57.72 | 46.15 | 64.54 | 52.53 | 79.63 | 73.68 | 87.15 | **85.36** |
| APER w/ VPT-Deep | 88.46 | 82.17 | 91.02 | 84.99 | 68.79 | 60.48 | 60.59 | 48.72 | 67.83 | 54.65 | **81.05** | **74.47** | 86.59 | 83.06 |
| APER w/ SSF | 87.78 | 81.98 | 91.72 | 86.13 | 68.94 | 60.60 | **62.81** | **51.48** | 69.15 | **56.64** | 80.53 | 74.00 | 85.66 | 81.92 |
| APER w/ Adapter | **90.65** | **85.15** | **92.21** | **86.73** | **72.35** | **64.33** | 60.53 | 49.57 | 67.18 | 55.24 | 80.75 | 74.37 | 85.95 | 84.35 |

## 5.1 IMPLEMENTATION DETAILS

**Dataset**: Following (Wang et al., 2022d; Yu et al., 2020), we evaluate the performance on CIFAR100 (Krizhevsky et al., 2009), CUB200 (Wah et al., 2011), and ImageNet-R (Hendrycks et al., 2021a). Since PTMs are often trained with ImageNet21K (Deng et al., 2009), evaluating PTM-based methods with ImageNet is meaningless. Hence, we advocate four new datasets that have *large domain gap* with ImageNet, namely ImageNet-A (Hendrycks et al., 2021b), ObjectNet (Barbu et al., 2019), Omnibenchmark (Zhang et al., 2022) and VTAB (Zhai et al., 2019). Among them, ImageNet-A and ObjectNet contain *challenging samples* that ImageNet pre-trained models cannot handle, while Omnibenchmark and VTAB contain diverse classes from multiple *complex realms*. To construct the CIL task, we sample 200 classes from ObjectNet and ImageNet-A, and 300 from Omnibenchmark. We sample 5 datasets from VTAB, each containing 10 classes, to construct the cross-domain CIL setting. Following (Rebuffi et al., 2017b), we shuffle the classes with the same random seed and split them into 'B/Base-$m$, Inc-$n$.' It means the first dataset contains $m$ classes, and each following dataset contains $n$ classes. $m = 0$ means the total classes are equally divided into each task.
**Comparison methods:** We first compare to SOTA PTM-based CIL methods L2P (Wang et al., 2022e), DualPrompt (Wang et al., 2022d), and CODA-Prompt (Smith et al., 2023). We also modify classical CIL methods LwF (Li & Hoiem, 2017), SDC (Yu et al., 2020), iCaRL (Rebuffi et al., 2017b), LUCIR (Hou et al., 2019), DER (Yan et al., 2021), FOSTER (Wang et al., 2022a), MEMO (Zhou et al., 2023b), FACT (Zhou et al., 2022a) to **utilize the same PTM as the initialization**. Apart from SimpleCIL, we also report the baseline, sequentially tuning the model, denoted as Finetune.
**Training details:** We use PyTorch (Paszke et al., 2019) to *deploy all models on Tesla V100 with the same network backbone*. As there are various PTMs publicly available (Wightman, 2019), we follow (Wang et al., 2022e;d) to choose the most representative ones, denoted as **ViT-B/16-IN1K** and **ViT-B/16-IN21K.** Both are pre-trained on ImageNet21K, while the former is additionally finetuned on ImageNet1K. During adaptation, we train the model with a batch size of 48 for 20 epochs and use SGD with momentum for optimization. The learning rate starts from 0.01 and decays with cosine annealing. The prompt length $p$ is 5 for VPT, and the projection dim $r$ is 16 for Adapter. The source code will be publicly available upon acceptance.
**Evaluation Metrics:** Denote the accuracy after the $b$-th stage as $\mathcal{A}_b$, we follow (Rebuffi et al., 2017b) to use $\mathcal{A}_B$ (last stage performance) and $\bar{\mathcal{A}} = \frac{1}{B} \sum_{b=1}^{B} \mathcal{A}_b$ (average performance) for evaluation.

## 5.2 BENCHMARK COMPARISON

We report the incremental performance against SOTA methods in Table 1, where all methods are based on the pre-trained ViT-B/16-IN21K. We also train these models with pre-trained ViT-B/16-IN1K and show the incremental trend in Figure 4(a)∼4(f). These data splits include settings with large and small base classes for a holistic evaluation.

Firstly, we can infer that the embeddings of PTMs are generalizable and can be directly applied for CIL to beat the SOTA. Specifically, the baseline SimpleCIL outperforms DualPrompt by **20%** on CUB and **8%** on ImageNet-A in terms of $\mathcal{A}_B$. However, strong PTMs can be further improved if they are adapted by APER, as downstream tasks have a large domain gap with the pre-trained dataset. Specifically, we find APER *consistently* outperforms SimpleCIL in seven benchmark datasets. In contrast, sequentially finetuning the model suffers severe forgetting, which verifies the effectiveness of the adapt and merge protocol. Since APER only requires tuning the PTM in the first stage, it requires less training time and extra parameters than L2P and DualPrompt, as shown in Figure 1. Among the variations of adapting techniques, we find *SSF and Adapter are more efficient than VPT*. We also

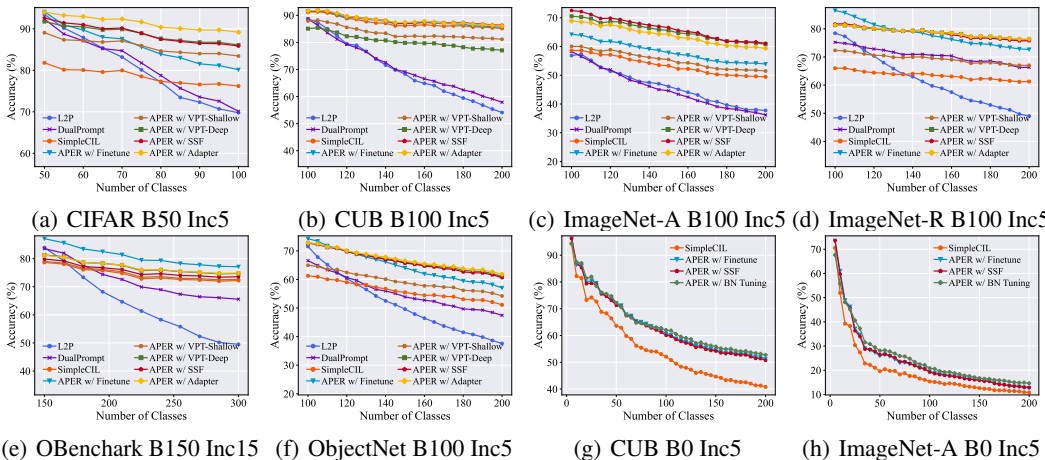

Figure 4: **(a)∼(f):** Incremental performance with **ViT-B/16-IN1K** as the backbone when half of the total classes are base classes. **(g)∼(h):** Incremental performance when using **ResNet18** as backbone. Since L2P and Dualprompt cannot be deployed with ResNet, we do not report their performance. APER consistently improves the performance of different backbones, *i.e.*, ViT and CNN. 'OBenchmark' stands for 'OmniBenchmark.'

compare to SOTA traditional CIL methods and modify their backbones into pre-trained ViT for a fair comparison. However, we can infer from Table 2 that these methods work poorly without exemplars.

Apart from ViTs, APER also works well with pre-trained CNNs. We adopt the pre-trained ResNet18 (He et al., 2016) for evaluation and plot the incremental performance in Figure 4(g),4(h). Results show that APER consistently boosts the performance of pre-trained ViTs and CNNs. See full results in Section D.

Lastly, as shown in Table 1, the performance on typical benchmarks is approaching saturation as they have a small domain gap with ImageNet. By contrast, due to the large domain gap between our newly established benchmarks and ImageNet, there is still space for improvement, indicating the effectiveness and necessity of these new benchmarks. We also consider a more challenging TV series incremental learning task in Section C.1.

Table 2: Comparison to SOTA classical CIL methods with **ViT-B/16-IN1K**. *All methods are deployed without exemplars.*

| Method | ObjectNet B0 Inc20 | | ImageNet-A B0 Inc20 | |
|---|---|---|---|---|
| | $\bar{\mathcal{A}}$ | $\mathcal{A}_B$ | $\bar{\mathcal{A}}$ | $\mathcal{A}_B$ |
| iCaRL | 33.43 | 19.18 | 29.22 | 16.16 |
| LUCIR | 41.17 | 25.89 | 31.09 | 18.59 |
| DER | 35.47 | 23.19 | 33.85 | 22.27 |
| FOSTER | 37.83 | 25.07 | 34.82 | 23.01 |
| MEMO | 38.52 | 25.41 | 36.37 | 24.46 |
| FACT | 60.59 | 50.96 | 60.13 | 49.82 |
| SimpleCIL | 62.11 | 51.13 | 59.67 | 49.44 |
| APER w/ SSF | **68.75** | **56.79** | **63.59** | **52.67** |

### 5.3 ABLATION STUDY

**Downscale features:** Since the feature of APER is aggregated with PTM and adapted model, which is twice that of a PTM. We conduct an ablation with APER w/ SSF on CIFAR100 Base50 Inc5 to show whether these features are essential for CIL. Specifically, we train a PCA (Pearson, 1901) model in the first stage to reduce embedding dimension for the following stages. Denote the target dimension as $k$, we train the PCA model $\text{PCA}([\phi^*(\mathbf{x}), \phi(\mathbf{x})]) : \mathbb{R}^d \to \mathbb{R}^k$, and append it to the feature extractor. Hence, the features and prototypes are projected to $k$ dimensions. We plot the performance with the change of $k$ in Figure 5(a). Specifically, APER obtains competitive performance to DualPrompt (with 768 dims) even if the features are projected to 50 dims. We also experiment by randomly sampling $k$ features from the original feature space and report the results in Figure 5(b). The conclusions are consistent with the former ones, showing that randomly sampling 200 dimensions of APER achieves the same performance scale as DualPrompt. The accuracy-dimension curves are shown in Figure 5(c).

**Sub-modules:** Since APER is concatenated with PTM and adapted model, we conduct ablations on ImageNet-A Base100 Inc5 with ViT-B/16-IN21K to compare APER w/ Finetune and its sub-modules. Specifically, we build SimpleCIL with $\phi(\cdot)$ and $\phi^*(\cdot)$, respectively, denoted as **SimpleCIL-PTM** and **SimpleCIL-Adapted**. The former represents the capability of PTM, while the latter stands for the power of the adapted model. Both are compositional modules in APER. Besides, we build SimpleCIL based on concatenated pre-trained ViT-B/16-IN21K and ViT-B/16-IN1K, denoted as **SimpleCIL-21K+1K**. It utilizes the aggregated features of two embedding functions, which has the same dimension as APER. As shown in Figure 5(d), SimpleCIL-Adapted outperforms SimpleCIL-PTM, indicating the importance of model adaptation. However, adapting the model also overwrites the high-level features, which reduces the model's generalizability. The adapted model suffers larger

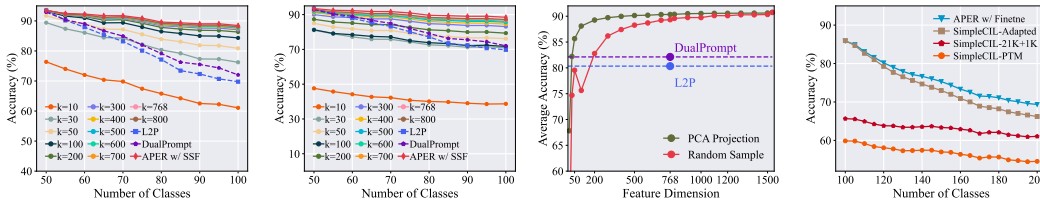

(a) PCA projected features (b) Randomly sampled features (c) Projected dimension and accuracy (d) Compositional components

Figure 5: Ablation study. We use PCA or random sample to downscale the dimension of aggregated embeddings in (a)∼(c). We also compare APER to its sub-modules for ablation in (d).

performance degradation than vanilla SimpleCIL, indicating the effect of generalizability in resisting forgetting. APER outperforms every sub-module with unified adaptivity and generalizability.

**Different PTMs:** Observing the performance gap between ViT-B/16-IN21K and ViT-B/16-IN1K, we seek to explore different kinds of PTMs on ImageNet-R Base0 Inc20. We choose publicly available PTMs, *i.e.*, ResNet18/50/152 (He et al., 2016), ViT-B/16-IN1K/21K, ViT-L/16-IN1K, ViT-B/16-DINO (Caron et al., 2021), ViT-B/16-SAM (Chen et al., 2022b), ViT-B/16-MAE (He et al., 2022b), ViT-B/16-CLIP (Radford et al., 2021) (image encoder) for a holistic evaluation, and report the results in Figure 6. We can draw three main conclusions. **1)** Pre-trained ViTs show better generalizability than ResNets. **2)** Larger ViTs generalize better than small ones, and ViTs trained with supervised loss perform better than unsupervised ones. **3)** Owing to the massive training corpus and the contrastive loss, CLIP performs better than ImageNet21K pre-trained ViTs. Finally, we find APER w/ Finetune consistently improves the performance of SimpleCIL for any PTM, thus validating its effectiveness.

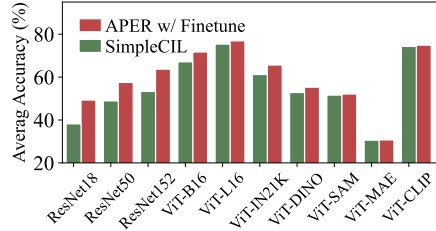

Figure 6: CIL with different kinds of PTMs. APER consistently improves the performance of different PTMs.

**Visualizations:** We visualize the learned decision boundaries with t-SNE (Van der Maaten & Hinton, 2008) on CIFAR100 dataset between two incremental stages, as shown in Figure 7(a), 7(b). We visualize the classes from the first and second incremental tasks with colorful dots and triangles. Correspondingly, the class prototypes are represented by squares. As we can infer from these figures, PTM works competitively, which well separates the instances into their corresponding classes. The class prototypes are situated at the center of each class, verifying their representativeness in recognition. When extending the model from the first to the second stage, we find APER performs well on both old and new classes. Visualizations verify the generalizability and adaptivity of APER. More visualizations are shown in Section C.4.

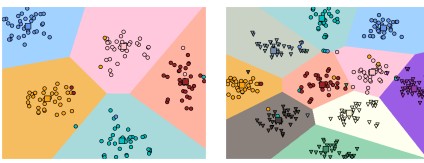

(a) First stage (b) Second stage

Figure 7: Visualization of the decision boundary on CIFAR100 between two incremental tasks. Old classes are shown in dots, and new classes are shown in triangles. Decision boundaries are shown with the shadow region.

## 6 CONCLUSION

Learning with incremental classes is of great importance in real-world applications, which requires *adaptivity* for updating and *generalizability* for knowledge transfer. In this paper, we systematically revisit CIL with PTMs and draw three conclusions. Firstly, a frozen PTM can provide *generalizable* embeddings for CIL, enabling a prototype-based classifier to outperform the current state-of-the-art. Secondly, due to the distribution gap between pre-trained and downstream datasets, PTMs can be further harnessed to enhance their *adaptivity*. To this end, we propose APER, which can be orthogonally combined with any parameter-efficient tuning method to unify generalizability and adaptivity for CIL. Lastly, due to data overlapping, traditional ImageNet-based benchmarks are unsuitable for evaluation in the era of PTM. Hence, we propose four new benchmarks to evaluate PTM-based CIL methods. Extensive experiments verify APER's state-of-the-art performance. Future work includes exploring task-specific tuning methods and structures.

**Limitations** include the restriction of exemplars. It turns into exemplar-based CIL if sufficient old class instances are available, where adaptivity can be further addressed through data rehearsal.

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

# Supplementary Material

Class-incremental learning (CIL) is of great importance to the machine learning community, and Pre-Trained Models (PTMs) have boosted its performance in recent years. In the main paper, we revisit PTM-based CIL and draw three conclusions. **1)** We empirically prove that frozen PTM can provide *generalizable* embeddings for CIL. To our surprise, a simple baseline (**SimpleCIL**) that continually sets the classifiers of PTM to prototype features can beat state-of-the-art performance even without training on the downstream task. **2)** We show that PTM can be further cultivated with *adaptivity* through model adaptation. We propose **AdaPt and mERge** (APER), which holds the advantages of PTM's generalizability and adapted model's adaptivity. **3)** Previous benchmarks are unsuitable in the era of PTM due to data overlapping. Hence, we propose four new benchmarks, namely ImageNet-A, ObjectNet, OmniBenchmark, and VTAB, that have a large domain gap with ImageNet pre-trained model for evaluation.

In the supplementary, we provide more details about the experimental results mentioned in the main paper, as well as additional empirical evaluations and discussions. The supplementary material is organized as follows:

- Section A provides the background information on vision transformer and parameter-efficient tuning methods. It also includes a discussion on model tuning in APER and a discussion about related concepts in continual learning. We also summarize and discuss the main contributions of this paper.

- Section B discusses the implementation details in the main paper, including the introduction of compared methods, hyper-parameters, pre-trained models, and datasets.

- Section C presents additional ablations of APER, including the influence of adapting stages, hyper-parameters, and classifier types. It also contains extra experimental evaluations that could not be included in the main paper due to page limits, such as more visualizations of Grad-CAM. Additionally, we also create a fine-grained TV series classification dataset that strictly has no data overlapping with the pre-training dataset. We also supply the incremental learning results on this new dataset.

- Section D reports the full experimental results of 7 datasets and 45 splits, including the numerical results and performance curves. It also includes experiments with pre-trained ResNets.

## A  BACKGROUNDS ABOUT ViT AND PARAMETER-EFFICIENT TUNING

In this section, we provide background information on vision transformers and parameter-efficient tuning techniques adopted in the main paper.

### A.1  VISION TRANSFORMER

The concept of Vision Transformers (ViTs) is first introduced in (Dosovitskiy et al., 2020) to the computer vision field. In a plain ViT, an RGB image $\mathbf{x} \in \mathbb{R}^{3 \times H \times W}$ is first divided into non-overlapping patches, where $(H, W)$ denotes the *height* and *width* of the input image. These patches are then appended with a class token `[CLS]` and then fed into an embedding layer followed by the vision transformer blocks with self-attention (Vaswani et al., 2017) as the core operation. Following the notations in the main paper, we denote the features after the embedding layer as $\mathbf{x}_e \in \mathbb{R}^{L \times d}$, and the first element in $\mathbf{x}_e$ is the `[CLS]` token. $L$ is the length of sequence, and $d$ is the embedding dim.

Each vision transformer block mainly consists of two modules, *i.e.*, a multi-head self-attention layer (MSA) and a two-layer MLP. In MSA, the tokens $\mathbf{x}_e$ are first linearly projected into query $Q \in \mathbb{R}^{L \times d}$, key $K \in \mathbb{R}^{L \times d}$, and value $V \in \mathbb{R}^{L \times d}$. The self-attention is performed via:

$$\text{Attention}(Q, K, V) = \text{Softmax}\left(\frac{QK^T}{\sqrt{d}}\right) V. \tag{10}$$

The output tokens are then sent to a LayerNorm (Ba et al., 2016) and the MLP block. Denote the output of Eq. 10 as $\mathbf{x}_\ell$, this process is formulated as:

$$\mathbf{x}_\ell + \text{MLP}(\text{LN}(\mathbf{x}_\ell)), \tag{11}$$

which aggregates the projected features and the original features for further information extraction. After the transformation of $N$ cascaded transformer blocks, ViT takes the `[CLS]` token as the feature for final recognition. We refer the readers to the original work (Dosovitskiy et al., 2020) for more details about ViT.

## A.2 PARAMETER-EFFICIENT TUNING

In this section, we introduce more details about the parameter-efficient tuning techniques adopted in the main paper, including visual prompt tuning (VPT) (Jia et al., 2022), adapters (Houlsby et al., 2019; Chen et al., 2022a), scale & shift (SSF) (Lian et al., 2022), and batch normalization tuning. Before addressing these methods, we revisit the target of parameter-efficient tuning, which aims to adapt the model with the least tunable parameters. Denote the tuning process as:

$$f^*(\mathbf{x}) = \mathcal{F}(f(\mathbf{x}), \mathcal{D}^1, \Theta), \tag{12}$$

which freezes the parameters in the pre-trained model and only adjusts the parameters in $\Theta$. In the main paper, we optimize the selected parameters via cross-entropy loss:

$$\min_\Theta \sum_{(\mathbf{x}_j, y_j) \in \mathcal{D}^1} \ell\left(f\left(\mathbf{x}_j\right), y_j\right). \tag{13}$$

### A.2.1 VISUAL PROMPT TUNING (VPT)

VPT (Jia et al., 2022) aims to prepend some learnable prompts $\mathbf{P} \in \mathbb{R}^{p \times d}$ to form the extended embedding features $[\mathbf{P}, \mathbf{x}_e]$, where $\mathbf{x}_e$ is the encoded features of the input image. The extended features are then fed into the subsequent transformer blocks of ViT to calculate the final embeddings. Depending on where the prompts are inserted, VPT can be further divided into two types: **VPT-Deep** and **VPT-Shallow**.

Specifically, VPT-Shallow only learns the prompts in the first transformer block. Denote the function of $k$-th transformer block as $L_k$, the operations of VPT-Shallow can be denoted as:

$$\begin{aligned}
[\mathbf{Z}_1, \mathbf{E}_1] &= L_1\left([\mathbf{P}, \mathbf{x}_e]\right) \\
[\mathbf{Z}_i, \mathbf{E}_i] &= L_i\left([\mathbf{Z}_{i-1}, \mathbf{E}_{i-1}]\right) \quad i = 2, 3, \ldots, N,
\end{aligned} \tag{14}$$

where $\mathbf{Z}_i \in \mathbb{R}^{p \times d}$ denotes the encoded feature of prompts, $N$ is the number of transformer blocks. Correspondingly, VPT-Deep learns the prompts in each transformer block:

$$[\_, \mathbf{E}_i] = L_i\left([\mathbf{P}_{i-1}, \mathbf{E}_{i-1}]\right) \quad i = 1, 2, \ldots, N, \tag{15}$$

where $\mathbf{E}_0 = \mathbf{x}_e$ is the encoded feature of image patches. During optimization, VPT freezes the pre-trained weights in the embedding layer and only optimizes these learnable prompts and classification head[1], *i.e.*, $\Theta = \theta_{\mathbf{P}} \cup \theta_W$.

Specifically, the number of tunable parameters in VPT-Shallow is $p \times d$, and that of VPT-Shallow is $p \times d \times N$, where $N$ is the number of transformer blocks. For example, we set the prompt length to 5 for the ViT-B/16 model. The tunable parameters are $5 \times 786 = 0.004$ million in VPT-Shallow and $5 \times 786 \times 12 = 0.046$ million for VPT-Deep, which is negligible compared to the ViT-B/16 with 86 million parameters.

### A.2.2 ADAPTER

Adapter (Houlsby et al., 2019; Chen et al., 2022a; Rebuffi et al., 2017a) is a bottleneck module that allows adjusting the output of ViT. Formally, it comprises a down-projection $W_{\text{down}} \in \mathbb{R}^{d \times r}$ to reduce the feature dimension, a non-linear activation function, and an up-projection $W_{\text{up}} \in \mathbb{R}^{r \times d}$ to project back to the original dimension. Following the implementation of AdaptFormer (Chen et al., 2022a), we replace the original MLP structure in ViT with the AdaptMLP structure. Specifically, denote the input of the MLP layer as $\mathbf{x}_\ell$, the output of AdaptMLP is formatted as:

$$\text{MLP}(\mathbf{x}_\ell) + s \cdot \text{ReLU}(\mathbf{x}_\ell W_{\text{down}}) W_{\text{up}}, \tag{16}$$

which is a residual structure. $s$ is an optional learnable parameter for re-scaling the output. During adaptation, the model freezes the pre-trained weights and only optimizes the extra parameters, *i.e.*, $\Theta = \theta_{W_{\text{down}}} \cup \theta_{W_{\text{up}}} \cup \theta_W$. Specifically, we set the hidden dim $r$ to 16, and the number of tunable parameters in the Adapter is approximately 0.3 million, which is negligible compared to the ViT-B/16 with 86 million parameters.

---

[1]Note that the classification head $W$ is only optimized in the adaptation process. Since we use a prototype-based classifier for classification after model merge, $W$ will be dropped after adaptation.

### A.2.3 SCALE & SHIFT

Scale & Shift (SSF) (Lian et al., 2022) aims to adjust the feature activation by scaling and shifting operations. SSF only appends an extra SSF layer after each operation layer (*e.g.*, MSA and MLP) and adjusts the output of these operations. Given the input $\mathbf{x}_i \in \mathbb{R}^{L \times d}$, the output $\mathbf{x}_o \in \mathbb{R}^{L \times d}$ follows:

$$\mathbf{x}_o = \gamma \otimes \mathbf{x}_i + \beta \,, \tag{17}$$

where $\gamma \in \mathbb{R}^d$ and $\beta \in \mathbb{R}^d$ are the scale and shift factors, respectively. $\otimes$ is Hadamard product (element-wise multiplication). The model optimizes the SSF layers and classification head, *i.e.*, $\Theta = \theta_{SSF} \cup \theta_W$, to trace the features of new tasks. In the implementation, we add the SSF layer after the MSA and MLP operations, and the number of tunable parameters in SSF is around $0.2$ million, which is negligible compared to the ViT-B/16 with $86$ million parameters.

Note that the SSF layer can also be deployed for the pre-trained residual networks. We append SSF layers after the residual blocks to re-scale the output for better adaptation.

### A.2.4 BATCH NORMALIZATION TUNING

If the PTM uses ResNet structures, we can also adjust the BN (Ioffe & Szegedy, 2015) parameters, including the running mean and running variance. Specifically, BN is designed to normalize each feature dimension and overcome internal covariate shift. It records the running mean and variance in the training stage:

$$
\begin{aligned}
\mu_{\mathcal{B}} &= \frac{1}{m} \sum_{i=1}^{m} x_i \\
\sigma_{\mathcal{B}}^2 &= \frac{1}{m} \sum_{i=1}^{m} (x_i - \mu_{\mathcal{B}})^2 \,,
\end{aligned}
\tag{18}
$$

where $x_i$ denotes the feature of the $i$-th instance and $m$ is the batch size. $\mu_{\mathcal{B}}$ and $\sigma_{\mathcal{B}}^2$ are then utilized to normalize the instances in the testing process. However, since the running mean and variance in PTM are compatible with the upstream data distribution, they could be *unstable* for the downstream tasks, leading to abnormal outputs (Sun et al., 2021; Niu et al., 2023).

To handle the incompatible BN statistics in pre-trained and incremental datasets, we first zero the running statistics in BN and then forward pass the data in $\mathcal{D}^1$ to record the current data distribution. In other words, BN can be updated by feeding the model with new class instances, *i.e.*, $f(\mathbf{x})$. These statistical parameters are then fit for the current data to alleviate the domain gap. Notably, BN tuning does not require backpropagation, making it an efficient solution.

### A.3 SUMMARY OF PTM TUNING

Parameter-efficient tuning enables the model adaptation with the least number of tunable parameters, which guarantees the model's *adaptivity*. Specifically, we also observe that fully finetuning could fail in cases where training data is rare. In such circumstances, parameter-efficient tuning could be a better solution for model adaptation. Accordingly, we notice APER w/ Finetune performs worse than SimpleCIL on CUB B0 Inc10, ObjectNet B0 Inc10, and OmniBenchmark B0 Inc30, while APER with other tuning methods perform better. On the other hand, since parameter-efficient tuning freezes the pre-trained weights, the *generalizability* of the model is also maintained. It must be noted that the PTM in the model merge is the *same* as the PTM before the model adaptation.

In summary, APER is a general framework for class-incremental learning, which can be applied with different types of backbones and tuning techniques. Specifically, we can use VPT-Deep, VPT-Shallow, Scale & Shift, Adapter for ViT, and Scale & Shift and BN tuning for CNN. In the implementation, we only choose one specific tuning structure among these techniques and do not consider the combination of multiple tuning methods.

### A.4 DISCUSSIONS ABOUT RELATED CONCEPTS IN CONTINUAL LEARNING

There is a famous trade-off in class-incremental learning, namely "stability-plasticity dilemma" (Grossberg, 2012; Mermillod et al., 2013; Mirzadeh et al., 2020). Among them, "stability"

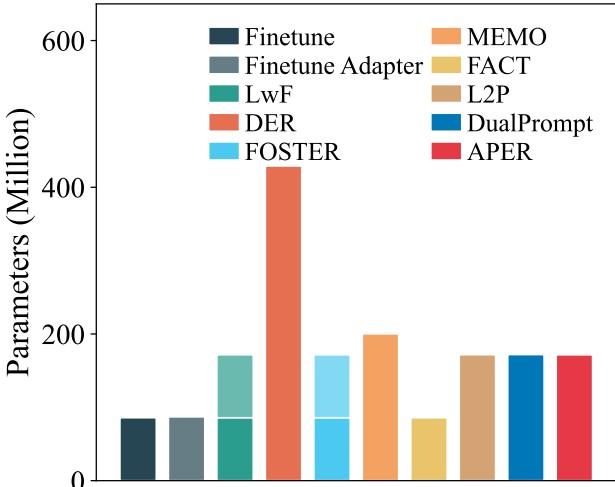

Figure 8: Number of total parameters of different compared methods. The bars with shadow denote the parameters used during training but dropped during inference. APER **obtains better performance than compared methods with the same scale or fewer parameters.**

denotes the ability of a continual learner to remember the old knowledge, while "plasticity" refers to the ability to learn new concepts. These concepts are similar to the concept of "generalizability and adaptivity" in this paper. However, there are some main differences that need to be highlighted among these concepts:

- "stability-plasticity dilemma" mainly refers to the problem of training from scratch, where the model needs to balance learning new concepts and remembering the old. These concepts are two ultimate goals of continual learning, which do not conflict with "generalizability and adaptivity" raised in this paper.

- "generalizability and adaptivity" in this paper is the new characteristic in the era of pre-trained models. Specifically, a randomly initialized model does not have such "generalizability," which cannot be directly applied to the downstream tasks. However, continual learners are born with "generalizability" if starting with a pre-trained model, and we observe a simple baseline (SimpleCIL) shows strong performance. Furthermore, we find "generalizability" is not enough for all downstream tasks, especially in the case that downstream task comes from a different distribution (as we see in the main paper). In this case, we need to enhance the pre-trained model's "adaptivity" by further tuning it with the downstream task. Finally, by aggregating the features extracted by the pre-trained and adapted models, we find a way to unify these characteristics in a single model.

In summary, the "generalizability and adaptivity" in this paper is a new characteristic in class-incremental learning with pre-trained models. We aim to unify these characteristics in CIL and propose our APER by aggregating the adapted and pre-trained models.

## A.5 SUMMARY OF CONTRIBUTIONS

Our main contributions can be summarized as follows: 1) We reveal that a simple baseline (Simple-CIL) outperforms even state-of-the-art methods when fairly compared using the same pre-trained model. It indicates that SimpleCIL should stand as a strong baseline in pre-trained model-based class-incremental learning in future works. 2) We point out two core factors when deploying pre-trained models in class-incremental learning, *i.e.*, generalizability and adaptivity. We also propose a simple yet effective method to unify these characteristics in a single model. 3) APER is a general framework that can be combined with various parameter-efficient tuning algorithms. It achieves state-of-the-art performance in extensive experiments on seven newly established benchmark datasets that have large domain gaps with the pre-trained dataset.

Table 3: Introduction about adopted model architectures in the main paper.

| Model | # params | Link |
|---|---|---|
| ViT-B/16-IN1K | 86M | Link |
| ViT-B/16-IN21K | 86M | Link |
| ResNet18 | 11M | Link |
| ResNet50 | 23M | Link |
| ResNet101 | 43M | Link |
| ResNet152 | 58M | Link |
| ViT-L/16-IN1K | 303M | Link |
| ViT-B/16-DINO | 86M | Link |
| ViT-B/16-SAM | 86M | Link |
| ViT-B/16-MAE | 86M | Link |
| ViT-B/16-CLIP | 86M | Link |

Table 4: Introduction about benchmark datasets. ObjectNet, OmniBenchmark, and VTAB contain massive classes, and we sample a subset from them to construct the incremental learning task.

| Dataset | # training instances | # testing instances | # Classes | Link |
|---|---|---|---|---|
| CIFAR100 | 50,000 | 10,000 | 100 | Link |
| CUB200 | 9,430 | 2,358 | 200 | Link |
| ImageNet-R | 24,000 | 6,000 | 200 | Link |
| ImageNet-A | 5,981 | 1,519 | 200 | Link |
| ObjectNet | 26,509 | 6,628 | 200 | Link |
| OmniBenchmark | 89,697 | 5,985 | 300 | Link |
| VTAB | 1,796 | 8,619 | 50 | Link |

## B  IMPLEMENTATION DETAILS

In this section, we discuss the detailed implementation in APER, including the pseudo code, introduction of compared methods, hyper-parameters, selection of pre-trained models, and discussion about datasets.

### B.1  PSEUDO CODE

We summarize the training protocol of APER in Algorithm 1. Given the pre-trained model, we first adapt it with the first training dataset via Eq. 5 to get the adapted model. Afterward, we freeze the pre-trained model and adapted model and merge the embeddings. For the subsequent tasks, we get a new dataset and replace the classifier weights with prototypical features (*i.e.*, class centers).

### B.2  COMPARED METHODS

We first introduce the compared methods in the main paper. These methods are as follows:

- **Finetune**: directly trains the model with new datasets incrementally, which leads to catastrophic forgetting;

- **Finetune Adapter (Chen et al., 2022a)**: freezes the pre-trained weights and sequentially optimizes the adapter module. To improve its performance, only the specific classifiers in the current dataset (*i.e.*, $\mathbf{w}_i, i \in Y_b$), are tuned, and the classifiers for former classes (*i.e.*, $\mathbf{w}_i, i \in \mathcal{Y}_{b-1}$) are frozen;

- **LwF (Li & Hoiem, 2017)**: utilizes knowledge distillation (Hinton et al., 2015) as a regularization term to overcome forgetting, which relies on the supervision of old model to produce soft targets;

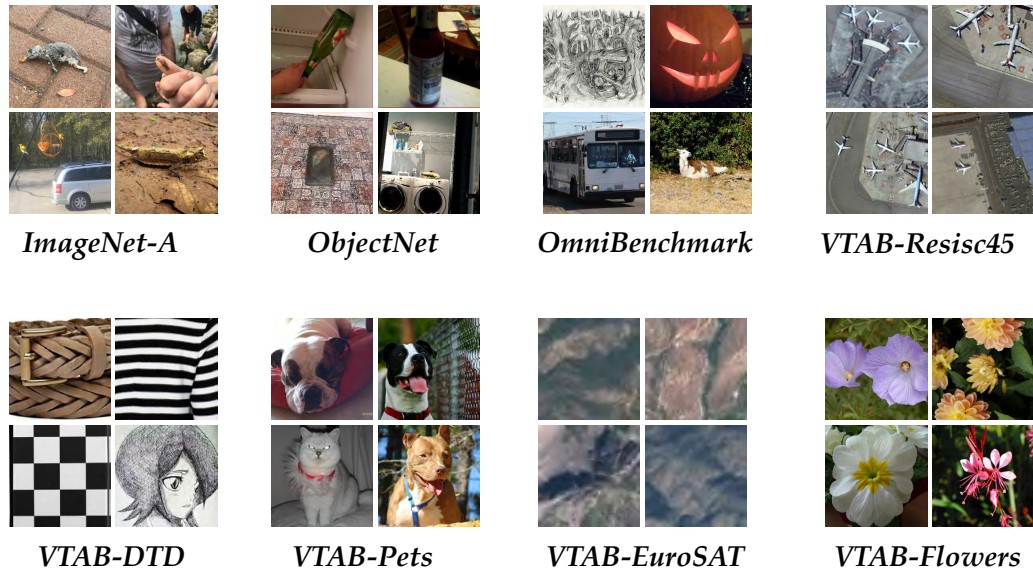

ImageNet-A       ObjectNet       OmniBenchmark       VTAB-Resisc45

VTAB-DTD       VTAB-Pets       VTAB-EuroSAT       VTAB-Flowers

Figure 9: Dataset visualizations of our new benchmarks. **ImageNet-A and ObjectNet contain hard samples that ImageNet pre-trained models cannot handle, while OmniBenchmark and VTAB contain classes from multiple domains.**

---

**Algorithm 1** AdaPt and mERge (APER) for CIL

---

**Input**: Incremental datasets: $\left\{ \mathcal{D}^1, \mathcal{D}^2, \cdots, \mathcal{D}^B \right\}$, Pre-trained Model: $f(\mathbf{x})$;
**Output**: Updated model;

1: Adapt the model to $\mathcal{D}^1$ via Eq. 5;                         ▷ Model adapt
2: Freeze the embedding functions $\phi^*(\cdot)$ and $\phi(\cdot)$;
3: Merge the embeddings, *i.e.*, $[\phi^*(\mathbf{x}), \phi(\mathbf{x})]$;             ▷ Model merge
4: **for** $b = 1, 2 \cdots, B$ **do**              ▷ Incremental learning
5:      Get the incremental training set $\mathcal{D}^b$;
6:      Extract the prototypes via Eq. 6;
7:      Replace the classifier weight with prototype;
    **return** the updated model;

---

- **SDC (Hou et al., 2019)**: combines feature distillation, metric learning, and normalized classifier for class-incremental learning. It builds the mapping between the features of the old model and the current model to resist forgetting;

- **DER (Yan et al., 2021)**: state-of-the-art class-incremental learning method, which creates a new backbone for each new incremental task. All these features are concatenated together to learn the unified classifier;

- **FOSTER (Wang et al., 2022a)**: state-of-the-art class-incremental learning method, which extends DER with a model compression stage to control the memory budget;

- **MEMO (Zhou et al., 2023b)**: state-of-the-art class-incremental learning method. It creates new residual layers instead of the whole backbone to reduce memory costs. In the implementation, we change the residual layers into transformer blocks to keep consistent with the ViT structure;

- **FACT (Zhou et al., 2022a)**: state-of-the-art class-incremental learning method, which also builds a prototype-based classifier. It reserves the embedding space for new classes in the incremental learning process;

- **L2P (Wang et al., 2022e)**: state-of-the-art PTM-based CIL method. It freezes the pre-trained model and optimizes a prompt pool to fit new patterns. To determine which prompt to use, it designs a 'key-value' pair for prompt matching and utilizes an extra pre-trained model to provide the embeddings for prompt retrieval;

- **DualPrompt (Wang et al., 2022d)**:state-of-the-art PTM-based class-incremental learning method. It extends L2P with two kinds of prompts, namely general and expert prompts. Similar to L2P, it also relies on another pre-trained model for prompt retrieval;

- **CODA-Prompt (Smith et al., 2023)**:state-of-the-art PTM-based class-incremental learning method. It replaces the hard prompt retrieval process in L2P via the attention mechanism. To enhance the diversity of prompts in the prompt pool, it designs an extra orthogonality constraint.

Note that all methods are based on the pre-trained ViT in the main paper. For methods requiring backbone expansion (*e.g.*, DER, MEMO, and FOSTER), we also use pre-trained ViT as the initialization of new backbones.

**Discussion about total parameters**: Since L2P and DualPrompt have prompt pools, they rely on another pre-trained ViT as the 'retriever' to search for the instance-specific prompt. Hence, APER shares the same scale of total parameters as these methods. We list the total number of parameters of these methods in Figure 8, which indicates that APER obtains better performance than the compared methods with the same scale or fewer parameters. Additionally, since APER utilizes parameter-efficient tuning techniques to obtain the adapted model, most of the parameters in the adapted model is the same as the pre-trained weight. Hence, the memory budget of APER can be further alleviated, which we will explore in future works.

## B.3  IMPLEMENTATIONS AND HYPER-PARAMETERS

For compared methods, we adopt the PyTorch implementation[2] of L2P[3] (Wang et al., 2022e) and DualPrompt[4] (Wang et al., 2022d). We follow the implementations in PyCIL[5] to re-implement other compared methods with ViT, *i.e.*, Finetune, Finetune Adapter, LwF, DER, FOSTER, MEMO, and FACT.

Specifically, for APER, the number of the prompt length $p$ in VPT is set to 5, and the projected dimension of the Adapter $r$ is set to 16. There are no other hyper-parameters for APER to set, which ensures the robustness of our proposed method. We conduct experiments to investigate the influence of these parameters in Section C.3. Following (Wang et al., 2022e), we use the same data augmentation for all methods, *i.e.*, random resized crop and horizontal flip. Input images are resized to 224×224 before feeding into the model. Following (Rebuffi et al., 2017b), all classes are randomly shuffled with Numpy random seed 1993 before splitting into incremental tasks. A specific case is VTAB, where we force the classes to emerge from domain to domain, as discussed in Section B.5 and D.7.

## B.4  PRE-TRAINED MODELS

Since there are various kinds of pre-trained models publicly available, we follow (Wang et al., 2022e;d) to choose the most commonly used ones in the main paper for evaluation, denoted as **ViT-B/16-IN1K** and **ViT-B/16-IN21K**. Specifically, both models are pre-trained on ImageNet21K (Deng et al., 2009), while ViT-B/16-IN1K is further finetuned with ImageNet1K. We follow timm (Wightman, 2019) implementation and report the details about these models in Table 3. For ResNet, we utilize the Pytorch (Paszke et al., 2019) pre-trained models. Additionally, we report the backbones adopted in the ablation study, including ViT-L/16-IN1K, ViT-B/16-DINO (Caron et al., 2021), ViT-B/16-SAM (Chen et al., 2022b), ViT-B/16-MAE (He et al., 2022b), ViT-B/16-CLIP (Radford et al., 2021) (image encoder), in the table. Among them, ViT-B/16-DINO and ViT-B/16-MAE are trained with self-supervised loss, and ViT-B/16-CLIP is trained on 400 million image-text pairs with contrastive loss.

---

[2]The original implementation `https://github.com/google-research/l2p` is based on JAX.
[3]`https://github.com/JH-LEE-KR/l2p-pytorch`
[4]`https://github.com/JH-LEE-KR/dualprompt-pytorch`
[5]`https://github.com/G-U-N/PyCIL`

## B.5 DATASETS

**Discussions about dataset selection:** In this section, we provide an introduction to the datasets used in the main paper. We list the details of seven adopted datasets in Table 4. Specifically, CIFAR100, CUB200, and ImageNet-R are benchmark CIL datasets widely adopted in (Rebuffi et al., 2017b; Wang et al., 2022d; Yu et al., 2020). However, due to the data overlap between ImageNet-based benchmarks and the pre-trained dataset, ImageNet is unsuitable for evaluating PTM-based CIL methods (Wang et al., 2022e). As a result, we introduce four new benchmarks for CIL that **1)** do not overlap with the ImageNet dataset, **2)** have a large domain gap with ImageNet, increasing the burden of PTM to generalize, and **3)** contain large-scale datasets from multiple realms that can form the cross-domain class-incremental benchmark. We list the detailed information below.

- **CIFAR100** (Krizhevsky et al., 2009) contains 100 classes with 60,000 images, of which 50,000 are training instances, and 10,000 are testing ones, with 100 images per class.
- **CUB200** (Wah et al., 2011) is a widely-used dataset for fine-grained visual categorization task. It contains 11,788 images of 200 subcategories belonging to birds, with 9,430 for training and 2,358 for testing.
- **ImageNet-R** (Hendrycks et al., 2021a) is introduced into CIL by (Wang et al., 2022d). It contains newly collected data of different styles, such as cartoons, graffiti, and origami, as well as hard examples from ImageNet that standard ImageNet pre-trained models fail to classify. Following (Wang et al., 2022d), there are 24,000 training instances and 6,000 testing instances from 200 classes.
- **ImageNet-A** (Hendrycks et al., 2021b) ImageNet-A is a dataset of real-world adversarially filtered images that fool current ImageNet pre-trained classifiers. It was exported from sites including iNaturalist, Flickr, and DuckDuckGo, and adversarially selected by removing examples that fail to fool ResNet50. We select 5,981 training instances and 1,519 testing instances from 200 classes.
- **ObjectNet** (Barbu et al., 2019) is a large, real-world dataset for object recognition with controlled variations in object backgrounds, rotations, and imaging viewpoints, making finetuning a challenge due to only small performance increases. When tested on ObjectNet, object detectors experience a 40~45% drop in performance compared to their performance on other benchmarks. The original ObjectNet contains classes from 313 classes, and we select a subset of 200 classes for class-incremental learning. Among them, 26,509 instances are for training, and 6,628 are for testing.
- **OmniBenchmark** (Zhang et al., 2022) is a concise and diverse benchmark for evaluating pre-trained model generalization across semantic super-concepts/realms. It contains 21 semantic realm-wise datasets that have no overlapping concepts. The original OmniBenchmark is constituted of 7,372 classes, from which we sample 300 categories to construct the class-incremental learning dataset. The subset contains 89,697 training instances and 5,985 testing instances. As these selected classes are from multiple *realms*, it is harder to conduct incremental learning with OmniBenchmark than other datasets due to the domain gap among different classes.
- **VTAB** (Zhai et al., 2019) includes 19 evaluation tasks spanning a variety of domains that can be grouped into three categories — *natural*, *specialized*, and *structured*. The Natural group includes images of the natural world captured through standard cameras, representing generic objects, fine-grained classes, or abstract concepts. The Specialized group utilizes images captured using specialist equipment, such as medical images or remote sensing. The Structured group derives from artificial environments that target understanding of specific changes between images, such as predicting the distance to an object in a 3D scene, counting objects, or detecting orientation. Since the original VTAB contains 19 datasets, we select 5 to construct a cross-domain class-incremental learning setting, *i.e.*, Resisc45 (Cheng et al., 2017), DTD (Cimpoi et al., 2014), Pets (Parkhi et al., 2012), EuroSAT (Helber et al., 2019), and Flowers (Nilsback & Zisserman, 2006). **In VTAB, we do not shuffle the classes and make the classes emerge from domain to domain, which is a more realistic incremental learning setting.**

We give the visualizations of our newly introduced datasets in Figure 9. As shown in the figure, ImageNet-A and ObjectNet contain hard samples that could be misclassified by the ImageNet pre-

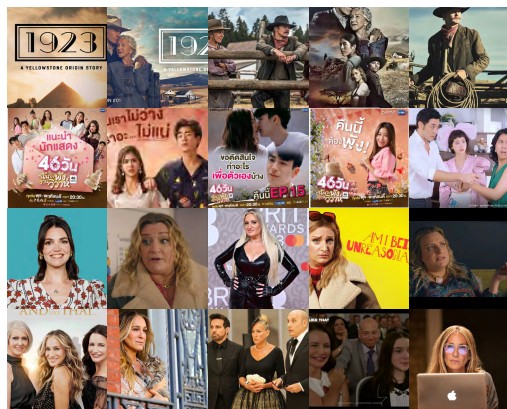

*1923*

*46 Days*

*Am I Being Unreasonable*

*And Just Like That*

Figure 10: The visualization of four classes in the TV series dataset. Each line corresponds to the same class. We collect these classes with TV series after the year 2021, and crawl images from Google and IMDB. Since these TV series are released after the collection of ImageNet, there is no data or class overlapping to the pre-training dataset.

trained model. In contrast, OmniBenchmark and VTAB contain cross-domain instances, which increase the difficulty of incremental learning. Since the distribution gap from domain to domain has not been explored in former CIL tasks, exploring the cross-task incremental learning problem with OmniBenchmark and VTAB is very challenging.

It must be noted that there are two main sources of the distribution shift, *i.e.*, class drift and distribution drift. The former denotes that the classes are totally different between the two sets. By directly matching the class names to ImageNet21K, we find only 17% classes are overlapped for ObjectNet and Omnibenchmark, while only 10% for VTAB. The overlapping rate of these newly established benchmarks is much lower than the typical benchmark CIFAR100 (56%). Hence, these datasets do have a large domain gap to ImageNet21K. Furthermore, although ImageNet-A and ImageNet-R share the same class space as ImageNet21K, the input distribution shifts substantially between them. As shown in Figure 9, these datasets contain hard samples or out-of-distribution instances that a pre-trained model cannot handle. In summary, the introduction of these new benchmarks enables a holistic comparison of different CIL methods in the era of PTM.

Apart from these new benchmarks, we also collect a new dataset with no class overlapping and data overlapping with ImageNet21K, and we report the collection process and experimental results in Section C.1.

## C   EXTRA EXPERIMENTAL EVALUATIONS

In this section, we conduct experiments to investigate the variations of APER. Specifically, we collect a new dataset without any class/data overlapping to ImageNet, and conduct experiments with it. We also compare the results of model adaptation with different incremental stages to determine the best solution in model adaptation. Besides, we also explore the influence of hyper-parameters in model adaptation, *e.g.*, the prompt length in VPT and the projection dimension in Adapter. Additionally, we provide additional Grad-CAM results for better visualization.

### C.1   EXPERIMENT ON NON-OVERLAPPING DATASET

In the main paper, we mainly focus on existing benchmark datasets in the research community that have large domain gaps to ImageNet, *e.g.*, CIFAR100 and CUB (typical benchmark for CIL evaluation), ImageNet-R (the benchmark for CIL with pre-trained models), ImageNet-A, OmniBenchmark, ObjectNet, and VTAB (benchmark datasets that have domain gap to ImageNet). However, one shall argue that even if these benchmarks contain out-of-distribution instances that the pre-trained model

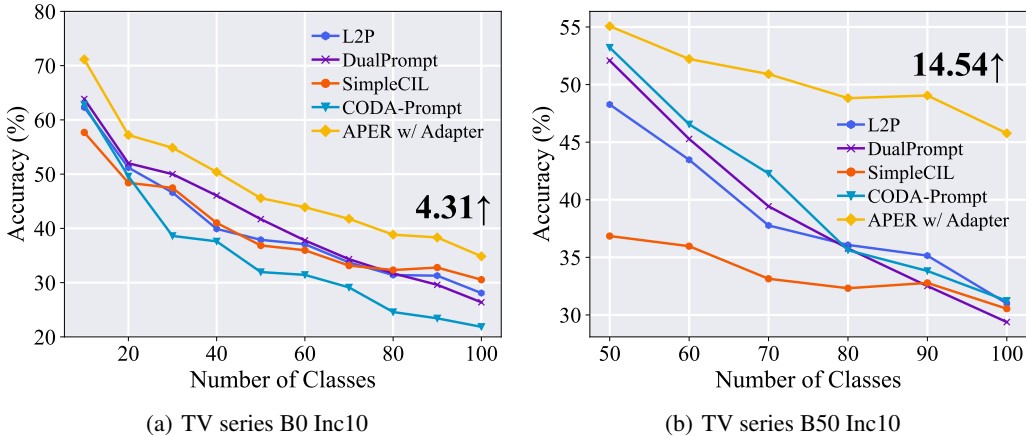

(a) TV series B0 Inc10          (b) TV series B50 Inc10

Figure 11: Experiments on TV series dataset. This is a strictly non-overlapping dataset to the pre-trained data, consisting of TV series images after 2021. APER still outperforms other competitors by a substantial margin on this dataset. We denote the improvement compared to the runner-up method at the end of the performance curve.

cannot recognize, the objects may still overlap with the pre-trained dataset. To tackle this problem, we manually collect a new dataset for evaluation, namely the TV series dataset.

Noticing that ImageNet is targeted for recognizing the common objects in the real world, we aim to collect a dataset that emerged recently, which strictly has no overlapping with ImageNet. TV series are emerging frequently in our daily lives, spreading all over the world. It inspires us that **"Can we collect a dataset for TV series classification?"** To achieve this goal, **we collect the TV series items from the internet and only keep those TV series after the year 2021**. Since the ImageNet dataset (Russakovsky et al., 2015) was collected before 2016, the images collected from these TV series will have no overlapping with the pre-trained dataset.

After collecting the TV series items, we manually collect images of these TV series from Google search and IMDB. These images contain the TV series poster, actor images, stage photos, etc. Hence, we can construct the classification task by "classifying the related images to the corresponding TV series." We further manually delete the repeated and irrelevant images from the raw image pool, and **randomly select 100 classes as the classification task**. Figure 10 shows a subset of this dataset. We will open-source this dataset upon acceptance.

After the data collection, we report the experimental results in Figure 11. Since there are 100 classes in total, we adopt two data splits, *i.e.*, Base 0 Inc 10 and Base 50 Inc10. We keep other settings the same as the main paper and compare APER w/ Adapter to L2P, DualPrompt, SimpleCIL, and CODA-Prompt. As we can infer from the figure, APER still outperforms other competitors by a substantial margin, *e.g.*, 4.31% in Figure 11(a) and 14.54% in Figure 11(b). It indicates that APER shows consistent improvement on various datasets (including the benchmark datasets in the main paper and this non-overlapping dataset), verifying APER a strong baseline in the era of class-incremental learning with pre-trained models.

## C.2 INFLUENCE OF ADAPTING STAGES

As discussed in the main paper, we only adapt the pre-trained model in the first incremental stage with $\mathcal{D}^1$ in APER. There are two reasons: 1) Sequentially tuning the model will suffer catastrophic forgetting. 2) Since we utilize a prototype-based classifier, tuning the model with multiple stages will result in incompatible features between former and new prototypes.

In this section, we conduct an ablation to determine the influence of adapting stages and report the results in Figure 12. We conduct the experiment on CIFAR100 Base0 Inc10 setting with pre-trained ViT-B/16-IN21K. There are 10 incremental stages in total. We denote the tuning stages as $T$ and train APER w/ Adapter for ablation. Specifically, we change the tuning stages among

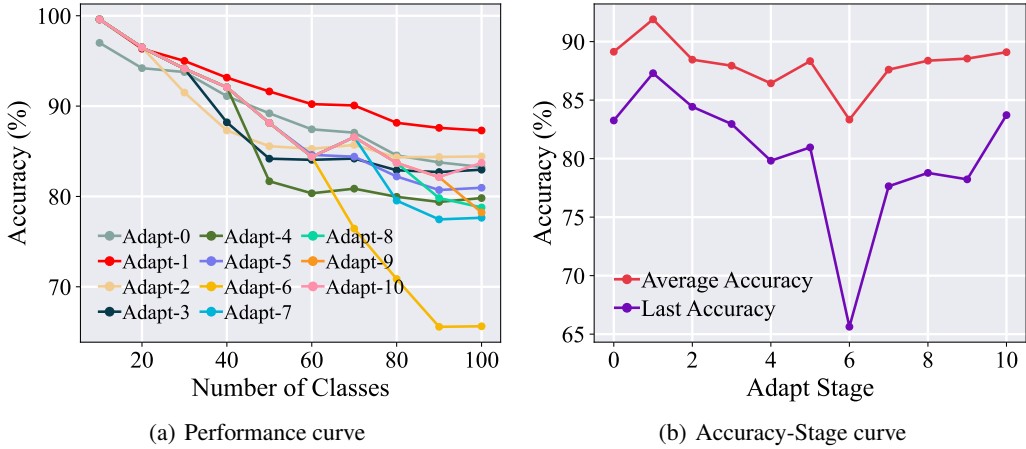

(a) Performance curve

(b) Accuracy-Stage curve

Figure 12: Influence of tuning stages. **Left**: the accuracy trend with the change of tuning stages. Adapt-$T$ denotes the model is adapted for the first $T$ incremental tasks. $T = 0$ denotes Simple-CIL. **Right**: the average/last accuracy with the change of tuning stages. APER **achieves the best performance when the model is only tuned with the first incremental task.**

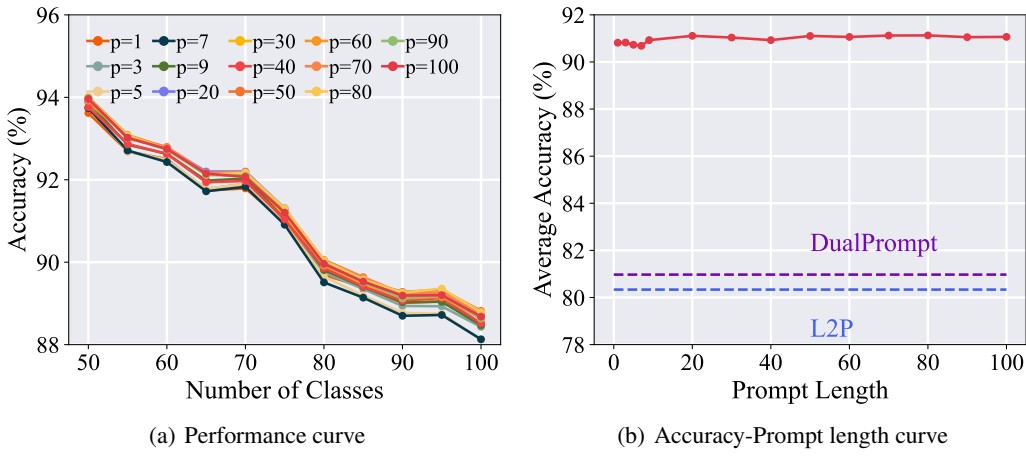

(a) Performance curve

(b) Accuracy-Prompt length curve

Figure 13: Influence of prompt length. **The performance of APER is robust with the change of the prompt length, and we set $p = 5$ as the default setting.**

$\{0, 1, 2, \cdots, 10\}$ to determine the influence on the final performance. In the first $T$ stages, we adapt the PTM incrementally with adapter and replace the classifier with prototypes. Afterward, in the $T$-th stage, we freeze the encoding functions and only extract prototypes for the following stages. $T = 0$ denotes vanilla SimpleCIL. To prevent forgetting, we freeze the classifier weight of former classes when learning new classes.

As shown in Figure 12(a), tuning the model with the first stage achieves the best performance among all settings. Specifically, multi-stage tuning harms generalizability and results in the incompatible features of former and new classes. We also plot the trend of average/last accuracy with the change of tuning stages in Figure 12(b), where $T = 1$ achieves the best performance in both measures.

## C.3 INFLUENCE OF HYPER-PARAMETERS

In this section, we explore the influence of hyper-parameters in APER. Specifically, since APER is optimized with parameter-efficient tuning, the only hyper-parameters come from these tuning

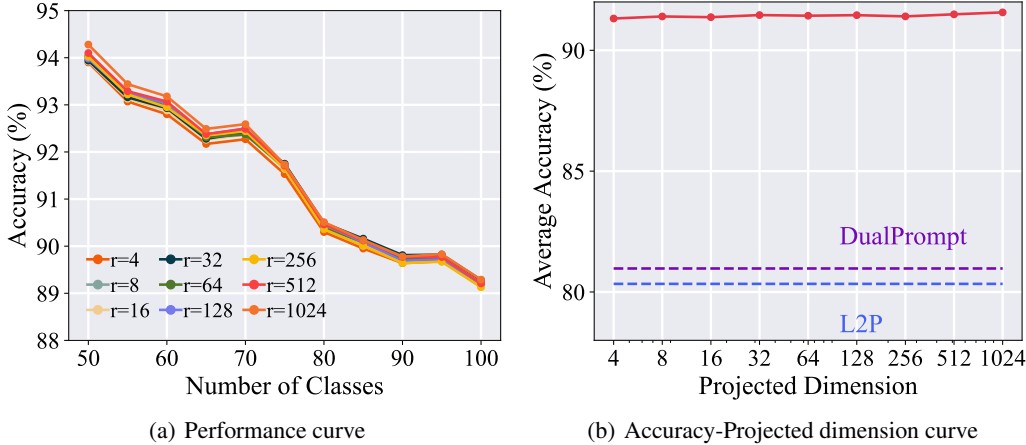

(a) Performance curve

(b) Accuracy-Projected dimension curve

Figure 14: Influence of projected dimension in adapter. **The performance of** APER **is robust with the change of the projected dimension, and we set** $r = 16$ **as the default setting.**

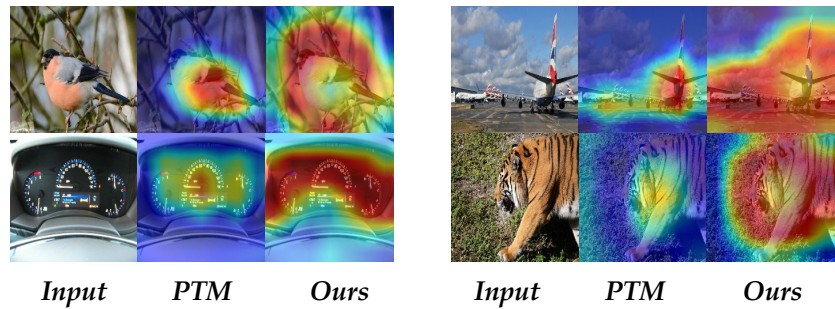

*Input*     *PTM*     *Ours*     *Input*     *PTM*     *Ours*

Figure 15: Grad-CAM results. **Compared to PTM,** APER **concentrates more on task-specific features.**

methods, *i.e.*, the prompt length $p$ in VPT and the projection dim $r$ in Adapter. We train the model on CIFAR100 B50 Inc5 with pre-trained ViT-B/16-IN1K in this section.

Firstly, we investigate the influence of the prompt length in Figure 13. The figure shows that APER's performance is robust with the change of the prompt length. Therefore, we use $p = 5$ as a default parameter for all settings. Similarly, we observe similar phenomena in Figure 14, where the performance is robust with the change of the projection dimension. As a result, we set $r = 16$ as a default parameter for all settings.

## C.4   GRAD-CAM RESULTS

Apart from the t-SNE visualizations in the main paper, we also visualize the Grad-CAM (Selvaraju et al., 2017) results on OmniBenchmark dataset based on pre-trained ResNet18. Grad-CAM is utilized to highlight the critical regions in the image for predicting the corresponding concept. The results are shown in Figure 15, indicating APER concentrates more on task-specific features than vanilla PTM, confirming the effectiveness of model adaptation in capturing task-specific features.

## C.5   INFLUENCE OF COSINE CLASSIFIER

In this section, we verify the necessity of using a cosine classifier in APER. Specifically, we report the performance of SimpleCIL and APER and their variations without the cosine classifier in Table 5. As we can infer from the table, replacing the cosine classifier with an unnormalized classifier will lead

Table 5: Ablation study on cosine classifier (CC) with ViT-B/16-IN1K. **Cosine classifier is essential for building PTM-based CIL models with prototypical classifiers.**

| Method | ObjectNet B100 Inc5 | | ImageNet-A B0 Inc40 | |
|---|---|---|---|---|
| | $\bar{\mathcal{A}}$ | $\mathcal{A}_B$ | $\bar{\mathcal{A}}$ | $\mathcal{A}_B$ |
| SimpleCIL w/o CC | 51.83 | 45.07 | 55.03 | 43.72 |
| SimpleCIL w/ CC | **56.02** | **51.13** | **58.53** | **49.44** |
| APER w/ SSF w/o CC | 62.17 | 54.76 | 62.75 | 50.91 |
| APER w/ SSF w/ CC | **66.31** | **60.64** | **65.74** | **56.09** |

to a performance gap of approximately 5%. Therefore, the cosine classifier is essential in building PTM-based models with prototypical classifiers.

It must be noted that we do not claim the SimpleCIL baseline to be with novelty since it is a direct and straightforward idea to apply a pre-trained model in class-incremental learning. However, our contribution lies in the observation that SimpleCIL sometimes beats current state-of-the-art, which points out the possible direction of future designs in CIL.

| Backbone | Method | $\bar{\mathcal{A}}$ | $\mathcal{A}_B$ |
|---|---|---|---|
| | L2P (Wang et al., 2022e) | 85.44 | 79.03 |
| | DualPrompt (Wang et al., 2022d) | 83.73 | 76.23 |
| ViT-B/16-IN1K | SimpleCIL | 82.79 | 76.21 |
| | APER w/ Finetune | 83.10 | 76.14 |
| | APER w/ VPT-Shallow | **88.73** | 82.90 |
| | APER w/ VPT-Deep | 87.07 | 80.38 |
| | APER w/ SSF | 85.76 | 78.97 |
| | APER w/ Adapter | 88.61 | **82.98** |
| | L2P (Wang et al., 2022e) | 85.94 | 79.93 |
| | DualPrompt (Wang et al., 2022d) | 87.87 | 81.15 |
| ViT-B/16-IN21K | SimpleCIL | 87.57 | 81.26 |
| | APER w/ Finetune | 87.67 | 81.27 |
| | APER w/ VPT-Shallow | 90.43 | 84.57 |
| | APER w/ VPT-Deep | 88.46 | 82.17 |
| | APER w/ SSF | 87.78 | 81.98 |
| | APER w/ Adapter | **90.65** | **85.15** |

Table 6: Results on CIFAR100 Base0 Inc5 setting.

| Backbone | Method | $\bar{\mathcal{A}}$ | $\mathcal{A}_B$ |
|---|---|---|---|
| | L2P (Wang et al., 2022e) | 88.88 | 83.36 |
| | DualPrompt (Wang et al., 2022d) | 87.88 | 81.27 |
| ViT-B/16-IN1K | SimpleCIL | 82.31 | 76.21 |
| | APER w/ Finetune | 82.87 | 76.09 |
| | APER w/ VPT-Shallow | 89.04 | 83.77 |
| | APER w/ VPT-Deep | 89.14 | 83.26 |
| | APER w/ SSF | 89.75 | 83.91 |
| | APER w/ Adapter | **90.94** | **85.75** |
| | L2P (Wang et al., 2022e) | 88.34 | 84.57 |
| | DualPrompt (Wang et al., 2022d) | 89.69 | 84.14 |
| ViT-B/16-IN21K | SimpleCIL | 87.13 | 81.26 |
| | APER w/ Finetune | 87.12 | 81.23 |
| | APER w/ VPT-Shallow | 90.25 | 85.04 |
| | APER w/ VPT-Deep | 90.40 | 84.62 |
| | APER w/ SSF | 90.61 | 85.14 |
| | APER w/ Adapter | **92.24** | **87.49** |

Table 7: Results on CIFAR100 Base0 Inc10 setting.

| Backbone | Method | $\bar{\mathcal{A}}$ | $\mathcal{A}_B$ |
|---|---|---|---|
| | L2P (Wang et al., 2022e) | 90.43 | 86.15 |
| | DualPrompt (Wang et al., 2022d) | 89.97 | 85.17 |
| ViT-B/16-IN1K | SimpleCIL | 81.12 | 76.21 |
| | APER w/ Finetune | 84.57 | 77.92 |
| | APER w/ VPT-Shallow | 87.86 | 83.32 |
| | APER w/ VPT-Deep | 89.85 | 85.08 |
| | APER w/ SSF | 90.40 | 85.44 |
| | APER w/ Adapter | **91.52** | **87.52** |
| | L2P (Wang et al., 2022e) | 90.68 | 86.99 |
| | DualPrompt (Wang et al., 2022d) | 91.21 | 86.85 |
| ViT-B/16-IN21K | SimpleCIL | 86.11 | 81.26 |
| | APER w/ Finetune | 86.06 | 81.29 |
| | APER w/ VPT-Shallow | 87.80 | 83.17 |
| | APER w/ VPT-Deep | 91.57 | 87.06 |
| | APER w/ SSF | 91.40 | 86.79 |
| | APER w/ Adapter | **92.41** | **88.48** |

Table 8: Results on CIFAR100 Base0 Inc20 setting.

| Backbone | Method | $\bar{\mathcal{A}}$ | $\mathcal{A}_B$ |
|---|---|---|---|
| | L2P (Wang et al., 2022e) | 80.34 | 69.78 |
| | DualPrompt (Wang et al., 2022d) | 80.97 | 70.11 |
| ViT-B/16-IN1K | SimpleCIL | 78.54 | 76.21 |
| | APER w/ Finetune | 85.98 | 80.15 |
| | APER w/ VPT-Shallow | 85.82 | 83.45 |
| | APER w/ VPT-Deep | 88.73 | 86.12 |
| | APER w/ SSF | 88.87 | 85.87 |
| | APER w/ Adapter | **91.43** | **89.20** |
| | L2P (Wang et al., 2022e) | 81.27 | 70.96 |
| | DualPrompt (Wang et al., 2022d) | 82.12 | 72.04 |
| ViT-B/16-IN21K | SimpleCIL | 83.78 | 81.26 |
| | APER w/ Finetune | 83.18 | 75.57 |
| | APER w/ VPT-Shallow | 86.25 | 84.13 |
| | APER w/ VPT-Deep | 91.04 | 88.71 |
| | APER w/ SSF | 89.91 | 87.17 |
| | APER w/ Adapter | **91.89** | **89.67** |

Table 9: Results on CIFAR100 Base50 Inc5 setting.

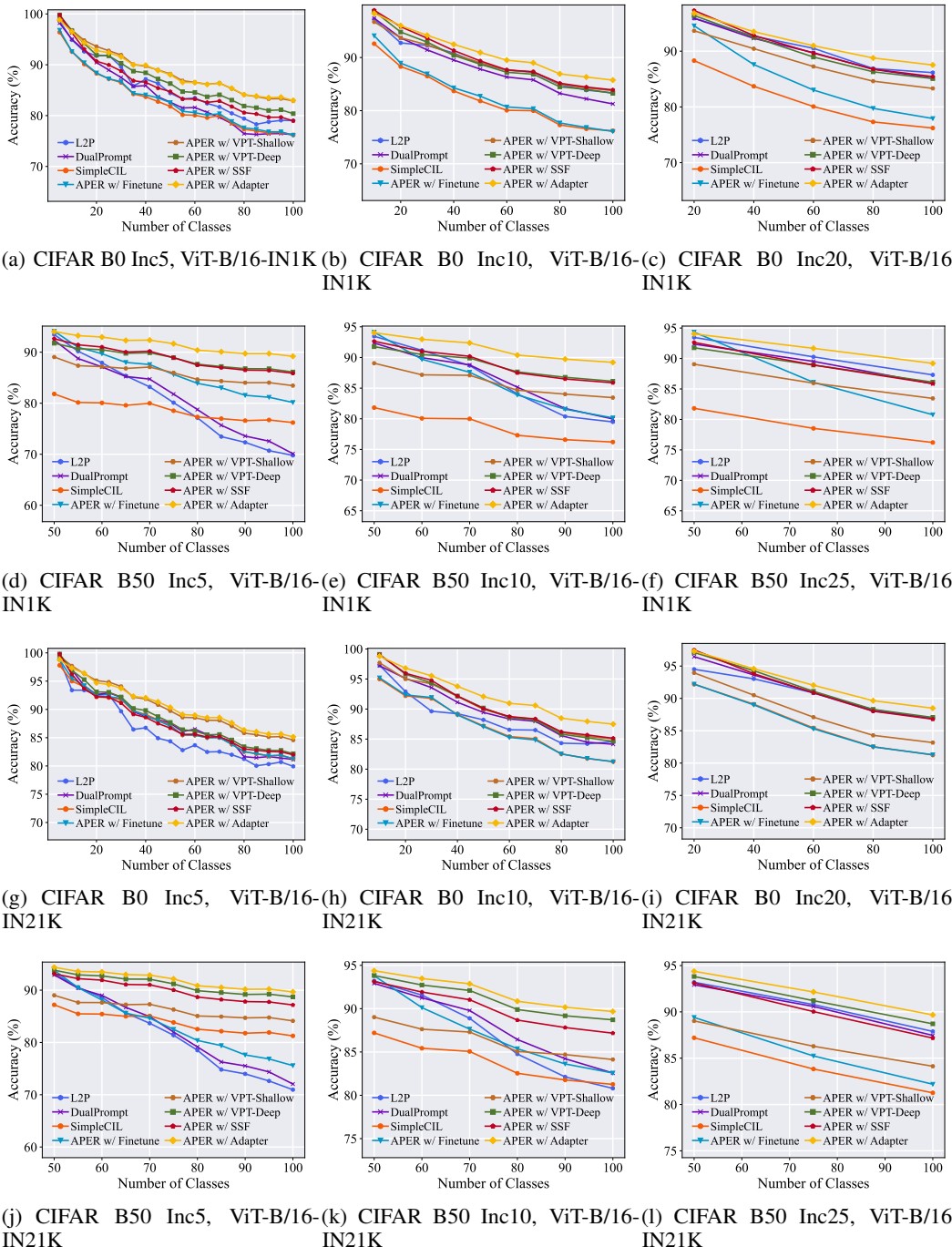

(a) CIFAR B0 Inc5, ViT-B/16-IN1K (b) CIFAR B0 Inc10, ViT-B/16-IN1K (c) CIFAR B0 Inc20, ViT-B/16-IN1K

(d) CIFAR B50 Inc5, ViT-B/16-IN1K (e) CIFAR B50 Inc10, ViT-B/16-IN1K (f) CIFAR B50 Inc25, ViT-B/16-IN1K

(g) CIFAR B0 Inc5, ViT-B/16-IN21K (h) CIFAR B0 Inc10, ViT-B/16-IN21K (i) CIFAR B0 Inc20, ViT-B/16-IN21K

(j) CIFAR B50 Inc5, ViT-B/16-IN21K (k) CIFAR B50 Inc10, ViT-B/16-IN21K (l) CIFAR B50 Inc25, ViT-B/16-IN21K

Figure 16: Experimental results on CIFAR100. **(a)∼(f):** Incremental performance comparison with **ViT-B/16-IN1K** as backbone. **(g)∼(l):** Incremental performance when using **ViT-B/16-IN21K** as backbone.

# D    FULL EXPERIMENTAL RESULTS ON 7 DATASETS

In this section, we report the full experimental performance on seven benchmark datasets with the numerical results and accuracy curves. We also report the results of residual networks with APER. Specifically, there are 45 dataset splits from seven datasets, and we report the performance of two backbones (ViT-B/16-IN1K and ViT-B/16-IN21K) on them.

| Backbone | Method | $\bar{\mathcal{A}}$ | $\mathcal{A}_B$ |
|---|---|---|---|
| | L2P (Wang et al., 2022e) | 86.21 | 79.49 |
| | DualPrompt (Wang et al., 2022d) | 86.32 | 79.97 |
| | SimpleCIL | 78.66 | 76.21 |
| ViT-B/16-IN1K | APER w/ Finetune | 86.17 | 80.15 |
| | APER w/ VPT-Shallow | 85.91 | 83.45 |
| | APER w/ VPT-Deep | 88.78 | 86.12 |
| | APER w/ SSF | 88.95 | 85.87 |
| | APER w/ Adapter | **91.45** | **89.20** |
| | L2P (Wang et al., 2022e) | 86.89 | 80.80 |
| | DualPrompt (Wang et al., 2022d) | 87.86 | 82.57 |
| | SimpleCIL | 83.88 | 81.26 |
| ViT-B/16-IN21K | APER w/ Finetune | 87.19 | 82.57 |
| | APER w/ VPT-Shallow | 86.31 | 84.13 |
| | APER w/ VPT-Deep | 91.07 | 88.71 |
| | APER w/ SSF | 89.96 | 87.17 |
| | APER w/ Adapter | **91.90** | **89.67** |

Table 10: Results on CIFAR100 Base50 Inc10 setting.

| Backbone | Method | $\bar{\mathcal{A}}$ | $\mathcal{A}_B$ |
|---|---|---|---|
| | L2P (Wang et al., 2022e) | 90.35 | 87.31 |
| | DualPrompt (Wang et al., 2022d) | 89.25 | 85.85 |
| | SimpleCIL | 78.85 | 76.21 |
| ViT-B/16-IN1K | APER w/ Finetune | 87.06 | 80.77 |
| | APER w/ VPT-Shallow | 86.16 | 83.45 |
| | APER w/ VPT-Deep | 88.94 | 86.12 |
| | APER w/ SSF | 89.15 | 85.87 |
| | APER w/ Adapter | **91.64** | **89.20** |
| | L2P (Wang et al., 2022e) | 90.63 | 87.89 |
| | DualPrompt (Wang et al., 2022d) | 90.32 | 87.46 |
| | SimpleCIL | 84.10 | 81.26 |
| ViT-B/16-IN21K | APER w/ Finetune | 85.61 | 82.19 |
| | APER w/ VPT-Shallow | 86.48 | 84.13 |
| | APER w/ VPT-Deep | 91.25 | 88.71 |
| | APER w/ SSF | 90.11 | 87.17 |
| | APER w/ Adapter | **92.07** | **89.67** |

Table 11: Results on CIFAR100 Base50 Inc25 setting.

## D.1    CIFAR100 RESULTS

For CIFAR100, we design 6 dataset splits to divide these 100 classes, namely **Base0 Inc5, Base0 Inc10, Base0 Inc20, Base 50 Inc5, Base50 Inc10, Base50 Inc25**. We report the results in Table 6, 7, 8, 9, 10, 11, and plot the corresponding incremental performance in Figure 16.

## D.2    CUB200 RESULTS

For CIFAR100, we design 8 dataset splits to divide these 200 classes, namely **Base0 Inc5, Base0 Inc10, Base0 Inc20, Base0 Inc40, Base 100 Inc5, Base100 Inc10, Base100 Inc20, Base100 Inc50.** We report the results in Table 12, 13, 14, 15, 16, 17, 18, 19 and plot the corresponding incremental performance in Figure 17.

| Backbone | Method | $\bar{\mathcal{A}}$ | $\mathcal{A}_B$ |
|---|---|---|---|
| | L2P (Wang et al., 2022e) | 64.08 | 49.11 |
| | DualPrompt (Wang et al., 2022d) | 68.20 | 54.32 |
| | SimpleCIL | 91.22 | 85.16 |
| ViT-B/16-IN1K | APER w/ Finetune | **91.52** | 85.67 |
| | APER w/ VPT-Shallow | 91.11 | 85.33 |
| | APER w/ VPT-Deep | 90.19 | 84.05 |
| | APER w/ SSF | 91.24 | **85.88** |
| | APER w/ Adapter | 91.18 | 85.20 |
| | L2P (Wang et al., 2022e) | 66.53 | 52.74 |
| | DualPrompt (Wang et al., 2022d) | 75.02 | 62.42 |
| | SimpleCIL | 92.46 | 86.73 |
| ViT-B/16-IN21K | APER w/ Finetune | **92.57** | **86.94** |
| | APER w/ VPT-Shallow | 92.29 | 86.60 |
| | APER w/ VPT-Deep | 91.37 | 85.96 |
| | APER w/ SSF | 92.17 | 86.73 |
| | APER w/ Adapter | 92.46 | 86.73 |

Table 12: Results on CUB200 Base0 Inc5 setting.

| Backbone | Method | $\bar{\mathcal{A}}$ | $\mathcal{A}_B$ |
|---|---|---|---|
| | L2P (Wang et al., 2022e) | 66.69 | 56.01 |
| | DualPrompt (Wang et al., 2022d) | 74.84 | 60.84 |
| | SimpleCIL | 90.96 | 85.16 |
| ViT-B/16-IN1K | APER w/ Finetune | **90.98** | **85.58** |
| | APER w/ VPT-Shallow | 90.70 | 85.54 |
| | APER w/ VPT-Deep | 89.48 | 83.42 |
| | APER w/ SSF | 90.67 | 85.37 |
| | APER w/ Adapter | 90.96 | 85.11 |
| | L2P (Wang et al., 2022e) | 67.05 | 56.25 |
| | DualPrompt (Wang et al., 2022d) | 77.47 | 66.54 |
| | SimpleCIL | 92.20 | **86.73** |
| ViT-B/16-IN21K | APER w/ Finetune | 91.82 | 86.39 |
| | APER w/ VPT-Shallow | 92.02 | 86.51 |
| | APER w/ VPT-Deep | 91.02 | 84.99 |
| | APER w/ SSF | 91.72 | 86.13 |
| | APER w/ Adapter | **92.21** | **86.73** |

Table 13: Results on CUB200 Base0 Inc10 setting.

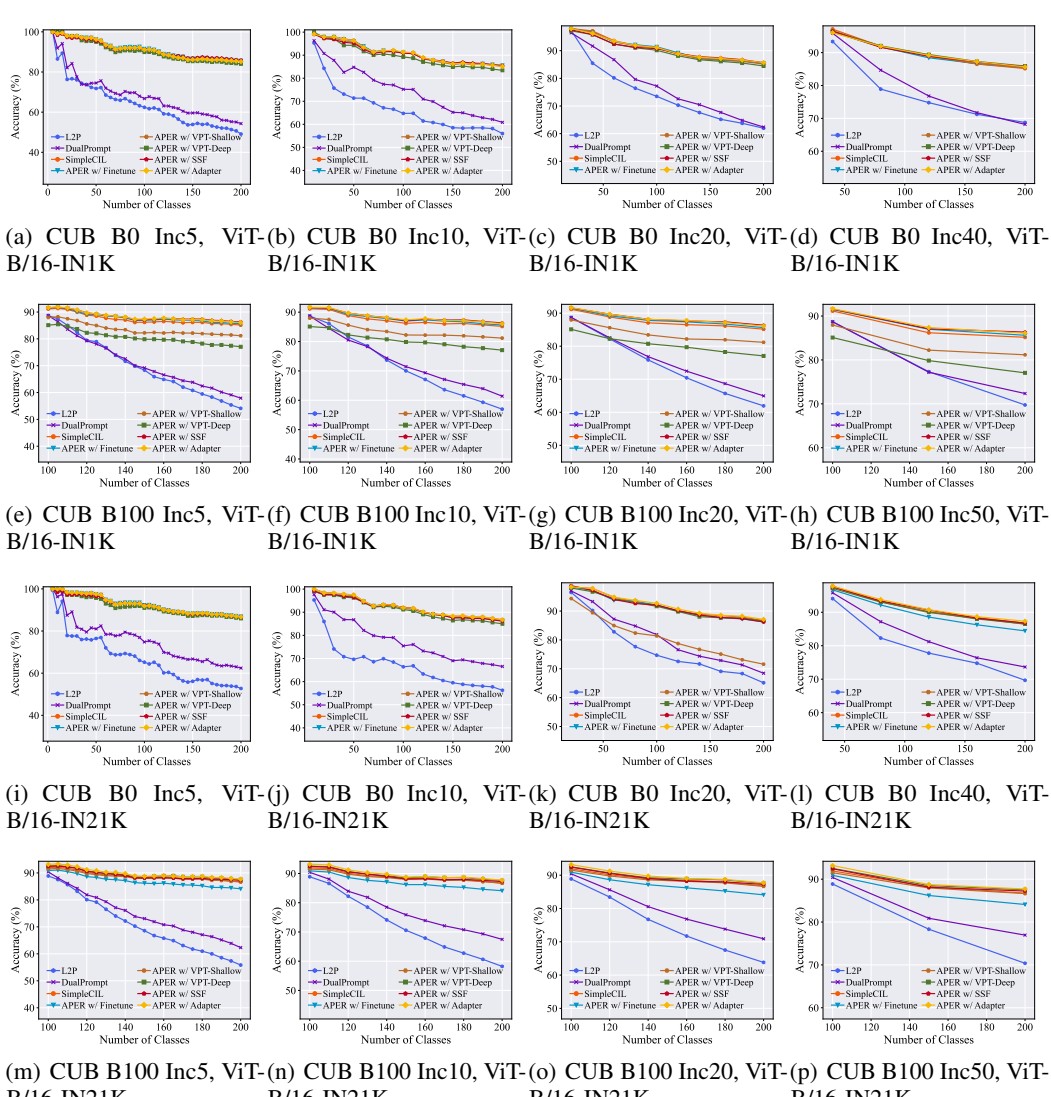

Figure 17: Experimental results on CUB. **(a)~(h):** Incremental performance comparison with **ViT-B/16-IN1K** as backbone. **(i)~(p):** Incremental performance when using **ViT-B/16-IN21K** as backbone.

| Backbone | Method | $\bar{\mathcal{A}}$ | $\mathcal{A}_B$ |
|---|---|---|---|
| | L2P (Wang et al., 2022e) | 74.17 | 61.95 |
| | DualPrompt (Wang et al., 2022d) | 76.96 | 62.38 |
| ViT-B/16-IN1K | SimpleCIL | 90.56 | 85.16 |
| | APER w/ Finetune | 90.63 | 85.41 |
| | APER w/ VPT-Shallow | 90.58 | 85.62 |
| | APER w/ VPT-Deep | 89.76 | 84.48 |
| | APER w/ SSF | 90.43 | 85.62 |
| | APER w/ Adapter | **90.68** | **85.67** |
| | L2P (Wang et al., 2022e) | 76.86 | 65.15 |
| | DualPrompt (Wang et al., 2022d) | 80.77 | 68.45 |
| ViT-B/16-IN21K | SimpleCIL | 91.82 | 86.73 |
| | APER w/ Finetune | 91.45 | 86.26 |
| | APER w/ VPT-Shallow | 80.78 | 71.59 |
| | APER w/ VPT-Deep | 91.33 | 86.51 |
| | APER w/ SSF | 91.47 | 86.22 |
| | APER w/ Adapter | **92.11** | **87.11** |

Table 14: Results on CUB200 Base0 Inc20 setting.

| Backbone | Method | $\bar{\mathcal{A}}$ | $\mathcal{A}_B$ |
|---|---|---|---|
| | L2P (Wang et al., 2022e) | 77.41 | 68.76 |
| | DualPrompt (Wang et al., 2022d) | 79.46 | 68.18 |
| ViT-B/16-IN1K | SimpleCIL | 89.92 | 85.16 |
| | APER w/ Finetune | 89.71 | 85.45 |
| | APER w/ VPT-Shallow | 89.88 | 85.41 |
| | APER w/ VPT-Deep | **90.24** | **85.92** |
| | APER w/ SSF | 89.97 | 85.75 |
| | APER w/ Adapter | 90.02 | 85.67 |
| | L2P (Wang et al., 2022e) | 79.72 | 69.68 |
| | DualPrompt (Wang et al., 2022d) | 82.85 | 73.65 |
| ViT-B/16-IN21K | SimpleCIL | 91.26 | 86.73 |
| | APER w/ Finetune | 89.62 | 84.44 |
| | APER w/ VPT-Shallow | 91.05 | 86.43 |
| | APER w/ VPT-Deep | 90.96 | 86.51 |
| | APER w/ SSF | 91.33 | 86.77 |
| | APER w/ Adapter | **91.67** | **87.32** |

Table 15: Results on CUB200 Base0 Inc40 setting.

| Backbone | Method | $\bar{\mathcal{A}}$ | $\mathcal{A}_B$ |
|---|---|---|---|
| ViT-B/16-IN1K | L2P (Wang et al., 2022e) | 69.73 | 54.11 |
| | DualPrompt (Wang et al., 2022d) | 70.91 | 57.90 |
| | SimpleCIL | 87.36 | 85.16 |
| | APER w/ Finetune | 87.96 | 85.62 |
| | APER w/ VPT-Shallow | 83.60 | 81.17 |
| | APER w/ VPT-Deep | 80.54 | 77.06 |
| | APER w/ SSF | **88.34** | **86.34** |
| | APER w/ Adapter | 88.33 | 86.09 |
| ViT-B/16-IN21K | L2P (Wang et al., 2022e) | 70.53 | 55.85 |
| | DualPrompt (Wang et al., 2022d) | 74.57 | 62.32 |
| | SimpleCIL | 88.93 | 86.73 |
| | APER w/ Finetune | 86.96 | 84.10 |
| | APER w/ VPT-Shallow | 88.66 | 86.64 |
| | APER w/ VPT-Deep | 89.50 | 87.53 |
| | APER w/ SSF | 89.15 | 87.23 |
| | APER w/ Adapter | **89.84** | **87.70** |

Table 16: Results on CUB200 Base100 Inc5 setting.

| Backbone | Method | $\bar{\mathcal{A}}$ | $\mathcal{A}_B$ |
|---|---|---|---|
| ViT-B/16-IN1K | L2P (Wang et al., 2022e) | 71.55 | 56.97 |
| | DualPrompt (Wang et al., 2022d) | 73.18 | 61.40 |
| | SimpleCIL | 87.40 | 85.16 |
| | APER w/ Finetune | 88.00 | 85.62 |
| | APER w/ VPT-Shallow | 83.62 | 81.17 |
| | APER w/ VPT-Deep | 80.52 | 77.06 |
| | APER w/ SSF | **88.37** | **86.34** |
| | APER w/ Adapter | 88.37 | 86.09 |
| ViT-B/16-IN21K | L2P (Wang et al., 2022e) | 72.30 | 58.23 |
| | DualPrompt (Wang et al., 2022d) | 77.47 | 67.46 |
| | SimpleCIL | 88.94 | 86.73 |
| | APER w/ Finetune | 86.98 | 84.10 |
| | APER w/ VPT-Shallow | 88.67 | 86.64 |
| | APER w/ VPT-Deep | 89.51 | 87.53 |
| | APER w/ SSF | 89.18 | 87.23 |
| | APER w/ Adapter | **89.86** | **87.70** |

Table 17: Results on CUB200 Base100 Inc10 setting.

| Backbone | Method | $\bar{\mathcal{A}}$ | $\mathcal{A}_B$ |
|---|---|---|---|
| ViT-B/16-IN1K | L2P (Wang et al., 2022e) | 74.14 | 61.94 |
| | DualPrompt (Wang et al., 2022d) | 75.69 | 64.97 |
| | SimpleCIL | 87.48 | 85.16 |
| | APER w/ Finetune | 88.03 | 85.62 |
| | APER w/ VPT-Shallow | 83.71 | 81.17 |
| | APER w/ VPT-Deep | 80.52 | 77.06 |
| | APER w/ SSF | **88.45** | **86.34** |
| | APER w/ Adapter | 88.42 | 86.09 |
| ViT-B/16-IN21K | L2P (Wang et al., 2022e) | 75.35 | 63.81 |
| | DualPrompt (Wang et al., 2022d) | 79.68 | 70.91 |
| | SimpleCIL | 89.00 | 86.73 |
| | APER w/ Finetune | 87.04 | 84.10 |
| | APER w/ VPT-Shallow | 88.72 | 86.64 |
| | APER w/ VPT-Deep | 89.57 | 87.53 |
| | APER w/ SSF | 89.23 | 87.23 |
| | APER w/ Adapter | **89.93** | **87.70** |

Table 18: Results on CUB200 Base100 Inc20 setting.

| Backbone | Method | $\bar{\mathcal{A}}$ | $\mathcal{A}_B$ |
|---|---|---|---|
| ViT-B/16-IN1K | L2P (Wang et al., 2022e) | 78.58 | 69.74 |
| | DualPrompt (Wang et al., 2022d) | 79.42 | 72.35 |
| | SimpleCIL | 87.50 | 85.16 |
| | APER w/ Finetune | 88.03 | 85.62 |
| | APER w/ VPT-Shallow | 83.79 | 81.17 |
| | APER w/ VPT-Deep | 80.67 | 77.06 |
| | APER w/ SSF | 88.34 | **86.34** |
| | APER w/ Adapter | **88.38** | 86.09 |
| ViT-B/16-IN21K | L2P (Wang et al., 2022e) | 79.19 | 70.38 |
| | DualPrompt (Wang et al., 2022d) | 82.74 | 76.95 |
| | SimpleCIL | 88.96 | 86.73 |
| | APER w/ Finetune | 87.07 | 84.10 |
| | APER w/ VPT-Shallow | 88.70 | 86.64 |
| | APER w/ VPT-Deep | 89.52 | 87.53 |
| | APER w/ SSF | 89.26 | 87.23 |
| | APER w/ Adapter | **89.87** | **87.70** |

Table 19: Results on CUB200 Base100 Inc50 setting.

## D.3 IMAGENET-R RESULTS

For ImageNet-R, we design 8 dataset splits to divide these 200 classes, namely **Base0 Inc5, Base0 Inc10, Base0 Inc20, Base0 Inc40, Base 100 Inc5, Base100 Inc10, Base100 Inc20, Base100 Inc50.** We report the results in Table 20, 21, 22, 23, 24, 25, 26, 27 and plot the corresponding incremental performance in Figure 18.

| Backbone | Method | $\bar{\mathcal{A}}$ | $\mathcal{A}_B$ |
|---|---|---|---|
| ViT-B/16-IN1K | L2P (Wang et al., 2022e) | 69.97 | 62.77 |
| | DualPrompt (Wang et al., 2022d) | 69.25 | 61.64 |
| | SimpleCIL | 68.04 | 61.28 |
| | APER w/ Finetune | 72.62 | 65.25 |
| | APER w/ VPT-Shallow | 71.85 | 64.53 |
| | APER w/ VPT-Deep | 73.17 | 65.55 |
| | APER w/ SSF | 72.16 | 64.93 |
| | APER w/ Adapter | **74.43** | **67.32** |
| ViT-B/16-IN21K | L2P (Wang et al., 2022e) | 66.53 | 59.22 |
| | DualPrompt (Wang et al., 2022d) | 63.31 | 55.22 |
| | SimpleCIL | 62.58 | 54.55 |
| | APER w/ Finetune | 70.51 | 62.42 |
| | APER w/ VPT-Shallow | 66.63 | 58.32 |
| | APER w/ VPT-Deep | 68.79 | 60.48 |
| | APER w/ SSF | 68.94 | 60.60 |
| | APER w/ Adapter | **72.35** | **64.33** |

Table 20: Results on ImageNet-R Base0 Inc5 setting.

| Backbone | Method | $\bar{\mathcal{A}}$ | $\mathcal{A}_B$ |
|---|---|---|---|
| ViT-B/16-IN1K | L2P (Wang et al., 2022e) | 75.28 | 69.33 |
| | DualPrompt (Wang et al., 2022d) | 74.48 | 68.64 |
| | SimpleCIL | 67.58 | 61.28 |
| | APER w/ Finetune | 73.34 | 65.77 |
| | APER w/ VPT-Shallow | 72.09 | 65.47 |
| | APER w/ VPT-Deep | 77.01 | 70.07 |
| | APER w/ SSF | 75.54 | 68.52 |
| | APER w/ Adapter | **77.29** | **70.47** |
| ViT-B/16-IN21K | L2P (Wang et al., 2022e) | 73.82 | 67.13 |
| | DualPrompt (Wang et al., 2022d) | 70.32 | 64.80 |
| | SimpleCIL | 61.99 | 54.55 |
| | APER w/ Finetune | 71.29 | 63.35 |
| | APER w/ VPT-Shallow | 70.19 | 62.75 |
| | APER w/ VPT-Deep | 74.46 | 66.47 |
| | APER w/ SSF | 73.07 | 65.00 |
| | APER w/ Adapter | **75.08** | **67.20** |

Table 21: Results on ImageNet-R Base0 Inc10 setting.

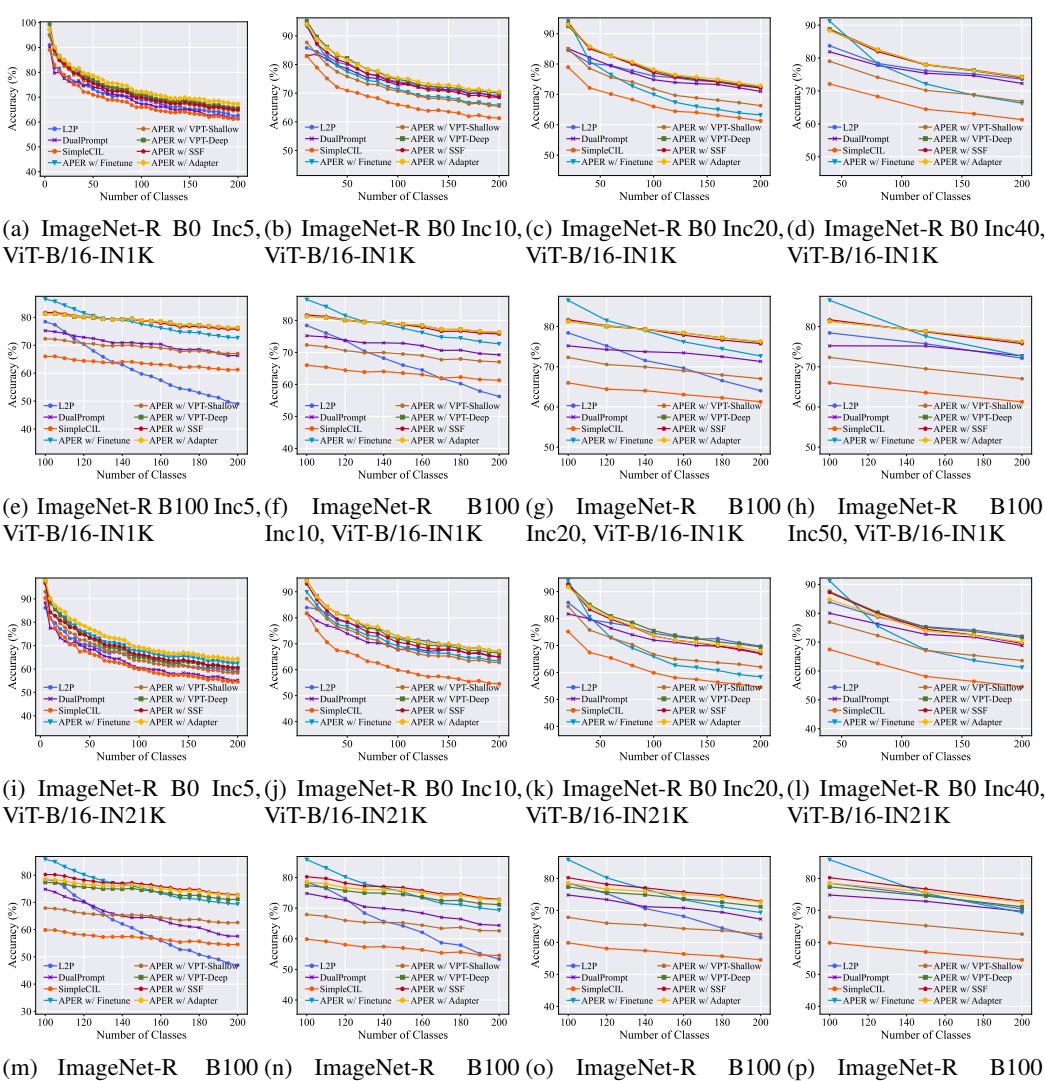

Figure 18: Experimental results on ImageNet-R. **(a)~(h):** Incremental performance comparison with **ViT-B/16-IN1K** as backbone. **(i)~(p):** Incremental performance when using **ViT-B/16-IN21K** as backbone.

| Backbone | Method | $\bar{\mathcal{A}}$ | $\mathcal{A}_B$ |
|---|---|---|---|
| | L2P (Wang et al., 2022e) | 76.78 | 72.35 |
| | DualPrompt (Wang et al., 2022d) | 76.23 | 70.96 |
| ViT-B/16-IN1K | SimpleCIL | 67.06 | 61.28 |
| | APER w/ Finetune | 71.98 | 63.23 |
| | APER w/ VPT-Shallow | 72.50 | 66.28 |
| | APER w/ VPT-Deep | 79.00 | 72.38 |
| | APER w/ SSF | 78.80 | 72.07 |
| | APER w/ Adapter | **79.39** | **72.87** |
| | L2P (Wang et al., 2022e) | 75.46 | **69.77** |
| | DualPrompt (Wang et al., 2022d) | 73.10 | 67.18 |
| ViT-B/16-IN21K | SimpleCIL | 61.26 | 54.55 |
| | APER w/ Finetune | 68.54 | 58.37 |
| | APER w/ VPT-Shallow | 68.83 | 62.03 |
| | APER w/ VPT-Deep | **77.05** | 69.47 |
| | APER w/ SSF | 75.47 | 67.02 |
| | APER w/ Adapter | 75.82 | 67.95 |

Table 22: Results on ImageNet-R Base0 Inc20 setting.

| Backbone | Method | $\bar{\mathcal{A}}$ | $\mathcal{A}_B$ |
|---|---|---|---|
| | L2P (Wang et al., 2022e) | 77.40 | 73.59 |
| | DualPrompt (Wang et al., 2022d) | 76.39 | 72.29 |
| ViT-B/16-IN1K | SimpleCIL | 65.84 | 61.28 |
| | APER w/ Finetune | 75.32 | 66.28 |
| | APER w/ VPT-Shallow | 71.80 | 66.83 |
| | APER w/ VPT-Deep | **79.99** | **74.27** |
| | APER w/ SSF | 79.70 | 73.97 |
| | APER w/ Adapter | 79.85 | 74.23 |
| | L2P (Wang et al., 2022e) | 76.82 | **72.07** |
| | DualPrompt (Wang et al., 2022d) | 73.91 | 68.81 |
| ViT-B/16-IN21K | SimpleCIL | 59.81 | 54.55 |
| | APER w/ Finetune | 71.77 | 61.25 |
| | APER w/ VPT-Shallow | 69.06 | 63.62 |
| | APER w/ VPT-Deep | **77.64** | 71.63 |
| | APER w/ SSF | 76.72 | 69.53 |
| | APER w/ Adapter | 76.03 | 70.00 |

Table 23: Results on ImageNet-R Base0 Inc40 setting.

| Backbone | Method | $\bar{\mathcal{A}}$ | $\mathcal{A}_B$ |
|---|---|---|---|
| | L2P (Wang et al., 2022e) | 61.46 | 49.03 |
| | DualPrompt (Wang et al., 2022d) | 70.54 | 66.31 |
| ViT-B/16-IN1K | SimpleCIL | 63.41 | 61.28 |
| | APER w/ Finetune | 78.13 | 72.67 |
| | APER w/ VPT-Shallow | 69.39 | 67.03 |
| | APER w/ VPT-Deep | 78.69 | 76.18 |
| | APER w/ SSF | 78.53 | 75.75 |
| | APER w/ Adapter | **78.72** | **76.30** |
| | L2P (Wang et al., 2022e) | 60.55 | 46.94 |
| | DualPrompt (Wang et al., 2022d) | 64.96 | 57.57 |
| ViT-B/16-IN21K | SimpleCIL | 56.88 | 54.55 |
| | APER w/ Finetune | 75.79 | 69.35 |
| | APER w/ VPT-Shallow | 64.89 | 62.58 |
| | APER w/ VPT-Deep | 74.15 | 71.15 |
| | APER w/ SSF | **76.42** | **72.87** |
| | APER w/ Adapter | 75.47 | 72.63 |

Table 24: Results on ImageNet-R Base100 Inc5 setting.

| Backbone | Method | $\bar{\mathcal{A}}$ | $\mathcal{A}_B$ |
|---|---|---|---|
| | L2P (Wang et al., 2022e) | 66.69 | 56.25 |
| | DualPrompt (Wang et al., 2022d) | 72.26 | 69.25 |
| ViT-B/16-IN1K | SimpleCIL | 63.40 | 61.28 |
| | APER w/ Finetune | 78.18 | 72.67 |
| | APER w/ VPT-Shallow | 69.37 | 67.03 |
| | APER w/ VPT-Deep | 78.67 | 76.18 |
| | APER w/ SSF | 78.50 | 75.75 |
| | APER w/ Adapter | **78.70** | **76.30** |
| | L2P (Wang et al., 2022e) | 64.87 | 53.38 |
| | DualPrompt (Wang et al., 2022d) | 69.24 | 64.37 |
| ViT-B/16-IN21K | SimpleCIL | 56.87 | 54.55 |
| | APER w/ Finetune | 75.86 | 69.35 |
| | APER w/ VPT-Shallow | 64.91 | 62.58 |
| | APER w/ VPT-Deep | 74.12 | 71.15 |
| | APER w/ SSF | **76.39** | **72.87** |
| | APER w/ Adapter | 75.46 | 72.63 |

Table 25: Results on ImageNet-R Base100 Inc10 setting.

| Backbone | Method | $\bar{\mathcal{A}}$ | $\mathcal{A}_B$ |
|---|---|---|---|
| | L2P (Wang et al., 2022e) | 70.91 | 64.04 |
| | DualPrompt (Wang et al., 2022d) | 73.42 | 71.32 |
| ViT-B/16-IN1K | SimpleCIL | 63.52 | 61.28 |
| | APER w/ Finetune | 78.39 | 72.67 |
| | APER w/ VPT-Shallow | 69.47 | 67.03 |
| | APER w/ VPT-Deep | 78.75 | 76.18 |
| | APER w/ SSF | 78.59 | 75.75 |
| | APER w/ Adapter | **78.78** | **76.30** |
| | L2P (Wang et al., 2022e) | 69.75 | 61.55 |
| | DualPrompt (Wang et al., 2022d) | 71.14 | 67.29 |
| ViT-B/16-IN21K | SimpleCIL | 57.00 | 54.55 |
| | APER w/ Finetune | 76.08 | 69.35 |
| | APER w/ VPT-Shallow | 64.97 | 62.55 |
| | APER w/ VPT-Deep | 74.20 | 71.15 |
| | APER w/ SSF | **76.45** | **72.87** |
| | APER w/ Adapter | 75.51 | 72.63 |

Table 26: Results on ImageNet-R Base100 Inc20 setting.

| Backbone | Method | $\bar{\mathcal{A}}$ | $\mathcal{A}_B$ |
|---|---|---|---|
| | L2P (Wang et al., 2022e) | 75.42 | 72.13 |
| | DualPrompt (Wang et al., 2022d) | 74.36 | 72.72 |
| ViT-B/16-IN1K | SimpleCIL | 63.61 | 61.28 |
| | APER w/ Finetune | **78.92** | 72.67 |
| | APER w/ VPT-Shallow | 69.62 | 67.03 |
| | APER w/ VPT-Deep | 78.78 | 76.18 |
| | APER w/ SSF | 78.71 | 75.75 |
| | APER w/ Adapter | 78.81 | **76.30** |
| | L2P (Wang et al., 2022e) | 74.60 | 70.51 |
| | DualPrompt (Wang et al., 2022d) | 72.45 | 69.71 |
| ViT-B/16-IN21K | SimpleCIL | 57.14 | 54.55 |
| | APER w/ Finetune | **76.84** | 69.35 |
| | APER w/ VPT-Shallow | 65.24 | 62.58 |
| | APER w/ VPT-Deep | 74.33 | 71.15 |
| | APER w/ SSF | 76.60 | **72.87** |
| | APER w/ Adapter | 75.66 | 72.63 |

Table 27: Results on ImageNet-R Base100 Inc50 setting.

## D.4 IMAGENET-A RESULTS

For ImageNet-A, we design 8 dataset splits to divide these 200 classes, namely **Base0 Inc5, Base0 Inc10, Base0 Inc20, Base0 Inc40, Base 100 Inc5, Base100 Inc10, Base100 Inc20, Base100 Inc50.** We report the results in Table 28, 29, 30, 31, 32, 33, 34, 35 and plot the corresponding incremental performance in Figure 19.

| Backbone | Method | $\bar{\mathcal{A}}$ | $\mathcal{A}_B$ |
|---|---|---|---|
| | L2P (Wang et al., 2022e) | 41.20 | 30.59 |
| | DualPrompt (Wang et al., 2022d) | 50.39 | 40.35 |
| ViT-B/16-IN1K | SimpleCIL | 61.20 | 49.44 |
| | APER w/ Finetune | **63.51** | **52.01** |
| | APER w/ VPT-Shallow | 54.82 | 43.65 |
| | APER w/ VPT-Deep | 58.87 | 47.86 |
| | APER w/ SSF | 63.30 | **52.01** |
| | APER w/ Adapter | 61.29 | 49.51 |
| | L2P (Wang et al., 2022e) | 35.53 | 26.12 |
| | DualPrompt (Wang et al., 2022d) | 46.12 | 32.82 |
| ViT-B/16-IN21K | SimpleCIL | 61.17 | 48.91 |
| | APER w/ Finetune | 61.60 | 49.44 |
| | APER w/ VPT-Shallow | 49.69 | 37.59 |
| | APER w/ VPT-Deep | 55.92 | 43.91 |
| | APER w/ SSF | **62.97** | **50.49** |
| | APER w/ Adapter | 61.26 | 48.98 |

Table 28: Results on ImageNet-A Base0 Inc5 setting.

| Backbone | Method | $\bar{\mathcal{A}}$ | $\mathcal{A}_B$ |
|---|---|---|---|
| | L2P (Wang et al., 2022e) | 47.16 | 38.48 |
| | DualPrompt (Wang et al., 2022d) | 52.56 | 42.68 |
| ViT-B/16-IN1K | SimpleCIL | 60.50 | 49.44 |
| | APER w/ Finetune | 61.57 | 50.76 |
| | APER w/ VPT-Shallow | 57.72 | 46.15 |
| | APER w/ VPT-Deep | 60.59 | 48.72 |
| | APER w/ SSF | **62.81** | **51.48** |
| | APER w/ Adapter | 60.53 | 49.57 |
| | L2P (Wang et al., 2022e) | 41.46 | 34.86 |
| | DualPrompt (Wang et al., 2022d) | 51.15 | 39.64 |
| ViT-B/16-IN21K | SimpleCIL | 60.63 | 48.91 |
| | APER w/ Finetune | 60.68 | 48.58 |
| | APER w/ VPT-Shallow | 58.85 | 47.66 |
| | APER w/ VPT-Deep | 58.00 | 46.28 |
| | APER w/ SSF | **62.80** | **51.48** |
| | APER w/ Adapter | 60.59 | 48.91 |

Table 29: Results on ImageNet-A Base0 Inc10 setting.

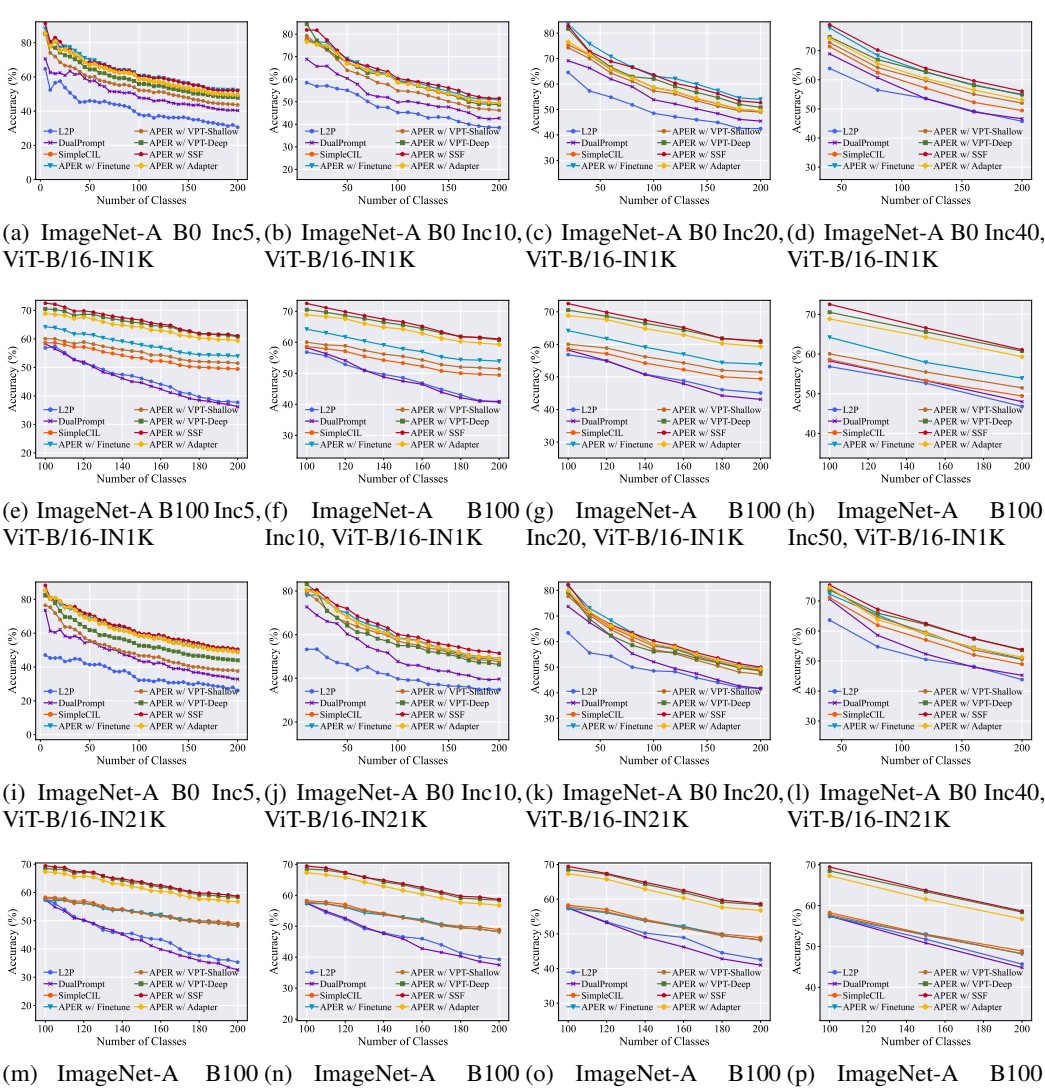

Figure 19: Experimental results on ImageNet-A. (a)∼(h): Incremental performance comparison with **ViT-B/16-IN1K** as backbone. (i)∼(p): Incremental performance when using **ViT-B/16-IN21K** as backbone.

| Backbone | Method | $\bar{\mathcal{A}}$ | $\mathcal{A}_B$ |
|---|---|---|---|
| | L2P (Wang et al., 2022e) | 50.01 | 42.47 |
| | DualPrompt (Wang et al., 2022d) | 55.25 | 45.43 |
| ViT-B/16-IN1K | SimpleCIL | 59.67 | 49.44 |
| | APER w/ Finetune | **64.79** | **53.98** |
| | APER w/ VPT-Shallow | 58.62 | 48.98 |
| | APER w/ VPT-Deep | 61.91 | 50.82 |
| | APER w/ SSF | 63.59 | 52.67 |
| | APER w/ Adapter | 59.89 | 49.51 |
| | L2P (Wang et al., 2022e) | 49.39 | 41.71 |
| | DualPrompt (Wang et al., 2022d) | 53.71 | 41.67 |
| ViT-B/16-IN21K | SimpleCIL | 59.77 | 48.91 |
| | APER w/ Finetune | 61.01 | 49.57 |
| | APER w/ VPT-Shallow | 58.39 | 47.20 |
| | APER w/ VPT-Deep | 58.48 | 48.52 |
| | APER w/ SSF | **61.30** | **50.03** |
| | APER w/ Adapter | 60.47 | 49.37 |

Table 30: Results on ImageNet-A Base0 Inc20 setting.

| Backbone | Method | $\bar{\mathcal{A}}$ | $\mathcal{A}_B$ |
|---|---|---|---|
| | L2P (Wang et al., 2022e) | 53.79 | 45.74 |
| | DualPrompt (Wang et al., 2022d) | 55.73 | 46.62 |
| ViT-B/16-IN1K | SimpleCIL | 58.53 | 49.44 |
| | APER w/ Finetune | 64.36 | 54.77 |
| | APER w/ VPT-Shallow | 60.74 | 52.01 |
| | APER w/ VPT-Deep | 63.47 | 54.97 |
| | APER w/ SSF | **65.74** | **56.09** |
| | APER w/ Adapter | 61.95 | 53.06 |
| | L2P (Wang et al., 2022e) | 52.16 | 43.81 |
| | DualPrompt (Wang et al., 2022d) | 54.92 | 45.25 |
| ViT-B/16-IN21K | SimpleCIL | 58.19 | 48.91 |
| | APER w/ Finetune | 60.27 | 50.63 |
| | APER w/ VPT-Shallow | 60.51 | 50.49 |
| | APER w/ VPT-Deep | 62.54 | 53.59 |
| | APER w/ SSF | **63.23** | **53.79** |
| | APER w/ Adapter | 60.58 | 51.28 |

Table 31: Results on ImageNet-A Base0 Inc40 setting.

| Backbone | Method | $\bar{\mathcal{A}}$ | $\mathcal{A}_B$ |
|---|---|---|---|
| ViT-B/16-IN1K | L2P (Wang et al., 2022e) | 46.12 | 37.74 |
| | DualPrompt (Wang et al., 2022d) | 45.21 | 36.19 |
| | SimpleCIL | 53.50 | 49.44 |
| | APER w/ Finetune | 58.26 | 53.92 |
| | APER w/ VPT-Shallow | 55.37 | 51.48 |
| | APER w/ VPT-Deep | 65.46 | 60.76 |
| | APER w/ SSF | **66.27** | **61.09** |
| | APER w/ Adapter | 63.90 | 59.32 |
| ViT-B/16-IN21K | L2P (Wang et al., 2022e) | 44.56 | 35.31 |
| | DualPrompt (Wang et al., 2022d) | 43.26 | 32.63 |
| | SimpleCIL | 53.26 | 48.91 |
| | APER w/ Finetune | 52.78 | 48.19 |
| | APER w/ VPT-Shallow | 52.81 | 48.26 |
| | APER w/ VPT-Deep | 63.23 | 58.39 |
| | APER w/ SSF | **63.73** | **58.66** |
| | APER w/ Adapter | 61.73 | 56.75 |

Table 32: Results on ImageNet-A Base100 Inc5 setting.

| Backbone | Method | $\bar{\mathcal{A}}$ | $\mathcal{A}_B$ |
|---|---|---|---|
| ViT-B/16-IN1K | L2P (Wang et al., 2022e) | 48.28 | 40.71 |
| | DualPrompt (Wang et al., 2022d) | 48.24 | 40.95 |
| | SimpleCIL | 53.54 | 49.44 |
| | APER w/ Finetune | 58.30 | 53.92 |
| | APER w/ VPT-Shallow | 55.42 | 51.48 |
| | APER w/ VPT-Deep | 65.45 | 60.76 |
| | APER w/ SSF | **66.28** | **61.09** |
| | APER w/ Adapter | 63.94 | 59.32 |
| ViT-B/16-IN21K | L2P (Wang et al., 2022e) | 47.14 | 39.30 |
| | DualPrompt (Wang et al., 2022d) | 46.27 | 37.53 |
| | SimpleCIL | 53.33 | 48.91 |
| | APER w/ Finetune | 52.82 | 48.19 |
| | APER w/ VPT-Shallow | 52.87 | 48.26 |
| | APER w/ VPT-Deep | 63.30 | 58.39 |
| | APER w/ SSF | **63.76** | **58.66** |
| | APER w/ Adapter | 61.77 | 56.75 |

Table 33: Results on ImageNet-A Base100 Inc10 setting.

| Backbone | Method | $\bar{\mathcal{A}}$ | $\mathcal{A}_B$ |
|---|---|---|---|
| ViT-B/16-IN1K | L2P (Wang et al., 2022e) | 50.44 | 45.08 |
| | DualPrompt (Wang et al., 2022d) | 49.87 | 43.10 |
| | SimpleCIL | 53.62 | 49.44 |
| | APER w/ Finetune | 58.40 | 53.92 |
| | APER w/ VPT-Shallow | 55.50 | 51.48 |
| | APER w/ VPT-Deep | 65.44 | 60.76 |
| | APER w/ SSF | **66.31** | **61.09** |
| | APER w/ Adapter | 63.97 | 59.32 |
| ViT-B/16-IN21K | L2P (Wang et al., 2022e) | 49.52 | 42.59 |
| | DualPrompt (Wang et al., 2022d) | 48.28 | 40.97 |
| | SimpleCIL | 53.38 | 48.91 |
| | APER w/ Finetune | 52.83 | 48.19 |
| | APER w/ VPT-Shallow | 52.90 | 48.26 |
| | APER w/ VPT-Deep | 63.24 | 58.39 |
| | APER w/ SSF | **63.76** | **58.66** |
| | APER w/ Adapter | 61.77 | 56.75 |

Table 34: Results on ImageNet-A Base100 Inc20 setting.

| Backbone | Method | $\bar{\mathcal{A}}$ | $\mathcal{A}_B$ |
|---|---|---|---|
| ViT-B/16-IN1K | L2P (Wang et al., 2022e) | 52.10 | 46.81 |
| | DualPrompt (Wang et al., 2022d) | 53.18 | 48.01 |
| | SimpleCIL | 53.79 | 49.44 |
| | APER w/ Finetune | 58.67 | 53.92 |
| | APER w/ VPT-Shallow | 55.67 | 51.48 |
| | APER w/ VPT-Deep | 65.62 | 60.76 |
| | APER w/ SSF | **66.73** | **61.09** |
| | APER w/ Adapter | 64.14 | 59.32 |
| ViT-B/16-IN21K | L2P (Wang et al., 2022e) | 51.62 | 45.68 |
| | DualPrompt (Wang et al., 2022d) | 51.04 | 44.83 |
| | SimpleCIL | 53.38 | 48.91 |
| | APER w/ Finetune | 52.79 | 48.19 |
| | APER w/ VPT-Shallow | 52.92 | 48.26 |
| | APER w/ VPT-Deep | 63.43 | 58.39 |
| | APER w/ SSF | **63.95** | **58.66** |
| | APER w/ Adapter | 61.87 | 56.75 |

Table 35: Results on ImageNet-A Base100 Inc50 setting.

## D.5 OBJECTNET RESULTS

For ObjectNet, we design 8 dataset splits to divide these 200 classes, namely **Base0 Inc5, Base0 Inc10, Base0 Inc20, Base0 Inc40, Base 100 Inc5, Base100 Inc10, Base100 Inc20, Base100 Inc50.** We report the results in Table 36, 37, 38, 39, 40, 41, 42, 43 and plot the corresponding incremental performance in Figure 20.

| Backbone | Method | $\bar{\mathcal{A}}$ | $\mathcal{A}_B$ |
|---|---|---|---|
| ViT-B/16-IN1K | L2P (Wang et al., 2022e) | 57.38 | 46.38 |
| | DualPrompt (Wang et al., 2022d) | 41.68 | 37.61 |
| | SimpleCIL | 63.71 | 51.13 |
| | APER w/ Finetune | **68.21** | **55.73** |
| | APER w/ VPT-Shallow | 64.81 | 52.10 |
| | APER w/ VPT-Deep | 63.08 | 50.11 |
| | APER w/ SSF | 67.29 | 54.72 |
| | APER w/ Adapter | 64.21 | 51.39 |
| ViT-B/16-IN21K | L2P (Wang et al., 2022e) | 58.13 | 46.68 |
| | DualPrompt (Wang et al., 2022d) | 50.87 | 41.75 |
| | SimpleCIL | 66.04 | 53.59 |
| | APER w/ Finetune | **68.04** | **55.75** |
| | APER w/ VPT-Shallow | 62.93 | 50.42 |
| | APER w/ VPT-Deep | 64.13 | 51.13 |
| | APER w/ SSF | 67.74 | 55.13 |
| | APER w/ Adapter | 66.31 | 53.83 |

Table 36: Results on ObjectNet Base0 Inc5 setting.

| Backbone | Method | $\bar{\mathcal{A}}$ | $\mathcal{A}_B$ |
|---|---|---|---|
| ViT-B/16-IN1K | L2P (Wang et al., 2022e) | 62.70 | 51.36 |
| | DualPrompt (Wang et al., 2022d) | 54.37 | 47.28 |
| | SimpleCIL | 63.12 | 51.13 |
| | APER w/ Finetune | 64.42 | 50.92 |
| | APER w/ VPT-Shallow | 64.51 | 52.66 |
| | APER w/ VPT-Deep | 66.66 | 53.62 |
| | APER w/ SSF | **67.28** | **54.57** |
| | APER w/ Adapter | 66.19 | 53.82 |
| ViT-B/16-IN21K | L2P (Wang et al., 2022e) | 63.78 | 52.19 |
| | DualPrompt (Wang et al., 2022d) | 59.27 | 49.33 |
| | SimpleCIL | 65.45 | 53.59 |
| | APER w/ Finetune | 61.41 | 48.34 |
| | APER w/ VPT-Shallow | 64.54 | 52.53 |
| | APER w/ VPT-Deep | 67.83 | 54.65 |
| | APER w/ SSF | **69.15** | **56.64** |
| | APER w/ Adapter | 67.18 | 55.24 |

Table 37: Results on ObjectNet Base0 Inc10 setting.

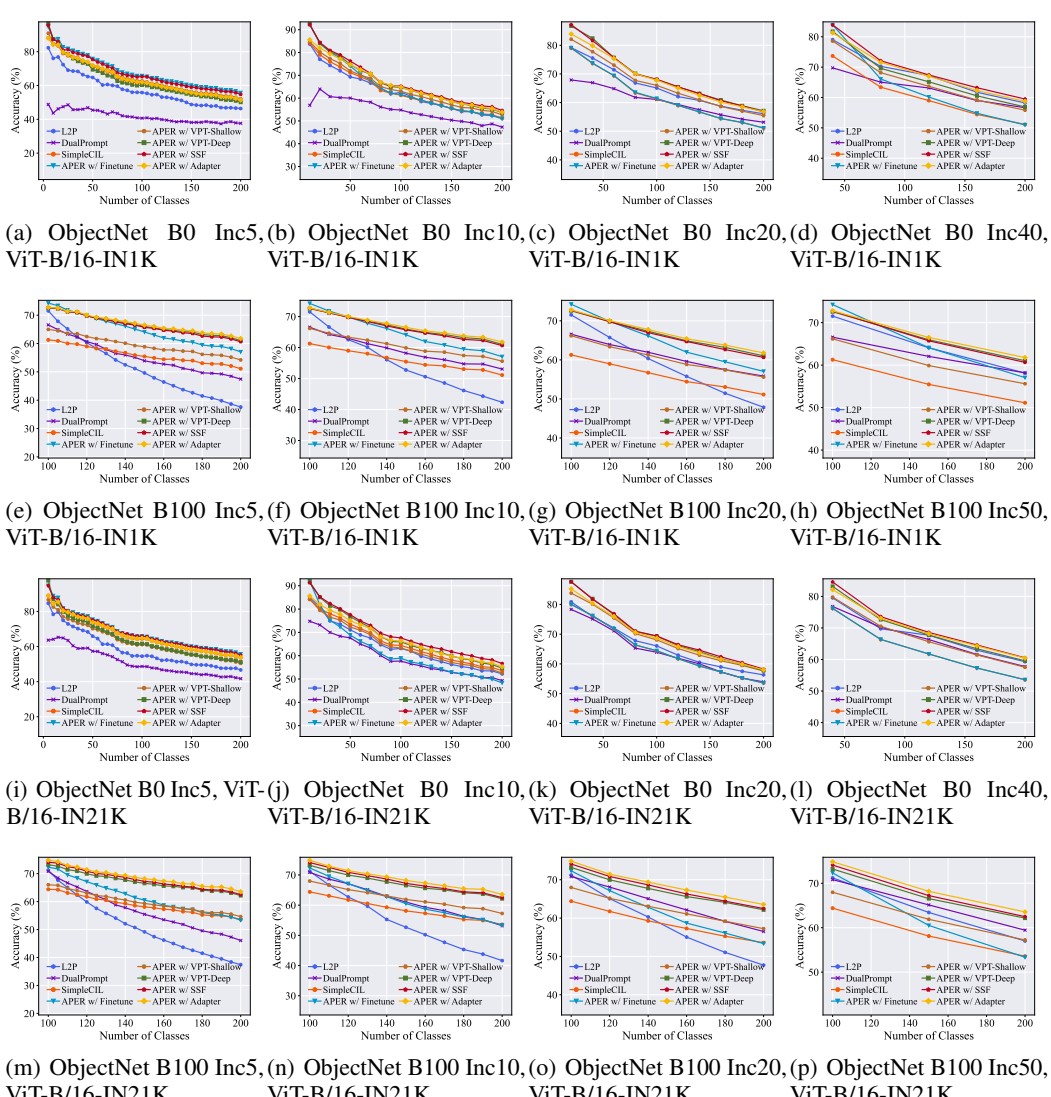

Figure 20: Experimental results on ObjectNet. **(a)~(h):** Incremental performance comparison with **ViT-B/16-IN1K** as backbone. **(i)~(p):** Incremental performance when using **ViT-B/16-IN21K** as backbone.

| Backbone | Method | $\bar{\mathcal{A}}$ | $\mathcal{A}_B$ |
|---|---|---|---|
| | L2P (Wang et al., 2022e) | 65.31 | 56.08 |
| | DualPrompt (Wang et al., 2022d) | 60.25 | 53.14 |
| ViT-B/16-IN1K | SimpleCIL | 62.11 | 51.13 |
| | APER w/ Finetune | 62.18 | 51.13 |
| | APER w/ VPT-Shallow | 66.20 | 55.45 |
| | APER w/ VPT-Deep | **68.85** | **57.14** |
| | APER w/ SSF | 68.75 | 56.79 |
| | APER w/ Adapter | 68.01 | 56.82 |
| | L2P (Wang et al., 2022e) | 65.89 | 56.31 |
| | DualPrompt (Wang et al., 2022d) | 64.20 | 53.95 |
| ViT-B/16-IN21K | SimpleCIL | 64.58 | 53.59 |
| | APER w/ Finetune | 64.59 | 53.58 |
| | APER w/ VPT-Shallow | 68.37 | 57.60 |
| | APER w/ VPT-Deep | 69.58 | 57.79 |
| | APER w/ SSF | **69.87** | **58.24** |
| | APER w/ Adapter | 68.97 | 58.03 |

Table 38: Results on ObjectNet Base0 Inc20 setting.

| Backbone | Method | $\bar{\mathcal{A}}$ | $\mathcal{A}_B$ |
|---|---|---|---|
| | L2P (Wang et al., 2022e) | 67.29 | 58.20 |
| | DualPrompt (Wang et al., 2022d) | 62.72 | 56.64 |
| ViT-B/16-IN1K | SimpleCIL | 60.32 | 51.13 |
| | APER w/ Finetune | 63.19 | 50.97 |
| | APER w/ VPT-Shallow | 65.08 | 55.82 |
| | APER w/ VPT-Deep | 66.79 | 56.89 |
| | APER w/ SSF | **69.23** | **59.49** |
| | APER w/ Adapter | 68.18 | 58.72 |
| | L2P (Wang et al., 2022e) | 67.84 | 59.31 |
| | DualPrompt (Wang et al., 2022d) | 66.54 | 57.77 |
| ViT-B/16-IN21K | SimpleCIL | 63.03 | 53.59 |
| | APER w/ Finetune | 63.04 | 53.59 |
| | APER w/ VPT-Shallow | 66.98 | 57.54 |
| | APER w/ VPT-Deep | 69.31 | 59.58 |
| | APER w/ SSF | **70.37** | **60.53** |
| | APER w/ Adapter | 69.63 | 60.43 |

Table 39: Results on ObjectNet Base0 Inc40 setting.

| Backbone | Method | $\bar{\mathcal{A}}$ | $\mathcal{A}_B$ |
|---|---|---|---|
| | L2P (Wang et al., 2022e) | 51.12 | 37.59 |
| | DualPrompt (Wang et al., 2022d) | 55.31 | 47.44 |
| | SimpleCIL | 56.02 | 51.13 |
| ViT-B/16-IN1K | APER w/ Finetune | 64.75 | 57.06 |
| | APER w/ VPT-Shallow | 59.38 | 54.22 |
| | APER w/ VPT-Deep | 66.55 | 61.03 |
| | APER w/ SSF | 66.31 | 60.64 |
| | APER w/ Adapter | **66.99** | **61.78** |
| | L2P (Wang et al., 2022e) | 50.85 | 37.44 |
| | DualPrompt (Wang et al., 2022d) | 56.65 | 46.08 |
| | SimpleCIL | 58.62 | 53.59 |
| ViT-B/16-IN21K | APER w/ Finetune | 61.65 | 53.35 |
| | APER w/ VPT-Shallow | 59.85 | 54.63 |
| | APER w/ VPT-Deep | 67.16 | 62.16 |
| | APER w/ SSF | 67.83 | 62.49 |
| | APER w/ Adapter | **68.66** | **63.55** |

Table 40: Results on ObjectNet Base100 Inc5 setting.

| Backbone | Method | $\bar{\mathcal{A}}$ | $\mathcal{A}_B$ |
|---|---|---|---|
| | L2P (Wang et al., 2022e) | 54.69 | 42.34 |
| | DualPrompt (Wang et al., 2022d) | 58.96 | 53.06 |
| | SimpleCIL | 56.00 | 51.13 |
| ViT-B/16-IN1K | APER w/ Finetune | 64.76 | 57.06 |
| | APER w/ VPT-Shallow | 60.48 | 55.61 |
| | APER w/ VPT-Deep | 66.51 | 61.03 |
| | APER w/ SSF | 66.26 | 60.64 |
| | APER w/ Adapter | **66.94** | **61.78** |
| | L2P (Wang et al., 2022e) | 54.26 | 41.63 |
| | DualPrompt (Wang et al., 2022d) | 61.61 | 53.18 |
| | SimpleCIL | 58.61 | 53.59 |
| ViT-B/16-IN21K | APER w/ Finetune | 61.63 | 53.35 |
| | APER w/ VPT-Shallow | 62.32 | 57.24 |
| | APER w/ VPT-Deep | 67.14 | 62.16 |
| | APER w/ SSF | 67.81 | 62.49 |
| | APER w/ Adapter | **68.65** | **63.55** |

Table 41: Results on ObjectNet Base100 Inc10 setting.

| Backbone | Method | $\bar{\mathcal{A}}$ | $\mathcal{A}_B$ |
|---|---|---|---|
| | L2P (Wang et al., 2022e) | 58.78 | 47.87 |
| | DualPrompt (Wang et al., 2022d) | 60.90 | 55.85 |
| | SimpleCIL | 55.94 | 51.13 |
| ViT-B/16-IN1K | APER w/ Finetune | 64.80 | 57.06 |
| | APER w/ VPT-Shallow | 60.45 | 55.61 |
| | APER w/ VPT-Deep | 66.47 | 61.03 |
| | APER w/ SSF | 66.24 | 60.64 |
| | APER w/ Adapter | **66.93** | **61.78** |
| | L2P (Wang et al., 2022e) | 58.40 | 47.73 |
| | DualPrompt (Wang et al., 2022d) | 63.62 | 56.51 |
| | SimpleCIL | 58.60 | 53.59 |
| ViT-B/16-IN21K | APER w/ Finetune | 61.74 | 53.35 |
| | APER w/ VPT-Shallow | 62.29 | 57.24 |
| | APER w/ VPT-Deep | 67.13 | 62.16 |
| | APER w/ SSF | 67.80 | 62.49 |
| | APER w/ Adapter | **68.67** | **63.55** |

Table 42: Results on ObjectNet Base100 Inc20 setting.

| Backbone | Method | $\bar{\mathcal{A}}$ | $\mathcal{A}_B$ |
|---|---|---|---|
| | L2P (Wang et al., 2022e) | 64.58 | 58.12 |
| | DualPrompt (Wang et al., 2022d) | 62.27 | 58.18 |
| | SimpleCIL | 55.96 | 51.13 |
| ViT-B/16-IN1K | APER w/ Finetune | 65.14 | 57.06 |
| | APER w/ VPT-Shallow | 60.56 | 55.61 |
| | APER w/ VPT-Deep | 66.52 | 61.03 |
| | APER w/ SSF | 66.36 | 60.64 |
| | APER w/ Adapter | **67.02** | **61.78** |
| | L2P (Wang et al., 2022e) | 63.89 | 57.02 |
| | DualPrompt (Wang et al., 2022d) | 65.13 | 59.44 |
| | SimpleCIL | 58.70 | 53.59 |
| ViT-B/16-IN21K | APER w/ Finetune | 62.06 | 53.35 |
| | APER w/ VPT-Shallow | 62.37 | 57.24 |
| | APER w/ VPT-Deep | 67.31 | 62.16 |
| | APER w/ SSF | 67.94 | 62.49 |
| | APER w/ Adapter | **68.87** | **63.55** |

Table 43: Results on ObjectNet Base100 Inc50 setting.

## D.6 OMNIBENCHMARK RESULTS

For OmniBenchmark, we design 6 dataset splits to divide these 300 classes, namely **Base0 Inc15, Base0 Inc30, Base0 Inc50, Base150 Inc15, Base 150 Inc30, Base150 Inc50.** We report the results in Table 44, 45, 46, 47, 48, 49, and plot the corresponding incremental performance in Figure 21.

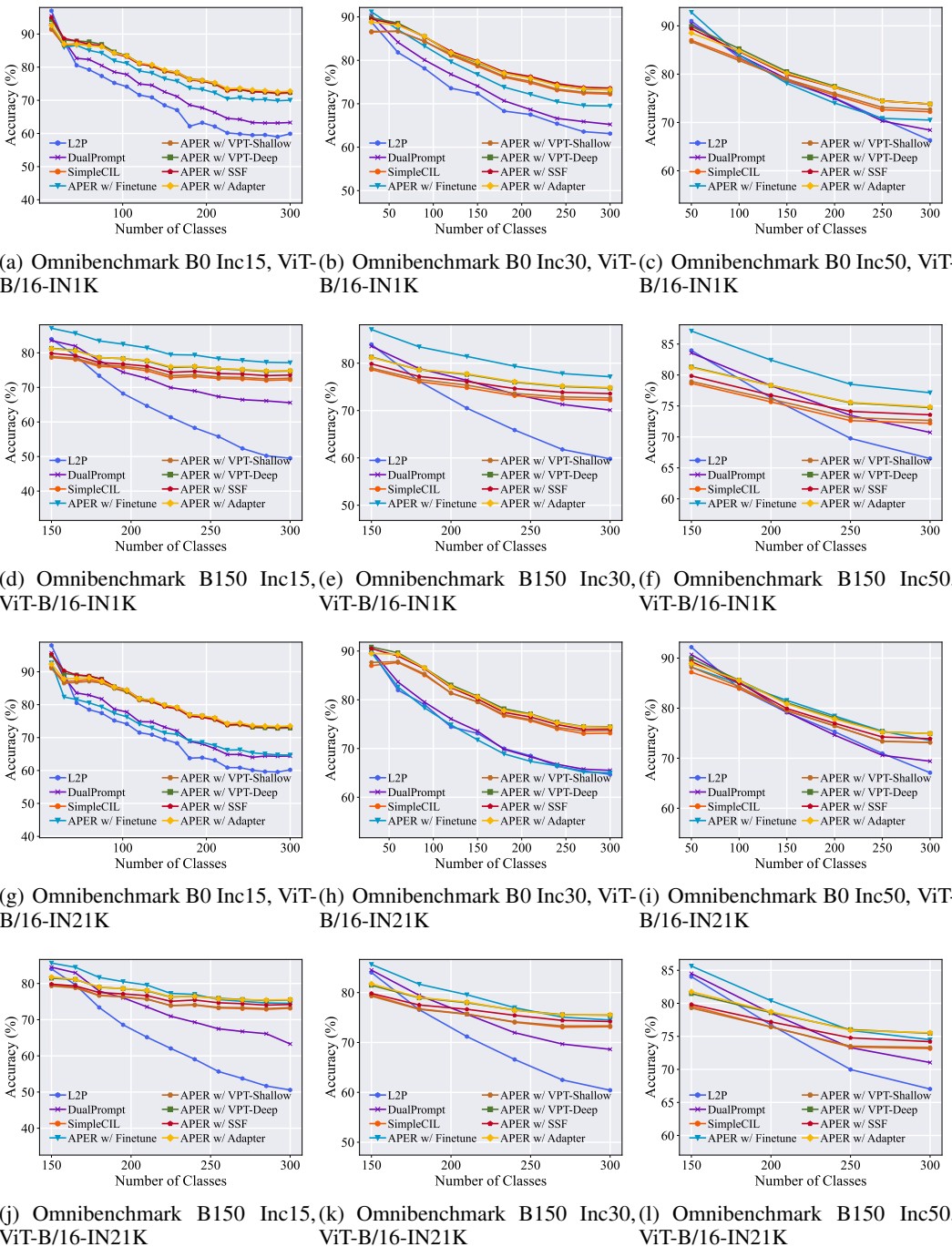

(a) Omnibenchmark B0 Inc15, ViT-B/16-IN1K

(b) Omnibenchmark B0 Inc30, ViT-B/16-IN1K

(c) Omnibenchmark B0 Inc50, ViT-B/16-IN1K

(d) Omnibenchmark B150 Inc15, ViT-B/16-IN1K

(e) Omnibenchmark B150 Inc30, ViT-B/16-IN1K

(f) Omnibenchmark B150 Inc50, ViT-B/16-IN1K

(g) Omnibenchmark B0 Inc15, ViT-B/16-IN21K

(h) Omnibenchmark B0 Inc30, ViT-B/16-IN21K

(i) Omnibenchmark B0 Inc50, ViT-B/16-IN21K

(j) Omnibenchmark B150 Inc15, ViT-B/16-IN21K

(k) Omnibenchmark B150 Inc30, ViT-B/16-IN21K

(l) Omnibenchmark B150 Inc50, ViT-B/16-IN21K

Figure 21: Experimental results on Omnibenchmark. **(a)∼(f):** Incremental performance comparison with **ViT-B/16-IN1K** as backbone. **(g)∼(l):** Incremental performance when using **ViT-B/16-IN21K** as backbone.

| Backbone | Method | $\bar{\mathcal{A}}$ | $\mathcal{A}_B$ |
|---|---|---|---|
| ViT-B/16-IN1K | L2P (Wang et al., 2022e) | 69.73 | 59.90 |
| | DualPrompt (Wang et al., 2022d) | 73.13 | 63.29 |
| | SimpleCIL | 79.27 | 72.21 |
| | APER w/ Finetune | 77.39 | 70.04 |
| | APER w/ VPT-Shallow | 79.53 | 72.58 |
| | APER w/ VPT-Deep | **79.75** | 72.36 |
| | APER w/ SSF | 79.71 | 72.43 |
| | APER w/ Adapter | 79.62 | **72.75** |
| ViT-B/16-IN21K | L2P (Wang et al., 2022e) | 70.29 | 60.19 |
| | DualPrompt (Wang et al., 2022d) | 73.69 | 64.39 |
| | SimpleCIL | 79.96 | 73.15 |
| | APER w/ Finetune | 72.86 | 64.70 |
| | APER w/ VPT-Shallow | 79.72 | 73.12 |
| | APER w/ VPT-Deep | 80.56 | 72.90 |
| | APER w/ SSF | **80.59** | 73.28 |
| | APER w/ Adapter | 80.40 | **73.53** |

Table 44: Results on OmniBenchmark Base0 Inc15 setting.

| Backbone | Method | $\bar{\mathcal{A}}$ | $\mathcal{A}_B$ |
|---|---|---|---|
| ViT-B/16-IN1K | L2P (Wang et al., 2022e) | 72.27 | 63.12 |
| | DualPrompt (Wang et al., 2022d) | 74.26 | 65.24 |
| | SimpleCIL | 78.61 | 72.21 |
| | APER w/ Finetune | 77.35 | 69.49 |
| | APER w/ VPT-Shallow | 78.81 | 72.48 |
| | APER w/ VPT-Deep | 79.91 | 73.23 |
| | APER w/ SSF | **80.07** | **73.62** |
| | APER w/ Adapter | 79.77 | 73.25 |
| ViT-B/16-IN21K | L2P (Wang et al., 2022e) | 73.36 | 64.69 |
| | DualPrompt (Wang et al., 2022d) | 73.92 | 65.52 |
| | SimpleCIL | 79.34 | 73.15 |
| | APER w/ Finetune | 73.02 | 65.03 |
| | APER w/ VPT-Shallow | 79.63 | 73.68 |
| | APER w/ VPT-Deep | **81.05** | **74.47** |
| | APER w/ SSF | 80.53 | 74.00 |
| | APER w/ Adapter | 80.75 | 74.37 |

Table 45: Results on OmniBenchmark Base0 Inc30 setting.

| Backbone | Method | $\bar{\mathcal{A}}$ | $\mathcal{A}_B$ |
|---|---|---|---|
| ViT-B/16-IN1K | L2P (Wang et al., 2022e) | 77.51 | 66.30 |
| | DualPrompt (Wang et al., 2022d) | 77.84 | 68.42 |
| | SimpleCIL | 78.23 | 72.21 |
| | APER w/ Finetune | 78.37 | 70.49 |
| | APER w/ VPT-Shallow | 78.38 | 72.70 |
| | APER w/ VPT-Deep | **80.25** | **73.83** |
| | APER w/ SSF | 79.91 | 73.80 |
| | APER w/ Adapter | 79.83 | 73.82 |
| ViT-B/16-IN21K | L2P (Wang et al., 2022e) | 78.10 | 67.10 |
| | DualPrompt (Wang et al., 2022d) | 78.29 | 69.40 |
| | SimpleCIL | 78.95 | 73.15 |
| | APER w/ Finetune | 80.35 | 73.55 |
| | APER w/ VPT-Shallow | 79.16 | 73.18 |
| | APER w/ VPT-Deep | **80.80** | **74.92** |
| | APER w/ SSF | 79.90 | 73.90 |
| | APER w/ Adapter | 80.55 | **74.92** |

Table 46: Results on OmniBenchmark Base0 Inc50 setting.

| Backbone | Method | $\bar{\mathcal{A}}$ | $\mathcal{A}_B$ |
|---|---|---|---|
| ViT-B/16-IN1K | L2P (Wang et al., 2022e) | 63.35 | 49.51 |
| | DualPrompt (Wang et al., 2022d) | 72.24 | 65.57 |
| | SimpleCIL | 74.42 | 72.21 |
| | APER w/ Finetune | **80.87** | **77.14** |
| | APER w/ VPT-Shallow | 74.86 | 72.66 |
| | APER w/ VPT-Deep | 77.10 | 74.74 |
| | APER w/ SSF | 75.73 | 73.57 |
| | APER w/ Adapter | 77.16 | 74.84 |
| ViT-B/16-IN21K | L2P (Wang et al., 2022e) | 63.96 | 50.58 |
| | DualPrompt (Wang et al., 2022d) | 72.62 | 63.29 |
| | SimpleCIL | 75.25 | 73.15 |
| | APER w/ Finetune | **78.71** | 74.50 |
| | APER w/ VPT-Shallow | 75.26 | 73.33 |
| | APER w/ VPT-Deep | 77.55 | 75.46 |
| | APER w/ SSF | 76.20 | 74.20 |
| | APER w/ Adapter | 77.60 | **75.54** |

Table 47: Results on OmniBenchmark Base150 Inc15 setting.

| Backbone | Method | $\bar{\mathcal{A}}$ | $\mathcal{A}_B$ |
|---|---|---|---|
| ViT-B/16-IN1K | L2P (Wang et al., 2022e) | 69.68 | 59.79 |
| | DualPrompt (Wang et al., 2022d) | 75.61 | 70.09 |
| | SimpleCIL | 74.56 | 72.21 |
| | APER w/ Finetune | **81.05** | **77.14** |
| | APER w/ VPT-Shallow | 75.01 | 72.66 |
| | APER w/ VPT-Deep | 77.22 | 74.74 |
| | APER w/ SSF | 75.88 | 73.57 |
| | APER w/ Adapter | 77.29 | 74.84 |
| ViT-B/16-IN21K | L2P (Wang et al., 2022e) | 70.22 | 60.42 |
| | DualPrompt (Wang et al., 2022d) | 74.98 | 68.63 |
| | SimpleCIL | 75.38 | 73.15 |
| | APER w/ Finetune | **78.91** | 74.50 |
| | APER w/ VPT-Shallow | 75.39 | 73.33 |
| | APER w/ VPT-Deep | 77.65 | 75.46 |
| | APER w/ SSF | 76.34 | 74.20 |
| | APER w/ Adapter | 77.73 | **75.54** |

Table 48: Results on OmniBenchmark Base150 Inc30 setting.

| Backbone | Method | $\bar{\mathcal{A}}$ | $\mathcal{A}_B$ |
|---|---|---|---|
| ViT-B/16-IN1K | L2P (Wang et al., 2022e) | 74.14 | 66.51 |
| | DualPrompt (Wang et al., 2022d) | 76.52 | 70.73 |
| | SimpleCIL | 74.79 | 72.21 |
| | APER w/ Finetune | **81.29** | **77.14** |
| | APER w/ VPT-Shallow | 75.21 | 72.66 |
| | APER w/ VPT-Deep | 77.47 | 74.74 |
| | APER w/ SSF | 76.07 | 73.57 |
| | APER w/ Adapter | 77.48 | 74.84 |
| ViT-B/16-IN21K | L2P (Wang et al., 2022e) | 74.44 | 67.03 |
| | DualPrompt (Wang et al., 2022d) | 76.84 | 71.04 |
| | SimpleCIL | 75.64 | 73.15 |
| | APER w/ Finetune | **79.11** | 74.50 |
| | APER w/ VPT-Shallow | 75.64 | 73.33 |
| | APER w/ VPT-Deep | 77.88 | 75.46 |
| | APER w/ SSF | 76.50 | 74.20 |
| | APER w/ Adapter | 77.99 | **75.54** |

Table 49: Results on OmniBenchmark Base150 Inc50 setting.

## D.7 VTAB RESULTS

Since VTAB is a complex dataset with multiple domains, we select five domains to construct a cross-domain class-incremental learning setting. Specifically, we fix the domain order to "Resisc45 → DTD → Pets → EuroSAT → Flowers". Each domain contains 10 classes, and we formulate the VTAB Base0 Inc10 setting. We report the results in Table 50 and plot the corresponding incremental performance in Figure 22.

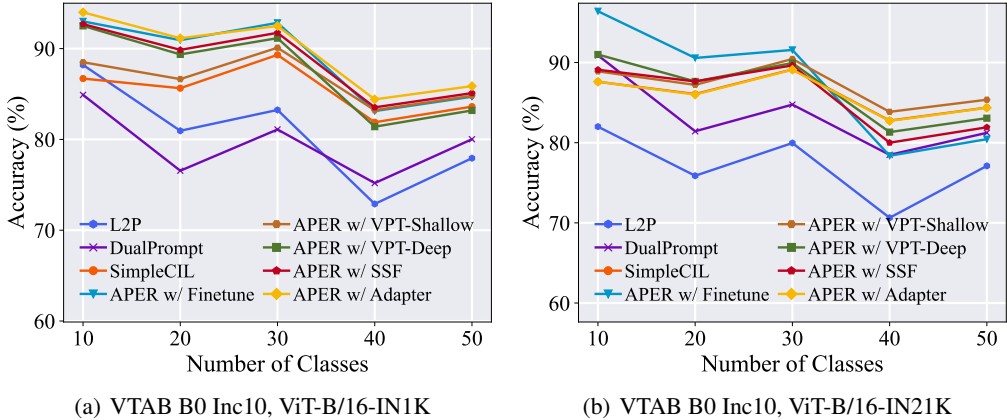

(a) VTAB B0 Inc10, ViT-B/16-IN1K          (b) VTAB B0 Inc10, ViT-B/16-IN21K

Figure 22: Experimental results on VTAB. **(a):** Incremental performance comparison with **ViT-B/16-IN1K** as backbone. **(b):** Incremental performance when using **ViT-B/16-IN21K** as backbone.

Table 50: Results on VTAB Base0 Inc10 setting.

| Backbone | Method | $\bar{\mathcal{A}}$ | $\mathcal{A}_B$ |
|---|---|---|---|
| | L2P (Wang et al., 2022e) | 80.64 | 77.93 |
| | DualPrompt (Wang et al., 2022d) | 79.55 | 80.02 |
| | SimpleCIL | 85.43 | 83.61 |
| ViT-B/16-IN1K | APER w/ Finetune | 88.92 | 84.70 |
| | APER w/ VPT-Shallow | 86.65 | 84.81 |
| | APER w/ VPT-Deep | 87.52 | 83.19 |
| | APER w/ SSF | 88.59 | 85.10 |
| | APER w/ Adapter | **89.58** | **85.87** |
| | L2P (Wang et al., 2022e) | 77.11 | 77.10 |
| | DualPrompt (Wang et al., 2022d) | 83.36 | 81.23 |
| | SimpleCIL | 85.99 | 84.38 |
| ViT-B/16-IN21K | APER w/ Finetune | **87.47** | 80.44 |
| | APER w/ VPT-Shallow | 87.15 | **85.36** |
| | APER w/ VPT-Deep | 86.59 | 83.06 |
| | APER w/ SSF | 85.66 | 81.92 |
| | APER w/ Adapter | 85.95 | 84.35 |

### D.8  APER WITH RESNET

In this section, we give the visualizations of APER to boost the performance of pre-trained ResNet. Specifically, we choose one split from each dataset and choose pre-trained ResNet18/50/101/152 as the backbone to evaluate the performance of APER. We report the numerical results in Table 51, 52, 53, 54, 55, 56 and plot the incremental performance in Figure 23. Results indicate APER also works competitively with ResNets.

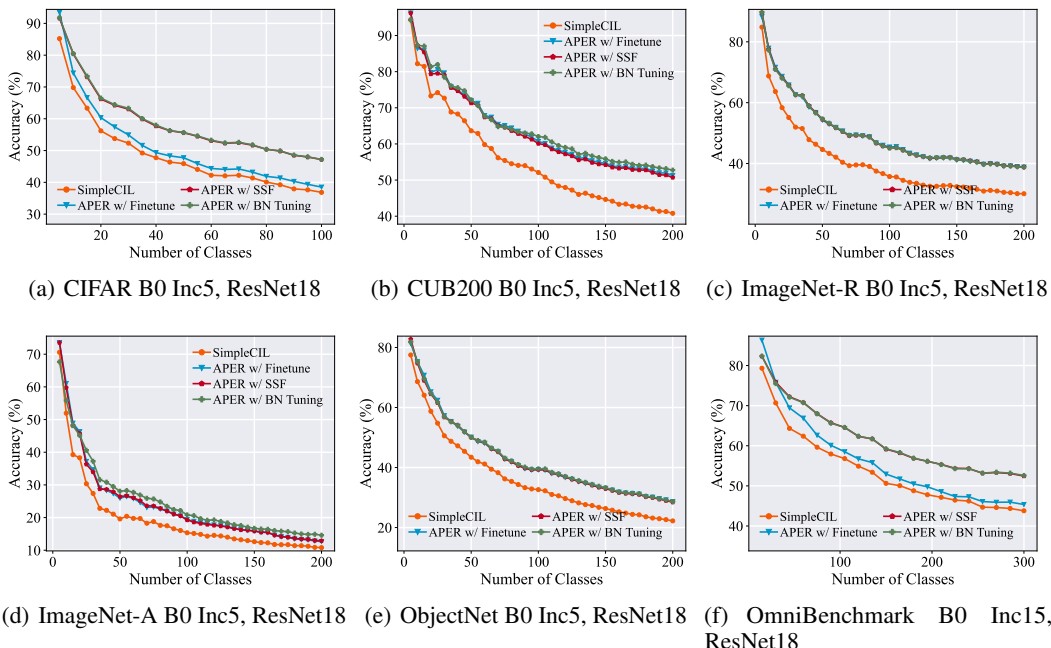

Figure 23: Experimental results with pre-trained **ResNet18** as backbone. APER **consistently improves the performance of pre-trained models in class-incremental learning.**

| Backbone | Method | $\bar{\mathcal{A}}$ | $\mathcal{A}_B$ |
|---|---|---|---|
| ResNet18 | SimpleCIL | 48.68 | 36.85 |
| | APER w/ Finetune | 51.37 | 38.51 |
| | APER w/ SSF | 58.80 | 47.17 |
| | APER w/ BN Tuning | **58.93** | **47.22** |
| ResNet50 | SimpleCIL | 62.91 | 52.84 |
| | APER w/ Finetune | **64.99** | 54.32 |
| | APER w/ SSF | 64.57 | **54.71** |
| | APER w/ BN Tuning | 64.56 | 54.67 |
| ResNet101 | SimpleCIL | 69.06 | 59.22 |
| | APER w/ Finetune | **71.93** | **62.00** |
| | APER w/ SSF | 70.89 | 60.59 |
| | APER w/ BN Tuning | 70.91 | 60.65 |
| ResNet152 | SimpleCIL | 72.13 | 61.62 |
| | APER w/ Finetune | **74.04** | **63.32** |
| | APER w/ SSF | 73.44 | 62.93 |
| | APER w/ BN Tuning | 73.45 | 62.89 |

Table 51: Results on CIFAR100 B0 Inc5 with pre-trained ResNet.

| Backbone | Method | $\bar{\mathcal{A}}$ | $\mathcal{A}_B$ |
|---|---|---|---|
| ResNet18 | SimpleCIL | 55.05 | 40.80 |
| | APER w/ Finetune | 63.82 | 51.27 |
| | APER w/ SSF | 63.38 | 50.72 |
| | APER w/ BN Tuning | **64.56** | **52.76** |
| ResNet50 | SimpleCIL | 63.20 | 50.59 |
| | APER w/ Finetune | 68.44 | 56.40 |
| | APER w/ SSF | **68.55** | **56.57** |
| | APER w/ BN Tuning | 66.28 | 53.52 |
| ResNet101 | SimpleCIL | 62.61 | 49.96 |
| | APER w/ Finetune | 66.36 | 54.88 |
| | APER w/ SSF | 66.59 | **54.92** |
| | APER w/ BN Tuning | **66.65** | 54.16 |
| ResNet152 | SimpleCIL | 62.67 | 49.36 |
| | APER w/ Finetune | **66.77** | **53.27** |
| | APER w/ SSF | 66.69 | **53.27** |
| | APER w/ BN Tuning | 65.51 | 51.61 |

Table 52: Results on CUB200 B0 Inc5 with pre-trained ResNet.

| Backbone | Method | $\bar{\mathcal{A}}$ | $\mathcal{A}_B$ |
|---|---|---|---|
| ResNet18 | SimpleCIL | 40.23 | 30.07 |
| | APER w/ Finetune | 49.54 | 38.83 |
| | APER w/ SSF | **49.57** | **38.87** |
| | APER w/ BN Tuning | 49.54 | 38.80 |
| ResNet50 | SimpleCIL | 50.35 | 43.00 |
| | APER w/ Finetune | **55.16** | **46.70** |
| | APER w/ SSF | 54.93 | 46.63 |
| | APER w/ BN Tuning | 54.91 | 46.50 |
| ResNet101 | SimpleCIL | 52.71 | 45.03 |
| | APER w/ Finetune | 59.58 | **51.97** |
| | APER w/ SSF | 59.43 | 51.72 |
| | APER w/ BN Tuning | **59.63** | **51.97** |
| ResNet152 | SimpleCIL | 54.48 | 47.33 |
| | APER w/ Finetune | **60.25** | **53.27** |
| | APER w/ SSF | 59.88 | 52.97 |
| | APER w/ BN Tuning | 59.99 | 53.10 |

Table 53: Results on ImageNet-R B0 Inc5 with pre-trained ResNet.

| Backbone | Method | $\bar{\mathcal{A}}$ | $\mathcal{A}_B$ |
|---|---|---|---|
| ResNet18 | SimpleCIL | 19.46 | 10.80 |
| | APER w/ Finetune | 23.83 | 12.64 |
| | APER w/ SSF | 23.79 | 12.90 |
| | APER w/ BN Tuning | **25.07** | **14.61** |
| ResNet50 | SimpleCIL | 35.11 | 22.32 |
| | APER w/ Finetune | **37.20** | **23.90** |
| | APER w/ SSF | 37.20 | 23.83 |
| | APER w/ BN Tuning | 34.86 | 22.25 |
| ResNet101 | SimpleCIL | 38.13 | 23.57 |
| | APER w/ Finetune | 40.03 | **26.07** |
| | APER w/ SSF | **40.25** | 26.00 |
| | APER w/ BN Tuning | 38.45 | 24.88 |
| ResNet152 | SimpleCIL | 39.98 | 26.66 |
| | APER w/ Finetune | 42.80 | **29.36** |
| | APER w/ SSF | **43.31** | 29.30 |
| | APER w/ BN Tuning | 39.69 | 26.27 |

Table 54: Results on ImageNet-A B0 Inc5 with pre-trained ResNet.

| Backbone | Method | $\bar{\mathcal{A}}$ | $\mathcal{A}_B$ |
|---|---|---|---|
| ResNet18 | SimpleCIL | 36.31 | 22.22 |
| | APER w/ Finetune | 42.98 | 28.64 |
| | APER w/ SSF | 42.93 | 28.46 |
| | APER w/ BN Tuning | **43.11** | **28.68** |
| ResNet50 | SimpleCIL | 52.47 | 37.67 |
| | APER w/ Finetune | 54.35 | **39.91** |
| | APER w/ SSF | **54.36** | 39.79 |
| | APER w/ BN Tuning | 54.23 | 39.85 |
| ResNet101 | SimpleCIL | 52.95 | 38.05 |
| | APER w/ Finetune | 54.85 | 40.30 |
| | APER w/ SSF | **54.88** | 40.28 |
| | APER w/ BN Tuning | 54.85 | **40.36** |
| ResNet152 | SimpleCIL | 53.88 | 39.42 |
| | APER w/ Finetune | **55.99** | **41.96** |
| | APER w/ SSF | 55.63 | 41.52 |
| | APER w/ BN Tuning | 55.80 | 41.70 |

Table 55: Results on ObjectNet B0 Inc5 with pre-trained ResNet.

| Backbone | Method | $\bar{\mathcal{A}}$ | $\mathcal{A}_B$ |
|---|---|---|---|
| ResNet18 | SimpleCIL | 53.70 | 43.79 |
| | APER w/ Finetune | 56.19 | 45.36 |
| | APER w/ SSF | **61.50** | 52.45 |
| | APER w/ BN Tuning | 61.49 | **52.60** |
| ResNet50 | SimpleCIL | 65.41 | 54.97 |
| | APER w/ Finetune | 66.75 | 55.51 |
| | APER w/ SSF | **67.50** | **56.91** |
| | APER w/ BN Tuning | 67.47 | 56.89 |
| ResNet101 | SimpleCIL | 65.20 | 54.90 |
| | APER w/ Finetune | **68.53** | 56.79 |
| | APER w/ SSF | 67.68 | 56.98 |
| | APER w/ BN Tuning | 67.93 | **57.18** |
| ResNet152 | SimpleCIL | 64.93 | 54.65 |
| | APER w/ Finetune | 66.43 | 55.07 |
| | APER w/ SSF | 67.18 | 56.61 |
| | APER w/ BN Tuning | **67.29** | **56.64** |

Table 56: Results on OmniBenchmark B0 Inc5 with pre-trained ResNet.

