# OpenReview forum: "Revisiting Class-Incremental Learning with Pre-Trained Models: Generalizability and Adaptivity are All You Need"
_ICLR.cc/2024/Conference — ICLR 2024 Conference Withdrawn Submission_

### Official Review · Reviewer_rLJS · 2023-10-24

**Soundness:** 3 good
**Presentation:** 3 good
**Contribution:** 3 good
**Rating:** 6
**Confidence:** 5

**Summary:**

This paper tackles the class-incremental learning problem with the help of pre-trained models. A simple baseline is evaluated with pre-trained models and it shows superior performance compared to other state-of-the-art methods. It can be further improved with model adaptation with several different existing adaptation methods. The embeddings from the pre-trained model and the adapted model are concatenated for classification. To ensure fair comparisons, four new benchmarks are proposed for testing all methods.

**Strengths:**

(1) The paper is well-written and easy to follow.
(2) It is interesting to see that a simple baseline can be good enough for many CIL settings.
(3) Figures 1-2 clearly demonstrate the motivations behind the main idea.
(4) Several popular options for adaptation are explained and evaluated.
(5) To strike a good balance, both pre-trained embeddings and adapted embeddings are fused for final decision-making.

**Weaknesses:**

(1) The overall method is simple. The main idea is to fuse the knowledge from two embeddings, which is new with pre-trained models, though.
(2) The performance in Table 1 shows that SimpleCIL is very competitive in some cases. This means that the proposed adaptation methods are not fully explored. More work can be done with adaptation to further improve the performance.
(3) The method assumes the adapted model is only learned in the first step, which is a weak point in some settings when the first phase only contains little information. Also, I am wondering why it is not adapted incrementally for the downstream tasks.

**Questions:**

(1) I would suggest authors discuss more failure cases and weaknesses of the proposed methods, which can be helpful for the readers.
(2) I would like to see responses raised in the Weaknesses section.

---

### Official Review · Reviewer_6wZj · 2023-10-29

**Soundness:** 2 fair
**Presentation:** 2 fair
**Contribution:** 2 fair
**Rating:** 3
**Confidence:** 5

**Summary:**

This paper investigates the task of continual learning (CL) from pre-trained models (PTMs), for which they propose a simple combination of base PTM and task-adapted (finetuned) variant. On two standard CL benchmarks, and four additional benchmarks with larger distribution shifts to model pretrainings, the authors report significant improvements compared to current rehearsal-free approaches, in particulary prompt-style adaptation approaches.

**Strengths:**

In my eyes, the major strength of this paper is the reported performance gap compared to other comparable methods. However, I am unsure about the significance (see below).

**Weaknesses:**

I have several issues and questions with the method and experiments provided in this paper, which I have ordered based on importance to be addressed.

__Larger Issues__

* My main problem stems with the comparability of the proposed method, and the overall novelty of the provided SimpleCIL/APER approach:
	* SimpleCIL is already not comparable to the benchmark approaches, as it stores and re-uses data from the training stream (albeit stored as averaged embeddings) at test time. As weights are frozen and the model does not change, these effectively equates to storing and re-using data context at test-time, which goes against the replay-free setting studied in e.g. all prompt-based baselines.
	* The same also holds for the proposed approach APER, even if embeddings are concatenated.
	* This makes comparison very difficult, and to me seems like the primary reason for the large performance gaps (as simple concatenation of embeddings alone shouldn't result in such a large performance gap).
	* It is essentially akin to retrieval-based/augmented approaches to continual learning, as e.g. studied in [3].
	* The combination of finetuned variant and base model in order to retain a better overall generalizing model very closely resembles insights from [1], showcasing how linear interpolation between finetuned pretrained model variants can offer models that can generalize across multiple tasks. This has also been extended to the continual learning domain in e.g. [2]. As model adaptation is inherently allowed in APER, it would be great to have a comparision to e.g. the interpolation-based setup in [2].

* Touching again on the point of comparability, some variants of APER (e.g. with Adapter or just finetuned) also requires two forward passes (which remains comparable to at least prompt-based settings), but also needs to store more parameters (all adapter weights, or the second fully finetuned model).

* As far as I understand, the authors also only perform adaptation on the first incremental data stream task. This is a very strong assumption to make only fitting to particular benchmark scenarios - both with respect to the length of the first task, but also the expected shifts across the overall data stream and the overall diversity. It would be great to get some insights on this aspect.

* The paper consistently juggles between PTMs being sufficient generalizable (e.g. "born with generalizability, which can be directly transferred to ... tasks ...) and not generalizing at all (when it comes to the introduction of the new benchmarks). While it does become clear that a sweet spot between both is what the authors are going for, jumping between both extremes does make things  more difficult to place. Consequently, I am quite curious about the relevance of the new benchmark motivation, with the goal of incorporating much higher shifts with respect to the pretraining data s.t. exisiting approaches that do not adapt "as much" fail. In these scenarios, where pretraining becomes less and less useful, I think continually training a model from scratch becomes a crucial baseline again.

* For the evaluation of different PTMs, before making the claim that pre-trained ViTs generalize better, it's important to clarify differences in training compute and the provided pretraining data. Are the ResNets also pretrained on IN21k? And to a comparable extend as the ViTs?



__Lesser Issues__

* The experiments conducted in Figure 2 provide insights that echo the results of previous work, which showcase that even well generalizing base models can be adapted to continual learning tasks. Would be great to reference these for completeness (e.g. [2,4,5])

* The formatting in this paper is off in a few places, and is overly dense, e.g.:
	* The "Adaptivity and Generalizability in CIL" title looks weirdly placed/formatted.
	* Vspaces are placed too aggressively
	* Parts of the text have different font sizes / spacing (e.g. p.5), which is not consistent with the overall ICLR formatting.

The t-SNE visualizations in Fig. 7 appear quite redundant, as no new or actionable information is provided, and could be simply removed to easen up on the paper density.

* The paper often makes it sound like new benchmark datasets are proposed, as opposed to just testing the model on existing ones, which simply have not been (as often) studied for CL (e.g. ImageNet-R was already studied in e.g. DualPrompt and follow-up works).

* Page 2 - Top: "..., we find essential differences between these protocols..." > This sounds like the authors are the first to figure out the difference between training (continually) from scratch and using a pretrained model, although the differentiation is common knowledge, as e.g. even addressed in the referenced work by Kumar et al. 2022.


* Wording:
	* "panacea" seems unnecessary obfuscating as a word - why not just remedy?
	* "overlapping between (...) tasks" - should potentially clarify that this likely refers to conceptual and not actual sample overlap?


__References__
* [1] "Robust Fine-Tuning of Zero-Shot Models", Wortsmann et al. 2022
* [2] "Momentum-based Weight Interpolation of Strong Zero-Shot Models for Continual Learning", Stojanovski et al. 2022
* [3] "Online Continual Learning Without the Storage Constraint", Prabhu et al. 2023
* [4] "An empirical investigation of the role of pre-training in lifelong learning", Mehta et al. 2022
* [5] "Effect of scale on catastrophic forgetting in neural networks", Ramasesh et al. 2022

**Questions:**

I am happy to adjust my rating if the larger issues listed above are adequately addressed - in particular with respect to the comparability of the experimental results, and the overall novelty of the proposed approach.

---

### Official Review · Reviewer_o8FW · 2023-10-31

**Soundness:** 4 excellent
**Presentation:** 3 good
**Contribution:** 2 fair
**Rating:** 5
**Confidence:** 4

**Summary:**

The authors discuss the usage of pre-trained models for class-incremental learning, given their strong representations. By starting from a strong starting point, this approach can learn new classes much faster than training from scratch, and instead the authors focus on generalizability of the existent strong features and how to quickly adapt the model to new classes. The authors investigate a wide selection of adaptation strategies within their framework, demonstrating strong class-incremental learning performance on a number of benchmarks.

**Strengths:**

## S1. Diversity of adaptation methods
The proposed method is formulated quite generally, with the ability to incorporate any general adaptation method. In Section 4, quite a wide range of prior adaptation methods are listed out, and the experiments impressively provide results for all of them.

## S2. Datasets and baselines
The authors show experimental results on a good number of datasets/benchmarks: CIFAR, CUB, ImageNet-R, ImageNet-A, ObjectNet, OmniBench, and VTAB. Results are reported as both final accuracy, as well average accuracy, which are standard metrics of incremental learning. A moderate selection of baseline methods are also selected, and across datasets, at least one of the APER + [Adaptation technique] methods tends to be the best.

## S3. Ablations, Appendix
There are a fairly good set of ablation studies: smaller dimensionality features, breaking down performance of the submodules of APER, and starting from different pre-trained models. There is also a very extensive appendix, providing more extensive background and implementation details, as well as additional experiments and ablations. Even if the methodology or insights themselves aren’t necessarily that novel, the thoroughness of the compiled information and experiments may remain a useful reference for anyone running incremental learning experiments with pre-trained models + adapt strategy.

**Weaknesses:**

# W1. Novelty compared to prior works
It’s not exactly clear what the novel contributions of this particular work are. As the Related Works amply breaks down, class-incremental learning starting from a pre-trained model has already been explored before (see also [a]). The generalizability of pre-trained large-scale models is a well-known characteristic by the community at this point, with the impressive ability to generalize in few-shot or even zero-shot ways demonstrated repeatedly; being also able to generalize for continual learning is not surprising. Even before the recent age of large-scale foundation models, making small adaptations to larger models for in class-incremental learning has been studied [b, c] before, including ImageNet pre-trained models. These earlier works are highly related but not referenced.

I also find it generally inappropriate to claim the simple baseline SimpleCIL as a contribution of this paper. Prototypes for incremental learning is a very common approach; a quick search of “prototype incremental learning” yields a plethora of such works, such as [d,e].

Finally, while it is impressive how many adaptation methods the authors study, all of these techniques are from prior work. As a result, the overall paper is a combination of previously known techniques.

# W2. Methodology
It’s not clear to me how principled freezing the model after the fine-tuning on just the first task is. This would appear to render the method especially well-tuned to the first task, more than the others. In incremental learning, task ordering is often somewhat arbitrary, so singling out one task (in this case the first) for special treatment seems inequitable.

# W3. Results
It’s not ultimately clear from Table 1 what the best adaptation technique is. Adapter seems to do best among the adaptation techniques on the datasets that authors describe as not “large domain gap”. The other methods (e.g. SSF, VPT-Deep) seem to do better for the “large domain gap datasets”, but some of them are comparable or beaten by the baselines for these “small domain gap datasets”.

Also, the APER plots in Figure 4 are mostly parallel, mainly with differences in task 1 performance. Because of model freezing after task 1, an APER variant’s performance is largely determined by how well the model was able to adapt to the first task, and then the rate of performance degradation over time is otherwise constant.

## Other minor comments:
- Eq 1: If this is empirical risk, then there is a 1/N term missing (where N is the total number of samples)
- pg 5: “APER will degrade to SimpleCIL, which guarantees the performance lower bound.” <= Is this a guaranteed lower bound? I envision it should be possible that an adaptation strategy could potentiall be worse than doing nothing.
- pg 6: “the superiority [of what?].”
- Section 5.1: Despite what the paper claims, it’s debatable how much of a domain shift these datasets have from ImageNet. I wouldn’t characterize it as a “large domain gap”.
- Section 5.3: “Since the feature of APER is aggregated with PTM and adapted model, which is twice that of a PTM” <= sentence fragment

[a] CVPR 2021: “Class-Incremental Learning with Strong Pre-Trained Models”\
[b] ECCV 2020: “Side-tuning: a baseline for network adaptation via additive side networks” \
[c] CVPR 2021: “Efficient feature transformations for discriminative and generative continual learning” \
[d] CVPR 2017: “iCaRL: Incremental Classifier and Representation Learning”\
[e] CVPR 2021: “Few-Shot Incremental Learning with Continually Evolved Classifiers”

**Questions:**

N/A

---

### Official Review · Reviewer_PEMr · 2023-11-01

**Soundness:** 3 good
**Presentation:** 3 good
**Contribution:** 3 good
**Rating:** 5
**Confidence:** 3

**Summary:**

The proposed approach involves leveraging a Pre-Trained Model (PTM) and an adapted model. The PTM provides a stable knowledge base, capturing generic features from a wide range of classes. The adapted model, on the other hand, is fine-tuned on incremental data to capture specialized features of new classes. The features from both models are concatenated to form a comprehensive representation. The paper conducts extensive experiments using various image datasets to validate the effectiveness of their approach.

**Strengths:**

* The authors address the domain gap problem between the pre-trained model and the incremental data in two ways: adapting the pre-trained model (PTM) to the initial incremental data, learning task-specific features. This adapted model is then merged with the PTM, creating a unified network that maintains generalizability across future tasks. By combining these two approaches, the authors aim to achieve a balance between adaptivity and generalizability.

* The authors present a well-thought-out strategy to balance the need for adaptivity to new classes with the necessity of maintaining generalizability across tasks.

**Weaknesses:**

* Distribution gap between the pre-trained and downstream datasets can lead to inaccurate prototypes if the embeddings from the pre-trained model and the adapted model are not homogeneous.

* Concatenating features does not ensure true adaptability to new classes, as it might lead to a model that is more of an ensemble of old and new knowledge rather than a seamlessly adapted model, raising the following concerns:

(1) Potential for Redundancy: There is a potential for redundancy in the concatenated features, especially if the PTM and the adapted model capture similar information.

(2) Ensemble-Like Behavior: The concatenated features create a scenario where the model behaves like an ensemble of the PTM and the adapted model. True adaptability would require the model to integrate new knowledge, adjusting its internal representations accordingly. To address this issue, they propose to use a prototype-based classifier that considers the distance between the input embedding and the class centers. One point that raises skepticism about the prototype-based classifier is its reliance on the average embedding to represent class prototypes, assuming the average embedding is a robust and accurate representation of a class.

(3) Lack of Interaction Between Old and New Knowledge: The concatenation approach lacks a mechanism for interaction between the old knowledge (represented by the PTM’s features) and the new knowledge (represented by the adapted model’s features). Adaptability might require a more dynamic integration of old and new knowledge, allowing the model to restructure its internal representations in light of new information.

**Questions:**

* The authors describe a two-stage adaptation process in which the pre-trained model is first fine-tuned on the current task and then further adapted using a parameter-efficient tuning algorithm. However, they do not provide a detailed explanation of how the adaptation process works or how the choice of adaptation algorithm might affect the performance of the proposed framework.

* The authors use a pre-trained vision transformer model as the starting point for their framework. However, they do not provide a detailed explanation of why they chose this particular model or how its architecture and pre-training task might affect the performance of the proposed framework.

* Though CIL is an evolving field and they are many works that might be overlooked, there are two papers that are closely related and there is no indication of them:
“Continual Learning with Foundation Models: An Empirical Study of Latent Replay” http://arxiv.org/abs/2205.00329
“An empirical investigation of the role of pre-training in lifelong learning”
https://arxiv.org/abs/2112.09153